# BENIGN OVERFITTING IN TWO-LAYER ReLU CONVOLUTIONAL NEURAL NETWORKS FOR XOR DATA

## ABSTRACT

Modern deep learning models are usually highly over-parameterized so that they can overfit the training data. Surprisingly, such overfitting neural networks can usually still achieve high prediction accuracy. To study this "benign overfitting" phenomenon, a line of recent works has theoretically studied the learning of linear models and two-layer neural networks. However, most of these analyses are still limited to the very simple learning problems where the Bayes-optimal classifier is linear. In this work, we investigate a class of XOR-type classification tasks with label-flipping noises. We show that, under a certain condition on the sample complexity and signal-to-noise ratio, an over-parameterized ReLU CNN trained by gradient descent can achieve near Bayes-optimal accuracy. Moreover, we also establish a matching lower bound result showing that when the previous condition is not satisfied, the prediction accuracy of the obtained CNN is an absolute constant away from the Bayes-optimal rate. Our result demonstrates that CNNs have a remarkable capacity to efficiently learn XOR problems, even in the presence of highly correlated features.

## 1 INTRODUCTION

Modern deep neural networks are often highly over-parameterized, with the number of parameters far exceeding the number of samples in the training data. This can lead to overfitting, where the model performs well on the training data but poorly on unseen test data. However, an interesting phenomenon, known as "benign overfitting", has been observed, where these models maintain remarkable performance on the test data despite the potential for overfitting (Neyshabur et al., 2019; Zhang et al., 2021). Bartlett et al. (2020) theoretically proved this phenomenon in linear regression and coined the term "benign overfitting".

There has been a recent surge of interest in studying benign overfitting from a theoretical perspective. A line of recent works provided significant insights under the settings of linear/kernel/random feature models (Belkin et al., 2019; 2020; Bartlett et al., 2020; Chatterji & Long, 2021; Hastie et al., 2022; Montanari & Zhong, 2022; Mei & Montanari, 2022; Tsigler & Bartlett, 2023). However, the analysis of benign overfitting in neural networks under gradient descent is much more challenging due to the non-convexity in the optimization and non-linearity of activation functions. Nevertheless, several recent works have made significant progress in this area. For instance, Frei et al. (2022) provided an upper bound of risk with smoothed leaky ReLU activation when learning the log-concave mixture data with label-flipping noise. By proposing a method named "signal-noise decomposition", Cao et al. (2022) established a condition for sharp phase transition between benign and harmful overfitting in the learning of two-layer convolutional neural networks (CNNs) with activation functions $\text{ReLU}^q$ ($q > 2$). Kou et al. (2023) further extended the analysis to ReLU neural networks, and established a condition for such a sharp phase transition with more general conditions. Despite the recent advances in the theoretical study of benign overfitting, the existing studies are still mostly limited to very simple learning problems where the Bayes-optimal classifier is linear.

In this paper, we study the benign overfitting phenomenon of two-layer ReLU CNNs in more complicated learning tasks where linear models will provably fail. Specifically, we consider binary classification problems where the label of the data is jointly determined by the presence of two types of signals with an XOR relation. We show that for this XOR-type of data, any linear predictor will only achieve 50% test accuracy. On the other hand, we establish tight upper and lower bounds of

the test error achieved by two-layer CNNs trained by gradient descent, and demonstrate that benign overfitting can occur even with the presence of label-flipping noises.

The contributions of our paper are as follows:

1. We establish matching upper and lower bounds on the test error of two-layer CNNs trained by gradient descent in learning XOR-type data. Our test error upper bound suggests that when the sample size, signal strength, noise level, and dimension meet a certain condition, then the test error will be nearly Bayes-optimal. This result is also demonstrated optimal by our lower bound, which shows that under a complementary condition, the test error will be a constant gap away from the Bayes-optimal rate. These results together demonstrate a similar "sharp" phase transition between benign and harmful overfitting of CNNs in learning XOR-type data, as previously shown in works by Cao et al. (2022) and Kou et al. (2023).

2. Our results demonstrate that CNNs can efficiently learn complicated data such as the XOR-type data. Notably, the conditions on benign/harmful overfitting we derive for XOR-type data are the same as the corresponding conditions established for linear logistic regression (Cao et al., 2021), two-layer fully connected networks (Frei et al., 2022) and two-layer CNNs (Kou et al., 2023), although these previous results on benign/harmful overfitting are established where the Bayes-optimal classifiers are linear. Therefore, our results imply that two-layer ReLU CNNs can learn XOR-type data as efficiently as using linear logistic regression or two-layer neural networks to learn Gaussian mixtures. Notably, learning the XOR-type data relies on different CNN filters to learn different features, while in previous works, the optimal case is in fact when all neurons are (almost) parallel to a single signal direction. Therefore, the result in this paper can better reveal the strength of the non-linear structure of neural networks.

3. Our work also considers the regime when the features in XOR-type data are highly correlated. To overcome the technical challenges caused by the high correlation in the training process, we introduce a novel proof technique called "virtual sequence comparison", to enable analyzing the learning of such highly correlated XOR-type data. We believe that this novel proof technique can find wide applications in related studies and therefore may be of independent interest.

A concurrent work (Xu et al., 2023) studies using two-layer fully connected neural networks to learn an XOR problem, where for fixed vectors $\mathbf{u}$ and $\mathbf{v}$, the two classes of data are generated from Gaussian clusters $N(\pm\mathbf{u}, \mathbf{I})$ and $N(\pm\mathbf{v}, \mathbf{I})$ respectively. The authors show that gradient descent can train a neural network to achieve benign overfitting, and can exhibit a "grokking" phenomenon. There are several differences between (Xu et al., 2023) and our work. First, they assume that the "signal vectors" $\mathbf{u}$ and $\mathbf{v}$ are orthogonal. In comparison, we can cover the case where the angle between $\mathbf{u}$ and $\mathbf{v}$ is small, assuming that the signal vectors and Gaussian noises are on different patches of the data. Moreover, they rely on a condition that the signal strength ($\|\mathbf{u}\|_2, \|\mathbf{v}\|_2$) increases with the sample size $n$, while our result indicates that if $n$ increases, then a smaller signal strength may be sufficient to achieve benign overfitting. In addition, they focus on the neural networks trained in the first $\sqrt{n}$-th iterations, while our results are for CNNs trained until convergence. At last, they mainly focus on the upper bound of test error, while our result gives matching upper and lower bounds.

## 1.1 ADDITIONAL RELATED WORKS

In this section, we introduce the related works in detail.

**Benign overfitting in linear/kernel/random feature models.** A key area of research aimed at understanding benign overfitting involves theoretical analysis of test error in linear/kernel/random feature models. Wu & Xu (2020); Mel & Ganguli (2021); Hastie et al. (2022) explored excess risk in linear regression, where the dimension and sample size are increased to infinity while maintaining a fixed ratio. These studies showed that the risk decreases in the over-parameterized setting relative to this ratio. In the case of random feature models, Liao et al. (2020) delved into double descent when the sample size, data dimension, and number of random features maintain fixed ratios, while Adlam et al. (2022) extended the model to include bias terms. Additionally, Misiakiewicz (2022); Xiao et al. (2022); Hu & Lu (2022) demonstrated that the risk curve of certain kernel predictors can exhibit multiple descent concerning the sample size and data dimension.

**Benign overfitting in neural networks.** In addition to theoretical analysis of test error in linear/kernel/random feature models, another line of research explores benign overfitting in neural networks. For example, Zou et al. (2021) investigated the generalization performance of constant stepsize stochastic gradient descent with iterate averaging or tail averaging in the over-parameterized

regime. Montanari & Zhong (2022) investigated two-layer neural networks and gave the interpolation of benign overfitting in the NTK regime. Meng et al. (2023) investigated gradient regularization in the over-parameterized setting and found benign overfitting even under noisy data. Additionally, Chatterji & Long (2023) bounded the excess risk of deep linear networks trained by gradient flow and showed that "randomly initialized deep linear networks can closely approximate or even match known bounds for the minimum $\ell_2$-norm interpolant".

**Learning the XOR function.** In the context of feedforward neural networks, Hamey (1998) pointed out that there is no existence of local minima in the XOR task. XOR-type data is particularly interesting as it is not linearly separable, making it sufficiently complex for backpropagation training to become trapped without achieving a global optimum. The analysis by Brutzkus & Globerson (2019) focused on XOR in ReLU neural networks that were specific to two-dimensional vectors with orthogonal features. Under the quadratic NTK regime, Bai & Lee (2020); Chen et al. (2020) proved the learning ability of neural networks for the XOR problems.

## 1.2 NOTATION

Given two sequences $x_n$ and $y_n$, we denote $x_n = O(y_n)$ if for some absolute constant $C_1 > 0$ and $N > 0$ such that $|x_n| \le C_1 |y_n|$ for all $n \ge N$. We denote $x_n = \Omega(y_n)$ if $y_n = O(x_n)$, and denote $x_n = \Theta(y_n)$ if $x_n = O(y_n)$ and $x_n = \Omega(y_n)$ both hold. We use $\widetilde{O}(\cdot)$, $\widetilde{\Omega}(\cdot)$, and $\widetilde{\Theta}(\cdot)$ to hide logarithmic factors in these notations, respectively. We use $\mathbf{1}(\cdot)$ to denote the indicator variable of an event. We say $y = \text{poly}(x_1, ..., x_k)$ if $y = O(\max\{|x_1|, ..., |x_k|\}^D)$ for some $D > 0$, and $y = \text{polylog}(x)$ if $y = \text{poly}(\log(x))$.

## 2 LEARNING XOR-TYPE DATA WITH TWO-LAYER CNNS

In this section, we introduce in detail the XOR-type learning problem and the two-layer CNN model considered in this paper. We first introduce the definition of an "XOR distribution".

**Definition 2.1** *Let* $\mathbf{a}, \mathbf{b} \in \mathbb{R}^d \backslash \{\mathbf{0}\}$ *with* $\mathbf{a} \perp \mathbf{b}$ *be two fixed vectors. For* $\boldsymbol{\mu} \in \mathbb{R}^d$ *and* $\overline{y} \in \{\pm 1\}$, *we say that* $\boldsymbol{\mu}$ *and* $\overline{y}$ *are jointly generated from distribution* $\mathcal{D}_{\text{XOR}}(\mathbf{a}, \mathbf{b})$ *if the pair* $(\boldsymbol{\mu}, \overline{y})$ *is randomly and uniformly drawn from the set* $\{(\mathbf{a} + \mathbf{b}, +1), (-\mathbf{a} - \mathbf{b}, +1), (\mathbf{a} - \mathbf{b}, -1), (-\mathbf{a} + \mathbf{b}, -1)\}$.

Definition 2.1 gives a generative model of $\boldsymbol{\mu} \in \mathbb{R}^d$ and $\overline{y} \in \{\pm 1\}$ and guarantees that the samples satisfy an XOR relation with respect to two basis vectors $\mathbf{a}, \mathbf{b} \in \mathbb{R}^d$. When $d = 2$, $\mathbf{a} = [1, \ 0]^\top$ and $\mathbf{b} = [0, \ 1]^\top$, Definition 2.1 recovers the XOR data model studied by Brutzkus & Globerson (2019). We note that Definition 2.1 is more general than the standard XOR model, especially because it does not require that the two basis vectors have equal lengths. Note that in this model, $\boldsymbol{\mu} = \pm(\mathbf{a} + \mathbf{b})$ when $\overline{y} = 1$, and $\boldsymbol{\mu} = \pm(\mathbf{a} - \mathbf{b})$ when $\overline{y} = -1$. When $\|\mathbf{a}\|_2 \ne \|\mathbf{b}\|_2$, it is easy to see that the two vectors $\mathbf{a} + \mathbf{b}$ and $\mathbf{a} - \mathbf{b}$ are not orthogonal. In fact, the angles between $\pm(\mathbf{a} + \mathbf{b})$ and $\pm(\mathbf{a} - \mathbf{b})$ play a key role in our analysis, and the classic setting where $\mathbf{a} + \mathbf{b}$ and $\mathbf{a} - \mathbf{b}$ are orthogonal is a relatively simple case covered in our analysis.

Although $\mathcal{D}_{\text{XOR}}(\mathbf{a}, \mathbf{b})$ and its simplified versions are classic models, we note that $\mathcal{D}_{\text{XOR}}(\mathbf{a}, \mathbf{b})$ alone may not be a suitable model to study the benign overfitting phenomena: the study of benign overfitting typically requires the presence of certain types of noises. In this paper, we consider a more complicated XOR-type model that introduces two types of noises: (i) the $\boldsymbol{\mu}$ vector is treated as a "signal patch", which is hidden among other "noise patches"; (ii) a label flipping noise is applied to the clean label $\overline{y}$ to obtain the observed label $y$. The detailed definition is given as follows.

**Definition 2.2** *Let* $\mathbf{a}, \mathbf{b} \in \mathbb{R}^d \backslash \{\mathbf{0}\}$ *with* $\mathbf{a} \perp \mathbf{b}$ *be two fixed vectors. Then each data point* $(\mathbf{x}, y)$ *with* $\mathbf{x} = [\mathbf{x}^{(1)\top}, \mathbf{x}^{(2)\top}]^\top \in \mathbb{R}^{2d}$ *and* $y \in \{\pm 1\}$ *is generated from* $\mathcal{D}$ *as follows:*
1. *$\boldsymbol{\mu} \in \mathbb{R}^d$ and $\overline{y} \in \{\pm 1\}$ are jointly generated from $\mathcal{D}_{\text{XOR}}(\mathbf{a}, \mathbf{b})$.*
2. *One of $\mathbf{x}^{(1)}, \mathbf{x}^{(2)}$ is randomly selected and then assigned as $\boldsymbol{\mu}$; the other is assigned as a randomly generated Gaussian vector $\boldsymbol{\xi} \sim N(\mathbf{0}, \sigma_p^2 \cdot (\mathbf{I} - \mathbf{a}\mathbf{a}^\top/\|\mathbf{a}\|^2 - \mathbf{b}\mathbf{b}^\top/\|\mathbf{b}\|^2))$.*
3. *The observed label $y \in \{\pm 1\}$ is generated with $\mathbb{P}(y = \overline{y}) = 1 - p$, $\mathbb{P}(y = -\overline{y}) = p$.*

Definition 2.2 divides the data input into two patches, assigning one of them as a signal patch and the other as a noise patch. In the noise patch, the covariance matrix is defined so that $\boldsymbol{\xi}$ is orthogonal

to the signal vector $\boldsymbol{\mu}$, which helps simplify our analysis. This type of model has been explored in previous studies Cao et al. (2022); Jelassi & Li (2022); Kou et al. (2023); Meng et al. (2023). However, our data model is much more challenging to learn due to the XOR relation between the signal patch $\boldsymbol{\mu}$ and the clean label $\overline{y}$. Specifically, we note that the data distributions studied in the previous works Cao et al. (2022); Jelassi & Li (2022); Kou et al. (2023); Meng et al. (2023) share the common property that the Bayes-optimal classifier is linear. On the other hand, for $(\mathbf{x}, y) \sim \mathcal{D}$ in Definition 2.2, it is easy to see that

$$(\mathbf{x}, y) \stackrel{d}{=} (-\mathbf{x}, y), \text{ and therefore } \mathbb{P}_{(\mathbf{x},y)\sim\mathcal{D}}(y \cdot \langle \boldsymbol{\theta}, \mathbf{x} \rangle > 0) = 1/2 \text{ for any } \boldsymbol{\theta} \in \mathbb{R}^{2d}. \tag{2.1}$$

In other words, all linear predictors will fail to learn the XOR-type data $\mathcal{D}$.

We consider using a two-layer CNN to learn the XOR-type data model $\mathcal{D}$ defined in Definition 2.2, where the CNN filters go over each patch of a data input. We focus on analyzing the training of the first-layer convolutional filters, and fixing the second-layer weights. Specifically, define

$$f(\mathbf{W}, \mathbf{x}) = \sum_{j=\pm 1} j \cdot F_j(\mathbf{W}_j, \mathbf{x}), \text{ where } F_j(\mathbf{W}_j, \mathbf{x}) = \frac{1}{m} \sum_{r=1}^{m} \sum_{p=1}^{2} \sigma(\langle \mathbf{w}_{j,r}, \mathbf{x}^{(p)} \rangle). \tag{2.2}$$

Here, $F_{+1}(\mathbf{W}_{+1}, \mathbf{x})$ and $F_{-1}(\mathbf{W}_{-1}, \mathbf{x})$ denote two parts of the CNN models with positive and negative second layer parameters respectively. Moreover, $\sigma(z) = \max\{0, z\}$ is the ReLU activation function, $m$ is the number of convolutional filters in each of $F_{+1}$ and $F_{-1}$. For $j \in \{\pm 1\}$ and $r \in [m]$, $\mathbf{w}_{j,r} \in \mathbb{R}^d$ denotes the weight of the $r$-th convolutional filter in $F_j$. We denote by $\mathbf{W}_j$ the collection of weights in $F_j$ and denote by $\mathbf{W}$ the total collection of all trainable CNN weights.

Given $n$ i.i.d. training data points $(\mathbf{x}_1, y_1), \ldots, (\mathbf{x}_n, y_n)$ generated from $\mathcal{D}$, we train the CNN model defined above by minimizing the objective function

$$L(\mathbf{W}) = \frac{1}{n} \sum_{i=1}^{n} \ell(y_i \cdot f(\mathbf{W}, \mathbf{x}_i)),$$

where $\ell(z) = \log(1 + \exp(-z))$ is the cross entropy loss function. We use gradient descent $\mathbf{w}_{j,r}^{(t+1)} = \mathbf{w}_{j,r}^{(t)} - \eta \nabla_{\mathbf{w}_{j,r}} L(\mathbf{W}^{(t)})$ to minimize the training loss $L(\mathbf{W})$. Here $\eta > 0$ is the learning rate, and $\mathbf{w}_{j,r}^{(0)}$ is given by Gaussian random initialization with each entry generated as $N(0, \sigma_0^2)$. Suppose that gradient descent gives a sequence of iterates $\mathbf{W}^{(t)}$. Our goal is to establish upper bounds on the training loss $L(\mathbf{W}^{(t)})$, and study the test error of the CNN, which is defined as

$$R(\mathbf{W}^{(t)}) := P_{(\mathbf{x},y)\sim\mathcal{D}}(yf(\mathbf{W}^{(t)}, \mathbf{x}) < 0).$$

## 3 MAIN RESULTS

In this section, we present our main theoretical results. We note that the signal patch $\boldsymbol{\mu}$ in Definition 2.2 takes values $\pm(\mathbf{a} + \mathbf{b})$ and $\pm(\mathbf{a} - \mathbf{b})$, and therefore we see that $\|\boldsymbol{\mu}\|_2^2 \equiv \|\mathbf{a}\|_2^2 + \|\mathbf{b}\|_2^2$ is not random. We use $\|\boldsymbol{\mu}\|_2$ to characterize the "signal strength" in the data model.

Our results are separated into two parts, focusing on different regimes regarding the angle between the two vectors $\pm(\mathbf{a} + \mathbf{b})$ and $\pm(\mathbf{a} - \mathbf{b})$. Note that by the symmetry of the data model, we can assume without loss of generality that the angle $\theta$ between $\mathbf{a} + \mathbf{b}$ and $\mathbf{a} - \mathbf{b}$ satisfies $0 \leq \cos\theta < 1$. Our first result focuses on a "classic XOR regime" where $0 \leq \cos\theta < 1/2$ is a constant. This case covers the classic XOR model where $\|\mathbf{a}\|_2 = \|\mathbf{b}\|_2$ and $\cos\theta = 0$. In our second result, we explore an "asymptotically challenging XOR regime" where $\cos\theta$ can be arbitrarily close to 1 with a rate up to $\widetilde{\Theta}(1/\sqrt{n})$. The two results are introduced in two subsections respectively.

### 3.1 THE CLASSIC XOR REGIME

In this subsection, we introduce our result on the classic XOR regime where $0 \leq \cos\theta < 1/2$. Our main theorem aims to show theoretical guarantees that hold with a probability at least $1 - \delta$ for some small $\delta > 0$. Meanwhile, our result aims to show that training loss $L(\mathbf{W}^{(t)})$ will converge below some small $\varepsilon > 0$. To establish such results, we require that the dimension $d$, sample size $n$, CNN width $m$, random initialization scale $\sigma_0$, label flipping probability $p$, and learning rate $\eta$ satisfy certain conditions related to $\varepsilon$ and $\delta$. These conditions are summarized below.

**Condition 3.1** *For certain $\varepsilon, \delta > 0$, suppose that*

1. *The dimension $d$ satisfies: $d = \widetilde{\Omega}(\max\{n^2, n\|\boldsymbol{\mu}\|_2^2 \sigma_p^{-2}\}) \cdot \mathrm{polylog}(1/\varepsilon) \cdot \mathrm{polylog}(1/\delta)$.*
2. *Training sample size $n$ and CNN width $m$ satisfy $m = \Omega(\log(n/\delta)), n = \Omega(\log(m/\delta))$.*
3. *Random initialization scale $\sigma_0$ satisfies: $\sigma_0 \le \widetilde{O}(\min\{\sqrt{n}/(\sigma_p d), n\|\boldsymbol{\mu}\|_2/(\sigma_p^2 d)\})$.*
4. *The label flipping probability $p$ satisfies: $p \le c$ for a small enough absolute constant $c > 0$.*
5. *The learning rate $\eta$ satisfies: $\eta = \widetilde{O}([\max\{\sigma_p^2 d^{3/2}/(n^2\sqrt{m}), \sigma_p^2 d/(nm)\}]^{-1})$.*
6. *The angle $\theta$ satisfies $\cos\theta < 1/2$.*

The first two conditions above on $d$, $n$, and $m$ are mainly to make sure that certain concentration inequality type results hold regarding the randomness in the data distribution and gradient descent random initialization. These conditions also ensure that the the learning problem is in a sufficiently over-parameterized setting, and similar conditions have been considered a series of recent works (Chatterji & Long, 2021; Frei et al., 2022; Cao et al., 2022; Kou et al., 2023). The condition on $\sigma_0$ ensures a small enough random initialization of the CNN filters to control the impact of random initialization in a sufficiently trained CNN model. The condition on learning rate $\eta$ is a technical condition for the optimization analysis.

The following theorem gives our main result under the classic XOR regime.

**Theorem 3.2** *For any $\varepsilon, \delta > 0$, if Condition 3.1 holds, then there exist constants $C_1, C_2, C_3 > 0$, such that with probability at least $1 - 2\delta$, the following results hold at $T = \Omega(nm/(\eta\varepsilon\sigma_p^2 d))$:*

1. *The training loss converges below $\varepsilon$, i.e., $L(\mathbf{W}^{(T)}) \le \varepsilon$.*
2. *If $n\|\boldsymbol{\mu}\|_2^4 \ge C_1\sigma_p^4 d$, then the CNN trained by gradient descent can achieve near Bayes-optimal test error: $R(\mathbf{W}^{(T)}) \le p + \exp(-C_2 n\|\boldsymbol{\mu}\|_2^4/(\sigma_p^4 d))$.*
3. *If $n\|\boldsymbol{\mu}\|_2^4 \le C_1\sigma_p^4 d$, then the CNN trained by gradient descent can only achieve sub-optimal error rate: $R(\mathbf{W}^{(T)}) \ge p + C_3$.*

The first result in Theorem 3.2 shows that under our problem setting, when learning the XOR-type data with two-layer CNNs, the training loss is guaranteed to converge to zero. This demonstrates the global convergence of gradient descent and ensures that the obtained CNN overfits the training data. Moreover, the second and the third results give upper and lower bounds of the test error achieved by the CNN under complimentary conditions. This demonstrates that the upper and the lower bounds are both tight, and $n\|\boldsymbol{\mu}\|_2^4 = \Omega(\sigma_p^4 d)$ is the necessary and sufficient condition for benign overfitting of CNNs in learning the XOR-type data. Moreover, by Theorem 3.2 we can see that an increase of $n\|\boldsymbol{\mu}\|_2^4/\sigma_p^4 d$ by a logarithmic factor can sometimes be sufficient to change the test error from a constant level to near optimal rate $p+o(1)$, and this matches the phenomena of sharp phase transition between benign and harmful overfitting demonstrated in (Cao et al., 2022; Kou et al., 2023).

The condition in Theorem 3.2 also matches the conditions discovered by previous works in other learning tasks. Specifically, Cao et al. (2021) studied the risk bounds of linear logistic regression in learning sub-Gaussian mixtures, and the risk upper and lower bounds in Cao et al. (2021) imply exactly the same conditions for small/large test errors. Moreover, Frei et al. (2022) studied the benign overfitting phenomenon in using two-layer fully connected Leaky-ReLU networks to learn sub-Gaussian mixtures, and established an upper bound of the test error that is the same as the upper bound in Theorem 3.2 with $\sigma_p = 1$. More recently, Kou et al. (2023) considered a multi-patch data model similar to the data model considered in this paper, but with the label given by a linear decision rule based on the signal patch, instead of the XOR decision rule considered in this paper. Under similar conditions, the authors also established similar upper and lower bounds of the test error. These previous works share a common nature that the Bayes-optimal classifiers of the learning problems are all linear. On the other hand, this paper studies an XOR-type data and we show in equation 2.1 that all linear models will fail to learn this type of data. Moreover, our results in Theorem 3.2 suggest that two-layer ReLU CNNs can still learn XOR-type data as efficiently as using linear logistic regression or two-layer neural networks to learn sub-Gaussian mixtures.

Recently, another line of works (Wei et al., 2019; Refinetti et al., 2021; Ji & Telgarsky, 2019; Telgarsky, 2023; Barak et al., 2022; Ba et al., 2023; Suzuki et al., 2023) studied a type of XOR problem under more general framework of learning "$k$-sparse parity functions". Specifically, if the label $y \in \{\pm 1\}$ is given by a 2-sparse parity function of the input $\mathbf{x} \in \{\pm 1\}^d$, then $y$ is essentially determined based on an XOR operation. For learning such a function, it has been demonstrated that

kernel methods will need $n = \omega(d^2)$ samples to achieve a small test error (Wei et al., 2019; Telgarsky, 2023). We remark that the data model studied in this paper is different from the XOR problems studied in (Wei et al., 2019; Refinetti et al., 2021; Ji & Telgarsky, 2019; Telgarsky, 2023; Barak et al., 2022; Ba et al., 2023; Suzuki et al., 2023), and therefore the results may not be directly comparable. Moreover, the 2-sparse parity problem has signal strength and noise level roughly equivalent to the setting $\|\boldsymbol{\mu}\|_2 = \sigma_p = \Theta(1)$. According to the conditions for benign and harmful overfitting in Theorem 3.2, $n = \omega(d)$ can lead to a small test error, which is better than the $n = \omega(d^2)$ sample complexity requirement of kernel methods. However, this $n = \omega(d)$ setting is not covered in our analysis due to the over-parameterization requirement $d = \tilde{\Omega}(n^2)$ in Condition 3.1, and thus the comparison is not rigorous. We believe that $d = \tilde{\Omega}(n^2)$ is only a technical condition to enable our current proof, and it may be an interesting future work direction to further weaken this condition.

## 3.2 THE ASYMPTOTICALLY CHALLENGING XOR REGIME

In Section 3.1, we precisely characterized the conditions for benign and harmful overfitting of two-layer CNNs under the "classic XOR regime" where $\cos\theta < 1/2$. Due to certain technical limitations, our analysis in Section 3.1 cannot be directly applied to the case where $\cos\theta \geq 1/2$. In this section, we present another set of results based on an alternative analysis that applies to $\cos\theta \geq 1/2$, and can even handle the case where $1 - \cos\theta = \tilde{\Theta}(1/\sqrt{n})$. However, this alternative analysis relies on several more strict, or different conditions compared to Condition 3.1, which are given below.

**Condition 3.3** *For a certain $\varepsilon > 0$, suppose that*

1. *The dimension $d$ satisfies: $d = \tilde{\Omega}(n^3 m^3 \|\boldsymbol{\mu}\|_2^2 \sigma_p^{-2}) \cdot \mathrm{polylog}(1/\varepsilon)$.*
2. *Training sample size $n$ and neural network width satisfy: $m = \Omega(\log(nd)), n = \Omega(\log(md))$.*
3. *The signal strength satisfies: $\|\boldsymbol{\mu}\|_2(1 - \cos\theta) \geq \tilde{\Omega}(\sigma_p m)$.*
4. *The label flipping probability $p$ satisfies: $p \leq c$ for a small enough absolute constant $c > 0$.*
5. *The learning rate $\eta$ satisfies: $\eta = \tilde{O}([\max\{\sigma_p^2 d^{3/2}/(n^2 m), \sigma_p^2 d/n\}]^{-1})$.*
6. *The angle $\theta$ satisfies: $1 - \cos\theta = \tilde{\Omega}(1/\sqrt{n})$.*

Compared with Condition 3.1, here we require a larger $d$ and also impose an additional condition on the signal strength $\|\boldsymbol{\mu}\|_2$. Moreover, our results are based on a specific choice of the Gaussian random initialization scale $\sigma_0$. The results are given in the following theorem.

**Theorem 3.4** *Consider gradient descent with initialization scale $\sigma_0 = nm/(\sigma_p d) \cdot \mathrm{polylog}(d)$. For any $\varepsilon > 0$, if Condition 3.3 holds, then there exists constant $C > 0$, such that with probability at least $1 - 1/\mathrm{polylog}(d)$, the following results hold at $T = \Omega(nm/(\eta\varepsilon\sigma_p^2 d))$:*

1. *The training loss converges below $\varepsilon$, i.e., $L(\mathbf{W}^{(T)}) \leq \varepsilon$.*
2. *If $\|\boldsymbol{\mu}\|_2^4(1 - \cos\theta)^2 \geq \tilde{\Omega}(m^2\sigma_p^4 d)$, then the CNN trained by gradient descent can achieve near Bayes-optimal test error: $R(\mathbf{W}^{(T)}) \leq p + \exp\{-Cn^2\|\boldsymbol{\mu}\|_2^4(1-\cos\theta)^2/(\sigma_p^6 d^3 \sigma_0^2)\} = p + o(1)$.*
3. *If $\|\boldsymbol{\mu}\|_2^4 \leq \tilde{O}(m\sigma_p^4 d)$, then the CNN trained by gradient descent can only achieve sub-optimal error rate: $R(\mathbf{W}^{(T)}) \geq p + C$.*

Theorem 3.4 also demonstrates the convergence of the training loss towards zero, and establishes upper and lower bounds on the test error achieved by the CNN trained by gradient descent. However, we can see that there is a gap in the conditions for benign and harmful overfitting: benign overfitting is only theoretically guaranteed when $\|\boldsymbol{\mu}\|_2^4(1 - \cos\theta)^2 \geq \tilde{\Omega}(m^2\sigma_p^4 d)$, while we can only rigorously demonstrate harmful overfitting when $\|\boldsymbol{\mu}\|_2^4 \leq \tilde{O}(m\sigma_p^4 d)$. Here the gap between these two conditions is a factor of order $m \cdot (1 - \cos\theta)^{-2}$. Therefore, our results are relatively tight when $m$ is small and when $\cos\theta$ is a constant away from 1 (but not necessarily smaller than $1/2$ which is covered in Section 3.1). Moreover, compared with Theorem 3.2, we see that there lacks a factor of sample size $n$ in both conditions for benign and harmful overfitting. We remark that this is due to our specific choice of $\sigma_0 = nm/(\sigma_p d) \cdot \mathrm{polylog}(d)$, which is in fact out of the range of $\sigma_0$ required in Condition 3.1. Therefore, Theorems 3.2 and 3.4 do not contradict each other. We are not clear whether the overfitting/harmful fitting condition can be unified for all $\theta$, exploring a more unified theory can be left as our future studies.

# 4 OVERVIEW OF PROOF TECHNIQUE

In this section, we briefly discuss our key technical tools in the proofs of Theorem 3.2 and Theorem 3.4. We define $T^* = \eta^{-1}\text{poly}(d, m, n, \varepsilon^{-1})$ as the maximum admissible number of iterations.

**Characterization of signal learning.** Our proof is based on a careful analysis of the training dynamics of the CNN filters. We denote $\mathbf{u} = \mathbf{a} + \mathbf{b}$ and $\mathbf{v} = \mathbf{a} - \mathbf{b}$. Then we have $\boldsymbol{\mu} = \pm\mathbf{u}$ for data with label $+1$, and $\boldsymbol{\mu} = \pm\mathbf{v}$ for data with label $-1$. Note that the ReLU activation function is always non-negative, which implies that the two parts of the CNN in equation 2.2, $F_{+1}(\mathbf{W}_{+1}, \mathbf{x})$ and $F_{-1}(\mathbf{W}_{-1}, \mathbf{x})$, are both always non-negative. Therefore by equation 2.2, we see that $F_{+1}(\mathbf{W}_{+1}, \mathbf{x})$ always contributes towards a prediction of the class $+1$, while $F_{-1}(\mathbf{W}_{-1}, \mathbf{x})$ always contributes towards a prediction of the class $-1$. Therefore, in order to achieve good test accuracy, the CNN filters $\mathbf{w}_{+1,r}$ and $\mathbf{w}_{-1,r}$ $r \in [m]$ must sufficiently capture the directions $\pm\mathbf{u}$ and $\pm\mathbf{v}$ respectively.

In the following, we take a convolution filter $\mathbf{w}_{+1,r}$ for some $r \in [m]$ as an example, to explain our analysis of learning dynamics of the signals $\pm\mathbf{u}$. For each training data input $\mathbf{x}_i$, we denote by $\boldsymbol{\mu}_i \in \{\pm\mathbf{u}, \pm\mathbf{v}\}$ the signal patch in $\mathbf{x}_i$ and by $\boldsymbol{\xi}_i$ the noise patch in $\mathbf{x}_i$. By the gradient descent update of $\mathbf{w}_{+1,r}^{(t)}$, we can easily obtain the following iterative equation of $\langle \mathbf{w}_{+1,r}^{(t)}, \mathbf{u} \rangle$:

$$\langle \mathbf{w}_{+1,r}^{(t+1)}, \mathbf{u} \rangle = \langle \mathbf{w}_{+1,r}^{(t)}, \mathbf{u} \rangle - \frac{\eta}{nm} \sum_{i \in S_{+\mathbf{u},+1} \cup S_{-\mathbf{u},-1}} \ell_i'^{(t)} \cdot \mathbf{1}\{\langle \mathbf{w}_{+1,r}^{(t)}, \boldsymbol{\mu}_i \rangle > 0\}\|\boldsymbol{\mu}\|_2^2$$
$$+ \frac{\eta}{nm} \sum_{i \in S_{-\mathbf{u},+1} \cup S_{+\mathbf{u},-1}} \ell_i'^{(t)} \cdot \mathbf{1}\{\langle \mathbf{w}_{+1,r}^{(t)}, \boldsymbol{\mu}_i \rangle > 0\}\|\boldsymbol{\mu}\|_2^2$$
$$+ \frac{\eta}{nm} \sum_{i \in S_{+\mathbf{v},-1} \cup S_{-\mathbf{v},+1}} \ell_i'^{(t)} \cdot \mathbf{1}\{\langle \mathbf{w}_{+1,r}^{(t)}, \boldsymbol{\mu}_i \rangle > 0\}\|\boldsymbol{\mu}\|_2^2 \cos\theta$$
$$- \frac{\eta}{nm} \sum_{i \in S_{-\mathbf{v},-1} \cup S_{+\mathbf{v},+1}} \ell_i'^{(t)} \cdot \mathbf{1}\{\langle \mathbf{w}_{+1,r}^{(t)}, \boldsymbol{\mu}_i \rangle > 0\}\|\boldsymbol{\mu}\|_2^2 \cos\theta, \quad (4.1)$$

where we denote $S_{\boldsymbol{\mu},y} = \{i \in [n], \boldsymbol{\mu}_i = \boldsymbol{\mu}, y_i = y\}$ for $\boldsymbol{\mu} \in \{\pm\mathbf{u}, \pm\mathbf{v}\}$, $y \in \{\pm 1\}$, and $\ell_i'^{(t)} = \ell'(y_i f(\mathbf{W}^{(t)}, \mathbf{x}_i))$. Based on the above equation, it is clear that when $\cos\theta = 0$, the dynamics of $\langle \mathbf{w}_{+1,r}^{(t)}, \mathbf{u} \rangle$ can be fairly easily characterized. Therefore, a major challenge in our analysis is to handle the additional terms with $\cos\theta$. Intuitively, since $\cos\theta < 1$, we would expect that the first two summation terms in equation 4.1 are the dominating terms so that the learning dynamics of $\pm\mathbf{u}$ will not be significantly different from the case $\cos\theta = 0$. However, a rigorous proof would require a very careful comparison of the loss derivative values $\ell_i'^{(t)}$, $i \in [n]$.

**Loss derivative comparison.** The following lemma is the key lemma we establish to characterize the ratio between loss derivatives of two different data in the "classic XOR regime".

**Lemma 4.1** *Under Condition 3.1, for all $i, k \in [n]$ and all $t \geq 0$, it holds that $\ell_i'^{(t)}/\ell_k'^{(t)} \leq 2 + o(1)$.*

Lemma 4.1 shows that the loss derivatives of any two different data points can always at most have a scale difference of a factor 2. With this result, we can see that in equation 4.1, for any $i, i' \in [n]$, a non-zero $\ell_i'^{(t)} \cdot \mathbf{1}\{\langle \mathbf{w}_{+1,r}^{(t)}, \boldsymbol{\mu}_i \rangle > 0\}\|\boldsymbol{\mu}\|_2^2$ will always dominate $\ell_{i'}'^{(t)} \cdot \mathbf{1}\{\langle \mathbf{w}_{+1,r}^{(t)}, \boldsymbol{\mu}_{i'} \rangle > 0\}\|\boldsymbol{\mu}\|_2^2 \cos\theta$ in the "classic XOR regime" where $\cos\theta < 1/2$. This is the major motivation of Lemma 4.1.

To study the signal learning under the "asymptotically challenging XOR regime" where $\cos\theta$ can be close to 1, Lemma 4.1 is clearly insufficient and tighter bounds on the ratios between loss derivatives are needed. Here we introduce a key technique called "virtual sequence comparison". Specifically, we define "virtual sequences" $\tilde{\ell}_i'^{(t)}$ for $i \in [n]$ and $t \geq 0$, which can be proved to be close to $\ell_i'^{(t)}$, $i \in [n]$, $t \geq 0$. There are two major advantages to studying such virtual sequences: (i) they follow a relatively clean dynamical system and are easy to analyze; (ii) they are independent of the actual weights $\mathbf{W}^{(t)}$ of the CNN and therefore $\tilde{\ell}_i'^{(t)}$, $i \in [n]$ are independent of each other, enabling the application of concentration inequalities. The details are given in the following three lemmas.

**Lemma 4.2** *Let $\boldsymbol{\xi}_i$, $i \in [n]$ be the noise vectors in the training data. Define*

$$\tilde{\ell}_i'^{(t)} = -1/(1 + \exp\{A_i^{(t)}\}), \qquad A_i^{(t+1)} = A_i^{(t)} - \eta/(nm^2) \cdot \tilde{\ell}_i'^{(t)} \cdot |S_i^{(0)}| \cdot \|\boldsymbol{\xi}_i\|_2^2.$$

*Here, $S_i^{(0)} = \{r \in [m] : \langle \mathbf{w}_{y_i,r}^{(0)}, \boldsymbol{\xi}_i \rangle > 0\}$, and $A_i^{(0)} = 0$ for all $i \in [n]$. It holds that*

$$|\tilde{\ell}_i'^{(t)} - \ell_i'^{(t)}| = \widetilde{O}(n/d), \quad \ell_i'^{(t)}/\tilde{\ell}_i'^{(t)}, \quad \tilde{\ell}_i'^{(t)}/\ell_i'^{(t)} \le 1 + \widetilde{O}(n/d),$$

*for all $t \in [T^*]$ and $i \in [n]$.*

Lemma 4.2 gives the small difference among $\tilde{\ell}_i'^{(t)}$ and $\ell_i'^{(t)}$. We give the analysis for $\tilde{\ell}_i'^{(t)}$ in the next lemma, which is much simpler than direct analysis on $\ell_i'^{(t)}$.

**Lemma 4.3** *Let $S_+$ and $S_-$ be the sets of indices satisfying $S_+ \subseteq [n]$, $S_- \subseteq [n]$ and $S_+ \cap S_- = \emptyset$, then with probability at least $1 - 2\delta$, it holds that*

$$\left| \sum_{i \in S_+} \tilde{\ell}_i'^{(t)} / \sum_{i \in S_-} \tilde{\ell}_i'^{(t)} - |S_+|/|S_-| \right| \le 2\mathcal{G}_{\mathrm{gap}}(|S_-|\sqrt{|S_+|} + |S_-|\sqrt{|S_+|})/|S_-|^2,$$

*where $\mathcal{G}_{\mathrm{gap}} = 20\sqrt{\log(2n/\delta)/m} \cdot \sqrt{\log(4/\delta)}$.*

Combining Lemma 4.2 with Lemma 4.3, we can show a precise comparison among multiple $\sum_i \ell_i'^{(t)}$'s. The results are given in the following lemma.

**Lemma 4.4** *Under Condition 3.3, if $S_+ \cap S_- = \emptyset$, and*

$$c_0 n - C\sqrt{n \cdot \log(8n/\delta)} \le |S_+|, |S_-| \le c_1 n + C\sqrt{n \cdot \log(8n/\delta)}$$

*hold for some constant $c_0, c_1, C > 0$, then with probability at least $1 - 2\delta$, it holds that*

$$\left| \sum_{i \in S_+} \ell_i'^{(t)} / (\sum_{i \in S_-} \ell_i'^{(t)}) - c_1/c_0 \right| \le 4c_1 C/(c_0^2) \cdot \sqrt{\log(8n/\delta)/n}.$$

By Lemma 4.4, we have a precise bound for $\sum_{i \in S_+} \ell_i'^{(t)} / \sum_{i \in S_-} \ell_i'^{(t)}$. The results can then be plugged into equation 4.1 to simplify the dynamical systems of $\langle \mathbf{w}_{+1,r}^{(t)}, \mathbf{u} \rangle$, $r \in [m]$, which characterizes the signal learning process during training.

Our analysis of the signal learning process is then combined with the analysis of how much training data noises $\boldsymbol{\xi}_i$ have been memorized by CNN filters, and then the training loss and test error can both be bounded based on their definitions. We defer the detailed analyses to the appendix.

## 5 EXPERIMENTS

In this section, we present simulation results on synthetic data to back up our theoretical analysis. Our experiments cover different choices of the training sample size $n$, the dimension $d$, and the signal strength $\|\boldsymbol{\mu}\|_2$. In all experiments, the test error is calculated based on 1000 i.i.d. test data.

Given dimension $d$ and signal strength $\|\boldsymbol{\mu}\|_2$, we generate XOR-type data according to Definition 2.2. We consider a challenging case where the vectors $\mathbf{a} + \mathbf{b}$ and $\mathbf{a} - \mathbf{b}$ has an angle $\theta$ with $\cos\theta = 0.8$. To determine $\mathbf{a}$ and $\mathbf{b}$, we uniformly select the directions of the orthogonal basis vectors $\mathbf{a}$ and $\mathbf{b}$, and determine their norms by solving the two equations

$$\|\mathbf{a}\|_2^2 + \|\mathbf{b}\|_2^2 = \|\boldsymbol{\mu}\|_2^2, \quad \|\mathbf{a}\|_2^2 - \|\mathbf{b}\|_2^2 = \|\boldsymbol{\mu}\|_2^2 \cdot \cos\theta.$$

The signal patch $\boldsymbol{\mu}$ and the clean label $\overline{y}$ are then jointly generated from $\mathcal{D}_{\mathrm{XOR}}(\mathbf{a}, \mathbf{b})$ in Definition 2.1. Moreover, the noise patch $\boldsymbol{\xi}$ is generated with standard deviation $\sigma_p = 1$, and the observed label $y$ is given by flipping $\overline{y}$ with probability $p = 0.1$. We consider a CNN model following the exact definition as in equation 2.2, where we set $m = 40$. To train the CNN, we use full-batch gradient descent starting with entry-wise Gaussian random initialization $N(0, \sigma_0^2)$ and set $\sigma_0 = 0.01$. We set the learning rate as $10^{-3}$ and run gradient descent for $T = 200$ training epochs.

Our goal is to verify the upper and lower bounds of the test error by plotting heatmaps of test errors under different sample sizes $n$, dimensions $d$, and signal strengths $\|\boldsymbol{\mu}\|_2$. We consider two settings:

1. In the first setting, we fix $d = 200$ and report the test accuracy for different choices of $n$ and $\|\boldsymbol{\mu}\|_2$. According to Theorem 3.2, we see that the phase transition between benign and harmful overfitting happens around the critical point where $n\|\boldsymbol{\mu}\|_2^4/(\sigma_p^4 d) = \Theta(1)$. Therefore, in the test accuracy heat map, we use the vertical axis to denote $n$ and the horizontal axis to denote the value of $\sigma_p^4 d/\|\boldsymbol{\mu}\|_2^4$. We report the test accuracy for $n$ ranging from 4 to 598, and the range of $\|\boldsymbol{\mu}\|_2$ is determined to make the value of $\sigma_p^4 d/\|\boldsymbol{\mu}\|_2^4$ be ranged from 0.1 to $10^1$. We also report two

---

[1] These ranges are selected to set up an appropriate range to showcase the change of the test accuracy.

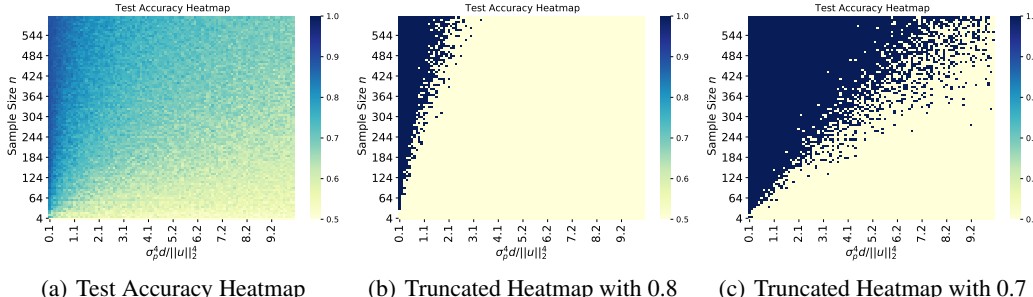

(a) Test Accuracy Heatmap  (b) Truncated Heatmap with 0.8  (c) Truncated Heatmap with 0.7

Figure 1: Heatmap of test accuracy under different values of $n$ and $\sigma_p^4 d/\|\boldsymbol{\mu}\|_2^4$. The $x$-axis represents the value of $\sigma_p^4 d/\|\boldsymbol{\mu}\|_2^4$, whereas the $y$-axis is sample size $n$. (a) displays the original heatmap of test accuracy, where high accuracy is colored blue and low accuracy is colored yellow. (b) and (c) show the truncated heatmap of test accuracy, where accuracy higher than $0.8$ or $0.7$ is colored blue, and accuracy lower than $0.8$ or $0.7$ is colored yellow.

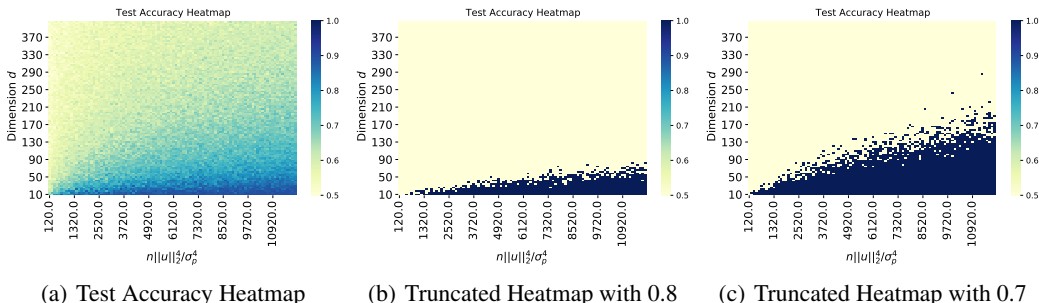

(a) Test Accuracy Heatmap  (b) Truncated Heatmap with 0.8  (c) Truncated Heatmap with 0.7

Figure 2: Heatmap of test accuracy under different values of $d$ and $n\|\boldsymbol{\mu}\|_2^4/\sigma_p^4$. The $x$-axis represents the value of $n\|\boldsymbol{\mu}\|_2^4/\sigma_p^4$, and $y$-axis represents dimension $d$. (a) displays the original heatmap of test accuracy, where high accuracy is represented by blue color and low accuracy is represented by yellow. (b) and (c) show the truncated heatmap of test accuracy, where accuracy higher than $0.8$ or $0.7$ is colored blue, and accuracy lower than $0.8$ or $0.7$ is colored yellow.

truncated heatmaps converting the test accuracy to binary values based on truncation thresholds $0.8$ and $0.7$, respectively. The results are given in Figure 1.

2. In the second setting, we fix $n = 80$ and report the test accuracy for different choices of $d$ and $\|\boldsymbol{\mu}\|_2$. Here, we use the vertical axis to denote $d$ and use the horizontal axis to denote the value of $n\|\boldsymbol{\mu}\|_2^4/\sigma_p^4$. In this heatmap, we set the range of $d$ to be from 10 to 406, and the range of $\|\|\boldsymbol{\mu}\|_2\|_2$ is chosen so that $n\|\boldsymbol{\mu}\|_2^4/\sigma_p^4$ ranges from 120 to 12000. Again, we also report two truncated heatmaps converting the test accuracy to binary values based on truncation thresholds $0.8$ and $0.7$, respectively. The results are given in Figure 2.

As shown in Figure 1 and Figure 2, it is evident that an increase in the training sample size $n$ or the signal length $\|\boldsymbol{\mu}\|_2$ can lead to an increase in the test accuracy. On the other hand, increasing the dimension $d$ results in a decrease in the test accuracy. These results are clearly intuitive and also match our theoretical results. Furthermore, we can see from the heatmaps in Figures 1 and 2 that the contours of the test accuracy are straight lines in the spaces $(\sigma_p^4 d/\|\boldsymbol{\mu}\|_2^4, n)$ and $(n\|\boldsymbol{\mu}\|_2^4/\sigma_p^4, d)$, and this observation is more clearly demonstrated with the truncated heatmaps. Therefore, we believe our experiment results here can provide strong support for our theory.

## 6  CONCLUSIONS

This paper focuses on studying benign overfitting in two-layer ReLU CNNs for XOR-type data. Our results reveal a sharp phase transition between benign and harmful overfitting, and demonstrate that CNNs have remarkable capacities to efficiently learn XOR problems even in the presence of highly correlated features. There are several interesting directions for future investigation. It is important to generalize our analysis to deep ReLU neural networks, and it is also interesting to study benign overfitting of neural networks in learning unbalanced data.

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

CONTENTS

## A   MORE DETAILS IN THE DISCUSSION OF PROOF

With Lemma 4.1, it is clear for us to see the main leading term when $\cos\theta < 1/2$. For instance, when $\langle \mathbf{w}_{+1,r}, \mathbf{u}\rangle > 0$ and $\cos\theta < 1/2$, it is easy to see that $-\frac{\eta}{nm}\sum_{i\in S_{+\mathbf{u},+1}}\ell_i'^{(t)}\cdot\mathbf{1}\{\langle\mathbf{w}_{+1,r}^{(t)},\boldsymbol{\mu}_i\rangle > 0\}\|\boldsymbol{\mu}\|_2^2$ is the main leading term. With the leading term, we can thus characterize the growth of $\langle\mathbf{w}_{+1,r}^{(t)},\boldsymbol{\mu}_i\rangle$. We have the following proposition.

**Proposition A.1** *Under Condition 3.1, the following points hold:*

*1. For any $r \in [m]$, the inner product between $\mathbf{u}$ and $\mathbf{w}_{-1,r}^{(t)}$ satisfies*

$$|\langle\mathbf{w}_{-1,r}^{(t)}, \mathbf{u}\rangle| \leq 2\sqrt{\log(12m/\delta)}\cdot\sigma_0\|\boldsymbol{\mu}\|_2 + \eta\|\boldsymbol{\mu}\|_2^2/m.$$

*2. For any $r \in [m]$, $\langle\mathbf{w}_{+1,r}^{(t)}, \mathbf{u}\rangle$ increases if $\langle\mathbf{w}_{+1,r}^{(0)}, \mathbf{u}\rangle > 0$, $\langle\mathbf{w}_{+1,r}^{(t)}, \mathbf{u}\rangle$ decreases if $\langle\mathbf{w}_{+1,r}^{(0)}, \mathbf{u}\rangle < 0$. Moreover, it holds that*

$$|\langle\mathbf{w}_{+1,r}^{(t)}, \mathbf{u}\rangle| \geq -2\sqrt{\log(12m/\delta)}\cdot\sigma_0\|\boldsymbol{\mu}\|_2 + cn\|\boldsymbol{\mu}\|_2^2/(\sigma_p^2 d)\cdot\log\left(\eta\sigma_p^2 d(t-1)/(12nm) + 2/3\right) - \eta\cdot\|\boldsymbol{\mu}\|_2^2/m.$$

*3. For $t = \Omega(nm/(\eta\sigma_p^2 d))$, the bound for $\left\|\mathbf{w}_{j,r}^{(t)}\right\|_2$ is given by:*

$$\Theta\left(\sigma_p^{-1}d^{-1/2}n^{1/2}\right)\cdot\log\left(\eta\sigma_p^2 d(t-1)/(12nm) + 2/3\right) \leq \left\|\mathbf{w}_{j,r}^{(t)}\right\|_2 \leq \Theta\left(\sigma_p^{-1}d^{-1/2}n^{1/2}\right)\cdot\log\left(2\eta\sigma_p^2 dt/(nm) + 1\right).$$

We are prepared to investigate the test error. Consider the test data $(\mathbf{x}, y) = ([\mathbf{u}, \boldsymbol{\xi}], +1)$ as an example. To classify this data correctly, it must have a high probability of $f(\mathbf{W}^{(t)}, \mathbf{x}) > 0$, where $f(\mathbf{W}^{(t)}, \mathbf{x}) = F_{+1}(\mathbf{W}_{+1}, \mathbf{x}) - F_{-1}(\mathbf{W}_{-1}, \mathbf{x})$. Both $F_{+1}(\mathbf{W}_{+1}, \mathbf{x})$ and $F_{-1}(\mathbf{W}_{-1}, \mathbf{x})$ are always non-negative, so it is necessary for $F_{+1}(\mathbf{W}_{+1}, \mathbf{x}) > F_{-1}(\mathbf{W}_{-1}, \mathbf{x})$ with high probability. The scale of $F_{+1}(\mathbf{W}_{+1}, \mathbf{x})$ is determined by $\langle\mathbf{w}_{+1,r}^{(t)}, \mathbf{u}\rangle$ and $\langle\mathbf{w}_{+1,r}^{(t)}, \boldsymbol{\xi}\rangle$, while the scale of $F_{-1}(\mathbf{W}_{-1}, \mathbf{x})$ is determined by $\langle\mathbf{w}_{-1,r}^{(t)}, \mathbf{u}\rangle$ and $\langle\mathbf{w}_{-1,r}^{(t)}, \boldsymbol{\xi}\rangle$. Note that the scale $\langle\mathbf{w}_{j,r}^{(t)}, \boldsymbol{\xi}\rangle$ is determined by $\|\mathbf{w}_{j,r}^{(t)}\|_2$. Therefore, to determine the test accuracy, we investigated $\langle\mathbf{w}_{+1,r}^{(t)}, \mathbf{u}\rangle$, $\langle\mathbf{w}_{-1,r}^{(t)}, \mathbf{u}\rangle$, and $\|\mathbf{w}_{j,r}^{(t)}\|_2$, as shown in Proposition A.1 above. Note that the bound in conclusion 1 in Proposition A.1 also hold for $\langle\mathbf{w}_{+1,r}^{(t)}, \mathbf{v}\rangle$, and the bound in conclusion 2 of Proposition A.1 still holds for $\langle\mathbf{w}_{-1,r}^{(t)}, \mathbf{v}\rangle$. Based on Proposition A.1, we can apply a standard technique which is also employed in Chatterji & Long (2021); Frei et al. (2022); Kou et al. (2023) to establish the proof of Theorem 3.2. A detailed proof can be found in the following sections.

## B   NOISE DECOMPOSITION AND ITERATION EXPRESSION

In this section, we give the analysis of the update rule for the noise decomposition. We provide an analysis of the noise decomposition. The gradient $\frac{\partial L(\mathbf{W})}{\partial\mathbf{w}_{j,r}}$ can be expressed as

$$\frac{\partial L(\mathbf{W})}{\partial\mathbf{w}_{j,r}} = \frac{1}{n}\sum_{i=1}^{n}\ell_i' y_i\frac{\partial f(\mathbf{W}, \mathbf{x}_i)}{\partial\mathbf{w}_{j,r}} = \frac{1}{nm}\sum_{i=1}^{n}\ell_i'\cdot(jy_i)\sum_{r=1}^{m}\sum_{p=1}^{2}\mathbf{1}\{\langle\mathbf{w}_{j,r}, \mathbf{x}_i^{(p)}\rangle > 0\}\mathbf{x}_i^{(p)}.$$

For each training sample point $\mathbf{x}_i$, we define $\boldsymbol{\mu}_i \in \{\pm\mathbf{u}, \pm\mathbf{v}\}$ represents the signal part in $\mathbf{x}_i$ and $\boldsymbol{\xi}_i$ represents the noise in $\mathbf{x}_i$. The updated rule of $\mathbf{w}_{j,r}^{(t)}$ hence can be expressed as

$$\mathbf{w}_{j,r}^{(t+1)} = \mathbf{w}_{j,r}^{(t)} - \frac{\eta j}{nm}\sum_{i\in S_{+\mathbf{u},+1}\cup S_{-\mathbf{u},-1}}\ell_i'^{(t)}\mathbf{1}\{\langle\mathbf{w}_{j,r}^{(t)},\boldsymbol{\mu}_i\rangle > 0\}\mathbf{u} + \frac{\eta j}{nm}\sum_{i\in S_{-\mathbf{u},+1}\cup S_{+\mathbf{u},-1}}\ell_i'^{(t)}\mathbf{1}\{\langle\mathbf{w}_{j,r}^{(t)},\boldsymbol{\mu}_i\rangle > 0\}\mathbf{u}$$

$$+ \frac{\eta j}{nm}\sum_{i\in S_{+\mathbf{v},-1}\cup S_{-\mathbf{v},+1}}\ell_i'^{(t)}\mathbf{1}\{\langle\mathbf{w}_{j,r}^{(t)},\boldsymbol{\mu}_i\rangle > 0\}\mathbf{v} - \frac{\eta j}{nm}\sum_{i\in S_{-\mathbf{v},-1}\cup S_{+\mathbf{v},+1}}\ell_i'^{(t)}\mathbf{1}\{\langle\mathbf{w}_{j,r}^{(t)},\boldsymbol{\mu}_i\rangle > 0\}\mathbf{v}$$

$$- \frac{\eta}{nm}\sum_{i=1}^{n}\ell_i'^{(t)}(jy_i)\mathbf{1}\{\langle\mathbf{w}_{j,r}^{(t)},\boldsymbol{\xi}_i\rangle > 0\}\boldsymbol{\xi}_i,$$

$$\text{(B.1)}$$

where $S_{\boldsymbol{\mu},y} = \{i \in [n], \boldsymbol{\mu}_i = \boldsymbol{\mu}, y_i = y\}$. Here, $\boldsymbol{\mu} \in \{\pm\mathbf{u}, \pm\mathbf{v}\}$, $y \in \{\pm1\}$. For instance, $S_{+\mathbf{u},-1} = \{i \in [n], \boldsymbol{\mu}_i = \mathbf{u}, y_i = -1\}$.

The lemma below presents an iterative expression for the coefficients.

**Lemma B.1** *Suppose that the update rule of $\mathbf{w}_{j,r}^{(t)}$ follows equation B.1, then $\langle \mathbf{w}_{j,r}^{(t)}, \boldsymbol{\xi}_i \rangle$ can be decomposed into*

$$\langle \mathbf{w}_{j,r}^{(t)}, \boldsymbol{\xi}_i \rangle = \langle \mathbf{w}_{j,r}^{(0)}, \boldsymbol{\xi}_i \rangle + \sum_{i'=1}^{n} \rho_{j,r,i'}^{(t)} \langle \boldsymbol{\xi}_{i'}, \boldsymbol{\xi}_i \rangle / \|\boldsymbol{\xi}_{i'}\|_2^2.$$

*Further denote $\overline{\rho}_{j,r,i}^{(t)} := \rho_{j,r,i}^{(t)} \mathbf{1}(\rho_{j,r,i}^{(t)} \geq 0), \underline{\rho}_{j,r,i}^{(t)} := \rho_{j,r,i}^{(t)} \mathbf{1}(\rho_{j,r,i}^{(t)} \leq 0)$. Then*

$$\langle \mathbf{w}_{j,r}^{(t)}, \boldsymbol{\xi}_i \rangle = \langle \mathbf{w}_{j,r}^{(0)}, \boldsymbol{\xi}_i \rangle + \sum_{i'=1}^{n} \overline{\rho}_{j,r,i'}^{(t)} \langle \boldsymbol{\xi}_{i'}, \boldsymbol{\xi}_i \rangle / \|\boldsymbol{\xi}_{i'}\|_2^2 + \sum_{i'=1}^{n} \underline{\rho}_{j,r,i'}^{(t)} \langle \boldsymbol{\xi}_{i'}, \boldsymbol{\xi}_i \rangle / \|\boldsymbol{\xi}_{i'}\|_2^2.$$

*Here, $\overline{\rho}_{j,r,i}^{(t)}$ and $\underline{\rho}_{j,r,i}^{(t)}$ can be defined by the following iteration:*

$$\overline{\rho}_{j,r,i}^{(t+1)} = \overline{\rho}_{j,r,i}^{(t)} - \frac{\eta}{nm} \cdot \ell_i^{(t)} \cdot \sigma'(\langle \mathbf{w}_{j,r}^{(t)}, \boldsymbol{\xi}_i \rangle) \cdot \|\boldsymbol{\xi}_i\|_2^2 \cdot \mathbf{1}(y_i = j),$$

$$\underline{\rho}_{j,r,i}^{(t+1)} = \underline{\rho}_{j,r,i}^{(t)} + \frac{\eta}{nm} \cdot \ell_i^{(t)} \cdot \sigma'(\langle \mathbf{w}_{j,r}^{(t)}, \boldsymbol{\xi}_i \rangle) \cdot \|\boldsymbol{\xi}_i\|_2^2 \cdot \mathbf{1}(y_i = -j), \qquad \overline{\rho}_{j,r,i}^{(0)} = \underline{\rho}_{j,r,i}^{(0)} = 0.$$

**Proof** [Proof of Lemma B.1] Note that $\mathbf{w}_{j,r}^{(t)} \in \text{span}\{\mathbf{u}, \mathbf{v}, \{\boldsymbol{\xi}_i\}_{i=1}^{n}\} + \mathbf{w}_{j,r}^{(0)}$, and $\text{span}\{\mathbf{u}, \mathbf{v}\} \perp \text{span}\{\{\boldsymbol{\xi}_i\}_{i=1}^{n}\}$. We can express $\mathbf{w}_{j,r}^{(t)}$ by

$$\mathbf{w}_{j,r}^{(t+1)} = \mathbf{w}_{j,r}^{(0)} - \frac{\eta}{nm} \sum_{s=0}^{t} \sum_{i=1}^{n} \ell_i'^{(s)} \cdot \sigma'(\langle \mathbf{w}_{j,r}^{(s)}, \boldsymbol{\xi}_i \rangle) \cdot jy_i \boldsymbol{\xi}_i - \mathbf{a}^{(t)},$$

where $\mathbf{a}^{(t)} \in \text{span}\{\mathbf{u}, \mathbf{v}\}$. By $\{\boldsymbol{\xi}_i\}_{i=1}^{n}$ and $\mathbf{u}, \mathbf{v}$ are linearly independent with probability 1, we have that $\rho_{j,r,i}^{(t)}$ has the unique expression

$$\rho_{j,r,i}^{(t)} = -\frac{\eta}{nm} \sum_{s=0}^{t} \ell_i'^{(s)} \cdot \sigma'(\langle \mathbf{w}_{j,r}^{(s)}, \boldsymbol{\xi}_i \rangle) \cdot \|\boldsymbol{\xi}_i\|_2^2 \cdot jy_i.$$

Now with the notation $\overline{\rho}_{j,r,i}^{(t)} := \rho_{j,r,i}^{(t)} \mathbf{1}(\rho_{j,r,i}^{(t)} \geq 0), \underline{\rho}_{j,r,i}^{(t)} := \rho_{j,r,i}^{(t)} \mathbf{1}(\rho_{j,r,i}^{(t)} \leq 0)$ and the fact $\ell_i'^{(s)} < 0$, we get

$$\overline{\rho}_{j,r,i}^{(t)} = -\frac{\eta}{nm} \sum_{s=0}^{t} \ell_i'^{(s)} \cdot \sigma'(\langle \mathbf{w}_{j,r}^{(s)}, \boldsymbol{\xi}_i \rangle) \cdot \|\boldsymbol{\xi}_i\|_2^2 \cdot \mathbf{1}(y_i = j),$$

$$\underline{\rho}_{j,r,i}^{(t)} = \frac{\eta}{nm} \sum_{s=0}^{t} \ell_i'^{(s)} \cdot \sigma'(\langle \mathbf{w}_{j,r}^{(s)}, \boldsymbol{\xi}_i \rangle) \cdot \|\boldsymbol{\xi}_i\|_2^2 \cdot \mathbf{1}(y_i = -j).$$

This completes the proof. ∎

## C CONCENTRATION

In this section, we present some fundamental lemmas that illustrate important properties of data and neural network parameters at their random initialization.

The next two lemmas provide concentration bounds on the number of candidates in specific sets. The proof is similar to that presented in Cao et al. (2022) and Kou et al. (2023). For the convenience of readers, we provide the proof here.

**Lemma C.1** *Suppose that $\delta > 0$. Then with probability at least $1 - \delta$, for any $i \in [n]$ it holds that*

$$\left| |S_i^{(0)}| - m/2 \right| \leq \sqrt{\frac{m \log(2n/\delta)}{2}}.$$

**Proof** [Proof of Lemma C.1] Note that $\left|S_i^{(0)}\right| = \sum_{r=1}^m \mathbf{1}\left[\left\langle \mathbf{w}_{y_i,r}^{(0)}, \boldsymbol{\xi}_i \right\rangle > 0\right]$ and $P\left(\left\langle \mathbf{w}_{y_i}^{(0)}, \boldsymbol{\xi}_i \right\rangle > 0\right) = 1/2$, then by Hoeffding's inequality, with probability at least $1 - \delta/n$, we have

$$\left|\frac{\left|S_i^{(0)}\right|}{m} - \frac{1}{2}\right| \leq \sqrt{\frac{\log(2n/\delta)}{2m}},$$

which completes the proof. ■

**Lemma C.2** *Suppose that $\delta > 0$. Then with probability at least $1 - \delta$,*

$$\left|\left|S_{\boldsymbol{\mu}_i, y_i}\right| - n(1-p)/4\right| \leq \sqrt{\frac{n\log(8n/\delta)}{2}}$$

*for all $i$ satisfies $\boldsymbol{\mu}_i = \pm\mathbf{u}$, $y_i = +1$ and $\boldsymbol{\mu}_i = \pm\mathbf{v}$, $y_i = -1$;*

$$\left|\left|S_{\boldsymbol{\mu}_i, y_i}\right| - np/4\right| \leq \sqrt{\frac{n\log(8n/\delta)}{2}}$$

*for all $i$ satisfies $\boldsymbol{\mu}_i = \pm\mathbf{u}$, $y_i = -1$ and $\boldsymbol{\mu}_i = \pm\mathbf{v}$, $y_i = +1$.*

**Proof** [Proof of Lemma C.2] The proof is quitely similar to the Proof of Lemma C.1, and we thus omit it. ■

The following lemma provides us with the bounds of noise norm and their inner product. The proof is similar to that presented in Cao et al. (2022) and Kou et al. (2023). For the convenience of readers, we provide the proof here.

**Lemma C.3** *Suppose that $\delta > 0$ and $d = \Omega(\log(4n/\delta))$. Then with probability at least $1 - \delta$,*

$$\sigma_p^2 d/2 \leq \sigma_p^2 d - C_0\sigma_p^2 \cdot \sqrt{d\log(4n/\delta)} \leq \left\|\boldsymbol{\xi}_i\right\|_2^2 \leq \sigma_p^2 d + C_0\sigma_p^2 \cdot \sqrt{d\log(4n/\delta)} \leq 3\sigma_p^2 d/2,$$
$$\left|\left\langle \boldsymbol{\xi}_i, \boldsymbol{\xi}_{i'} \right\rangle\right| \leq 2\sigma_p^2 \cdot \sqrt{d\log(4n^2/\delta)}$$

*for all $i, i' \in [n]$. Here $C_0 > 0$ is some absolute value.*

**Proof** By Bernstein's inequality, with probability at least $1 - \delta/(2n)$ we have

$$\left|\left\|\boldsymbol{\xi}_i\right\|_2^2 - \sigma_p^2 d\right| \leq C_0\sigma_p^2 \cdot \sqrt{d\log(4n/\delta)}.$$

Therefore, if we set appropriately $d = \Omega(\log(4n/\delta))$, we get

$$\sigma_p^2 d/2 \leq \left\|\boldsymbol{\xi}_i\right\|_2^2 \leq 3\sigma_p^2 d/2.$$

Moreover, clearly $\left\langle \boldsymbol{\xi}_i, \boldsymbol{\xi}_{i'} \right\rangle$ has mean zero. For any $i, i'$ with $i \neq i'$, by Bernstein's inequality, with probability at least $1 - \delta/(2n^2)$ we have

$$\left|\left\langle \boldsymbol{\xi}_i, \boldsymbol{\xi}_{i'} \right\rangle\right| \leq 2\sigma_p^2 \cdot \sqrt{d\log(4n^2/\delta)}.$$

Applying a union bound completes the proof. ■

The following lemma provides a bound on the norm of the randomly initialized CNN filter $\mathbf{w}_{j,r}^{(0)}$, as well as the inner product between $\mathbf{w}_{j,r}^{(0)}$ and $\mathbf{u}$, $\mathbf{v}$, and $\boldsymbol{\xi}_i$. The proof is similar to that presented in Cao et al. (2022) and Kou et al. (2023). For the convenience of readers, we provide the proof here.

**Lemma C.4** *Suppose that $d = \Omega(\log(mn/\delta))$, $m = \Omega(\log(1/\delta))$. Let $\boldsymbol{\mu} = \mathbf{u}$ or $\boldsymbol{\mu} = \mathbf{v}$, then with probability at least $1 - \delta$,*

$$\sigma_0^2 d/2 \leq \|\mathbf{w}_{j,r}^{(0)}\|_2^2 \leq 3\sigma_0^2 d/2,$$

$$|\langle \mathbf{w}_{j,r}^{(0)}, \boldsymbol{\mu} \rangle| \leq \sqrt{2\log(12m/\delta)} \cdot \sigma_0 \|\boldsymbol{\mu}\|_2,$$

$$|\langle \mathbf{w}_{j,r}^{(0)}, \boldsymbol{\xi}_i \rangle| \leq 2\sqrt{\log(12mn/\delta)} \cdot \sigma_0 \sigma_p \sqrt{d}$$

*for all $r \in [m], j \in \{\pm 1\}$ and $i \in [n]$. Moreover,*

$$\min_r |\langle \mathbf{w}_{j,r}^{(0)}, \boldsymbol{\xi}_i \rangle| \geq \sigma_0 \sigma_p \sqrt{d}\delta/8m$$

*for all $j \in \{\pm 1\}$ and $i \in [n]$.*

**Proof** [Proof of Lemma C.4] First of all, the initial weights $\mathbf{w}_{j,r}^{(0)} \sim \mathcal{N}(\mathbf{0}, \sigma_0 \mathbf{I})$. By Bernstein's inequality, with probability at least $1 - \delta/(6m)$ we have

$$|\|\mathbf{w}_{j,r}^{(0)}\|_2^2 - \sigma_0^2 d| \leq C\sigma_0^2 \cdot \sqrt{d\log(12m/\delta)}$$

for some absolute value $C > 0$. Therefore, if we set appropriately $d = \Omega(\log(mn/\delta))$, we have with probability at least $1 - \delta/3$, for all $j \in \{\pm 1\}$ and $r \in [m]$,

$$\sigma_0^2 d/2 \leq \|\mathbf{w}_{j,r}^{(0)}\|_2^2 \leq 3\sigma_0^2 d/2.$$

Next, it is clear that for each $r \in [m], j \cdot \langle \mathbf{w}_{j,r}^{(0)}, \boldsymbol{\mu} \rangle$ is a Gaussian random variable with mean zero and variance $\sigma_0^2 \|\boldsymbol{\mu}\|_2^2$. Therefore, by Gaussian tail bound and union bound, with probability at least $1 - \delta/6$, for all $j \in \{\pm 1\}$ and $r \in [m]$,

$$|\langle \mathbf{w}_{j,r}^{(0)}, \boldsymbol{\mu} \rangle| \leq \sqrt{2\log(12m/\delta)} \cdot \sigma_0 \|\boldsymbol{\mu}\|_2.$$

Similarly we get the bound of $|\langle \mathbf{w}_{j,r}^{(0)}, \boldsymbol{\xi}_i \rangle|$. To prove the last inequality, by the definition of normal distribution we have

$$P\big(|\langle \mathbf{w}_{j,r}^{(0)}, \boldsymbol{\xi_i} \rangle| \geq \sigma_0 \sigma_p \sqrt{d}\delta/8m\big) \geq 1 - \frac{\delta}{2m},$$

therefore we have

$$P\big(\min_r |\langle \mathbf{w}_{j,r}^{(0)}, \boldsymbol{\xi_i} \rangle| \geq \sigma_0 \sigma_p \sqrt{d}\delta/8m\big) \geq \left(1 - \frac{\delta}{2m}\right)^m \geq 1 - \delta.$$

This completes the proof. ■

## D COEFFICIENT SCALE ANALYSIS

In this section, we provide an analysis of the coefficient scale during the whole training procedure, and show in detail to provide a bound on the loss derivatives. All the results presented here are based on the conclusions derived in Appendix C. We want to emphasize that all the results in this section, as well as in the subsequent sections, are conditional on the event $\mathcal{E}$, which refers to the event that all the results in Appendix C hold.

### D.1 PRELIMINARY LEMMAS

In this section, we present a lemma, which provides a valuable insight into the behavior of discrete processes and their continuous counterparts.

**Lemma D.1** *Suppose that a sequence $a_t, t \geq 0$ follows the iterative formula*

$$a_{t+1} = a_t + \frac{c}{1 + be^{a_t}},$$

*for some $1 \geq c \geq 0$ and $b \geq 0$. Then it holds that*

$$x_t \leq a_t \leq \frac{c}{1 + be^{a_0}} + x_t$$

*for all $t \geq 0$. Here, $x_t$ is the unique solution of*

$$x_t + be^{x_t} = ct + a_0 + be^{a_0}.$$

**Proof** [Proof of Lemma D.1] Consider a continuous-time sequence $x_t$, $t \geq 0$ defined by the integral equation

$$x_t = x_0 + c \int_0^t \frac{d\tau}{1 + be^{x_\tau}}, \quad x_0 = a_0. \tag{D.1}$$

Obviously, $x_t$ is an increasing function of $t$, and $x_t$ satisfies

$$\frac{dx_t}{dt} = \frac{c}{1 + be^{x_t}}, \quad x_0 = a_0.$$

By solving this equation, we have

$$x_t + be^{x_t} = ct + a_0 + be^{a_0}.$$

It is obviously that the equation above has unique solution. We first show the lower bound of $a_t$. By equation D.1, we have

$$x_{t+1} = x_t + c \int_t^{t+1} \frac{d\tau}{1 + be^{x_\tau}}$$

$$\leq x_t + c \int_t^{t+1} \frac{d\tau}{1 + be^{x_t}} = x_t + \frac{c}{1 + be^{x_t}}.$$

Note that $c \leq 1$, $x + c/(1 + be^x)$ is an increasing function. By comparison theorem we have $a_t \geq x_t$. For the other side, we have

$$a_t = a_0 + \sum_{\tau=0}^{t} \frac{c}{1 + be^{a_\tau}}$$

$$\leq a_0 + \sum_{\tau=0}^{t} \frac{c}{1 + be^{x_\tau}}$$

$$= a_0 + \frac{c}{1 + be^{a_0}} + \sum_{\tau=1}^{t} \frac{c}{1 + be^{x_\tau}}$$

$$\leq a_0 + \frac{c}{1 + be^{a_0}} + c \int_0^t \frac{d\tau}{1 + be^{x_\tau}}$$

$$= a_0 + \frac{c}{1 + be^{a_0}} + \int_0^t dx_\tau = a_0 + \frac{c}{1 + be^{a_0}} + x_t - x_0$$

$$= \frac{c}{1 + be^{a_0}} + x_t.$$

Here, the first inequality is by $a_t \geq x_t$, the second inequality is by the definition of integration, the third equality is by $\frac{dx_t}{dt} = \frac{c}{1+be^{x_t}}$. We thus complete the proof. ∎

## D.2 SCALE IN COEFFICIENTS

In this section, we start our analysis of the scale in the coefficients during the training procedure.

We give the following proposition. Here, we remind the readers that $\text{SNR} = \|\boldsymbol{\mu}\|_2/(\sigma_p\sqrt{d})$.

**Proposition D.2** *For any $0 \leq t \leq T^*$, it holds that*

$$0 \leq |\langle \mathbf{w}_{j,r}^{(t)}, \mathbf{u} \rangle|, |\langle \mathbf{w}_{j,r}^{(t)}, \mathbf{v} \rangle| \leq 8n \cdot \text{SNR}^2 \log(T^*), \tag{D.2}$$

$$0 \leq \overline{\rho}_{j,r,i}^{(t)} \leq 4\log(T^*), \quad 0 \geq \underline{\rho}_{j,r,i}^{(t)} \geq -2\sqrt{\log(12mn/\delta)} \cdot \sigma_0\sigma_p\sqrt{d} - 32\sqrt{\frac{\log(4n^2/\delta)}{d}} n \log(T^*) \tag{D.3}$$

*for all $j \in \{\pm 1\}$, $r \in [m]$ and $i \in [n]$.*

We use induction to prove Proposition D.2. We introduce several technical lemmas which are applied into the inductive proof of Proposition D.2.

**Lemma D.3** *Under Condition 3.1, suppose equation D.2 and equation D.3 hold at iteration $t$. Then, for all $r \in [m], j \in \{\pm 1\}$ and $i \in [n]$, it holds that*

$$\big|\langle \mathbf{w}_{j,r}^{(t)} - \mathbf{w}_{j,r}^{(0)}, \boldsymbol{\xi}_i \rangle - \underline{\rho}_{j,r,i}^{(t)}\big| \le 16\sqrt{\frac{\log(4n^2/\delta)}{d}} n \log(T^*), \quad j \ne y_i,$$

$$\big|\langle \mathbf{w}_{j,r}^{(t)} - \mathbf{w}_{j,r}^{(0)}, \boldsymbol{\xi}_i \rangle - \overline{\rho}_{j,r,i}^{(t)}\big| \le 16\sqrt{\frac{\log(4n^2/\delta)}{d}} n \log(T^*), \quad j = y_i.$$

**Proof** [Proof of Lemma D.3] For $j \ne y_i$ and any $t \ge 0$, we have $\overline{\rho}_{j,r,i'}^{(t)} = 0$ for all $i' \in [n]$, and so

$$\langle \mathbf{w}_{j,r}^{(t)} - \mathbf{w}_{j,r}^{(0)}, \boldsymbol{\xi}_i \rangle = \sum_{i'=1}^{n} \overline{\rho}_{j,r,i'}^{(t)} \big\|\boldsymbol{\xi}_{i'}\big\|_2^{-2} \cdot \langle \boldsymbol{\xi}_{i'}, \boldsymbol{\xi}_i \rangle + \sum_{i'=1}^{n} \underline{\rho}_{j,r,i'}^{(t)} \big\|\boldsymbol{\xi}_{i'}\big\|_2^{-2} \cdot \langle \boldsymbol{\xi}_{i'}, \boldsymbol{\xi}_i \rangle$$

$$= \sum_{i'=1}^{n} \underline{\rho}_{j,r,i'}^{(t)} \big\|\boldsymbol{\xi}_{i'}\big\|_2^{-2} \cdot \langle \boldsymbol{\xi}_{i'}, \boldsymbol{\xi}_i \rangle = \underline{\rho}_{j,r,i}^{(t)} + \sum_{i' \ne i} \underline{\rho}_{j,r,i'}^{(t)} \big\|\boldsymbol{\xi}_{i'}\big\|_2^{-2} \cdot \langle \boldsymbol{\xi}_{i'}, \boldsymbol{\xi}_i \rangle.$$

Note that we have

$$\left| \sum_{i' \ne i} \underline{\rho}_{j,r,i'}^{(t)} \big\|\boldsymbol{\xi}_{i'}\big\|_2^{-2} \cdot \langle \boldsymbol{\xi}_{i'}, \boldsymbol{\xi}_i \rangle \right| \le \sum_{i' \ne i} \big|\underline{\rho}_{j,r,i'}^{(t)}\big| \big\|\boldsymbol{\xi}_{i'}\big\|_2^{-2} \cdot \big|\langle \boldsymbol{\xi}_{i'}, \boldsymbol{\xi}_i \rangle\big|$$

$$\le 4\sqrt{\frac{\log(4n^2/\delta)}{d}} \sum_{i' \ne i} \big|\underline{\rho}_{j,r,i'}^{(t)}\big| = 4\sqrt{\frac{\log(4n^2/\delta)}{d}} \sum_{i' \ne i} \big|\underline{\rho}_{j,r,i'}^{(t)}\big|$$

$$\le 16\sqrt{\frac{\log(4n^2/\delta)}{d}} n \log(T^*),$$

where the first inequality is by triangle inequality; the second inequality is by Lemma C.3; the last inequality is by equation D.3 that $|\underline{\rho}_{j,r,i}^{(t)}| \le \log(T^*)$. We complete the proof for $j \ne y_i$.

Similarly, for $j = y_i$ we have

$$\langle \mathbf{w}_{j,r}^{(t)} - \mathbf{w}_{j,r}^{(0)}, \boldsymbol{\xi}_i \rangle = \underline{\rho}_{j,r,i}^{(t)} + \sum_{i' \ne i} \underline{\rho}_{j,r,i'}^{(t)} \big\|\boldsymbol{\xi}_{i'}\big\|_2^{-2} \cdot \langle \boldsymbol{\xi}_{i'}, \boldsymbol{\xi}_i \rangle,$$

and also

$$\left| \sum_{i' \ne i} \overline{\rho}_{j,r,i'}^{(t)} \big\|\boldsymbol{\xi}_{i'}\big\|_2^{-2} \cdot \langle \boldsymbol{\xi}_{i'}, \boldsymbol{\xi}_i \rangle \right| \le \sum_{i' \ne i} \big|\overline{\rho}_{j,r,i'}^{(t)}\big| \big\|\boldsymbol{\xi}_{i'}\big\|_2^{-2} \cdot \big|\langle \boldsymbol{\xi}_{i'}, \boldsymbol{\xi}_i \rangle\big|$$

$$\le 4\sqrt{\frac{\log(4n^2/\delta)}{d}} \sum_{i' \ne i} \big|\overline{\rho}_{j,r,i'}^{(t)}\big| = 4\sqrt{\frac{\log(4n^2/\delta)}{d}} \sum_{i' \ne i} \big|\overline{\rho}_{j,r,i'}^{(t)}\big|$$

$$\le 16\sqrt{\frac{\log(4n^2/\delta)}{d}} n \log(T^*),$$

where the first inequality is by triangle inequality; the second inequality is by Lemma C.3; the last inequality is by equation D.3. We complete the proof for $j = y_i$. ∎

Now, we define

$$\kappa = 56\sqrt{\frac{\log(4n^2/\delta)}{d}} n \log(T^*) + 10\sqrt{\log(12mn/\delta)} \cdot \sigma_0 \sigma_p \sqrt{d} + 64n \cdot \mathrm{SNR}^2 \log(T^*). \quad \text{(D.4)}$$

By Condition 3.1, it is easy to verify that $\kappa$ is a negligible term. The lemma below gives us a direct characterization of the neural networks output with respect to the time $t$.

**Lemma D.4** *Under Condition 3.1, suppose equation D.2 and equation D.3 hold at iteration $t$. Then, for all $r \in [m]$ it holds that*

$$F_{-y_i}(W_{-y_i}^{(t)}, \mathbf{x}_i) \leq \frac{\kappa}{2}, \quad -\frac{\kappa}{2} + \frac{1}{m} \sum_{r=1}^{m} \overline{\rho}_{y_i,r,i}^{(t)} \leq F_{y_i}(\mathbf{W}_{y_i}^{(t)}, \mathbf{x}_i) \leq \frac{1}{m} \sum_{r=1}^{m} \overline{\rho}_{y_i,r,i}^{(t)} + \frac{\kappa}{2}.$$

*Here, $\kappa$ is defined in equation D.4.*

**Proof** [Proof of Lemma D.4] Without loss of generality, we may assume $y_i = +1$. According to Lemma D.3, we have

$$F_{-y_i}(\mathbf{W}_{-y_i}^{(t)}, \mathbf{x}_i) = \frac{1}{m} \sum_{r=1}^{m} \left[ \sigma(\langle \mathbf{w}_{-y_i,r}^{(t)}, \mathbf{v} \rangle) + \sigma(\langle \mathbf{w}_{-y_i,r}^{(t)}, \boldsymbol{\xi}_i \rangle) \right]$$

$$\leq 32n \cdot \text{SNR}^2 \log(T^*) + \frac{1}{m} \sum_{r=1}^{m} \sigma\left( \langle \mathbf{w}_{-y_i,r}^{(0)}, \boldsymbol{\xi}_i \rangle + \sum_{i=1}^{n} \underline{\rho}_{-y_i,r,i}^{(t)} \cdot 16\sqrt{\frac{\log(4n^2/\delta)}{d}} \right)$$

$$\leq 32n \cdot \text{SNR}^2 \log(T^*) + 2\sqrt{\log(12mn/\delta)} \cdot \sigma_0 \sigma_p \sqrt{d} + 16n\sqrt{\frac{\log(4n^2/\delta)}{d}} \log(T^*)$$

$$\leq \kappa/2.$$

Here, the first inequality is by equation D.2, and the second inequality is due to equation D.3, Lemma C.4 and Condition 3.1. The last inequality is by the definition of $\kappa$ in equation D.4.

For the analysis of $F_{y_i}(\mathbf{W}_{y_i}^{(t)}, \mathbf{x}_i)$, we first have

$$\sigma(\langle \mathbf{w}_{y_i,r}^{(t)}, \boldsymbol{\xi}_i \rangle) \geq \langle \mathbf{w}_{y_i,r}^{(t)}, \boldsymbol{\xi}_i \rangle$$

$$\geq \langle \mathbf{w}_{y_i,r}^{(0)}, \boldsymbol{\xi}_i \rangle + \overline{\rho}_{y_i,r,i}^{(t)} - 16n\sqrt{\frac{\log(4n^2/\delta)}{d}} \log(T^*)$$

$$\geq \overline{\rho}_{y_i,r,i}^{(t)} - 2\sqrt{\log(12mn/\delta)} \cdot \sigma_0 \sigma_p \sqrt{d} - 16n\sqrt{\frac{\log(4n^2/\delta)}{d}} \log(T^*), \quad \text{(D.5)}$$

where the second inequality is by Lemma D.3, the third inequality comes from Lemma C.4. For the other side, we have

$$\sigma(\langle \mathbf{w}_{y_i,r}^{(t)}, \boldsymbol{\xi}_i \rangle) \leq |\langle \mathbf{w}_{y_i,r}^{(t)}, \boldsymbol{\xi}_i \rangle|$$

$$\leq |\langle \mathbf{w}_{y_i,r}^{(0)}, \boldsymbol{\xi}_i \rangle| + \overline{\rho}_{y_i,r,i}^{(t)} + 16n\sqrt{\frac{\log(4n^2/\delta)}{d}} \log(T^*)$$

$$\leq 2\sqrt{\log(12mn/\delta)} \cdot \sigma_0 \sigma_p \sqrt{d} + \overline{\rho}_{y_i,r,i}^{(t)} + 16n\sqrt{\frac{\log(4n^2/\delta)}{d}} \log(T^*). \quad \text{(D.6)}$$

Here, the second inequality is by triangle inequality and the last inequality is by Lemma C.4. Moreover, it is clear that for $\boldsymbol{\mu}_i \in \{\pm \mathbf{u}, \pm \mathbf{v}\}$,

$$0 \leq \frac{1}{m} \sum_{r=1}^{m} \sigma(\langle \mathbf{w}_{j,r}^{(t)}, \boldsymbol{\mu}_i \rangle) \leq 32n \cdot \text{SNR}^2 \log(T^*),$$

combined this with equation D.5 and equation D.6, we have

$$-\frac{\kappa}{2} + \frac{1}{m} \sum_{r=1}^{m} \overline{\rho}_{y_i,r,i}^{(t)} \leq F_{y_i}(\mathbf{W}_{y_i}^{(t)}, \mathbf{x}_i) \leq \frac{1}{m} \sum_{r=1}^{m} \overline{\rho}_{y_i,r,i}^{(t)} + \frac{\kappa}{2}.$$

This completes the proof. ∎

**Lemma D.5** *Under Condition 3.1, suppose equation D.2 and equation D.3 hold at iteration $t$. Then, for all $i \in [n]$, it holds that*

$$-\frac{\kappa}{2} + \frac{1}{m} \sum_{r=1}^{m} \overline{\rho}_{y_i,r,i}^{(t)} \leq y_i f(\mathbf{W}^{(t)}, \mathbf{x}_i) \leq \frac{\kappa}{2} + \frac{1}{m} \sum_{r=1}^{m} \overline{\rho}_{y_i,r,i}^{(t)}.$$

*Here, $\kappa$ is defined in equation D.4.*

**Proof** [Proof of Lemma D.5] Note that

$$y_i f(\mathbf{W}^{(t)}, \mathbf{x}_i) = F_{y_i}\big(\mathbf{W}^{(t)}_{y_i}, \mathbf{x}_i\big) - F_{-y_i}\big(\mathbf{W}^{(t)}_{-y_i}, \mathbf{x}_i\big),$$

the conclusion directly holds from Lemma D.4 ∎

**Lemma D.6** *Under Condition 3.1, suppose equation D.2 and equation D.3 hold for any iteration $t \leq T$. Then for any $t \leq T$, it holds that:*

1. *$1/m \cdot \sum_{r=1}^{m} \big[\overline{\rho}^{(t)}_{y_i,r,i} - \overline{\rho}^{(t)}_{y_k,r,k}\big] \leq \log(2) + 2\kappa + 4\sqrt{\log(2n/\delta)/m}$ for all $i, k \in [n]$.*
2. *Define $S_i^{(t)} := \big\{r \in [m] : \big\langle \mathbf{w}^{(t)}_{y_i,r}, \boldsymbol{\xi}_i \big\rangle > 0\big\}$ and $S_{j,r}^{(t)} := \big\{i \in [n] : y_i = j, \big\langle \mathbf{w}^{(t)}_{j,r}, \boldsymbol{\xi}_i \big\rangle > 0\big\}$. For all $i \in [n]$, $r \in [m]$ and $j \in \{\pm 1\}$, $S_i^{(0)} \subseteq S_i^{(t)}$, $S_{j,r}^{(0)} \subseteq S_{j,r}^{(t)}$.*
3. *Define $\overline{c} = \frac{2\eta\sigma_p^2 d}{nm}$, $\underline{c} = \frac{\eta\sigma_p^2 d}{3nm}$, $\overline{b} = e^{-\kappa}$ and $\underline{b} = e^{\kappa}$, and let $\overline{x}_t$, $\underline{x}_t$ be the unique solution of*

$$\overline{x}_t + \overline{b}e^{\overline{x}_t} = \overline{c}t + \overline{b},$$
$$\underline{x}_t + \underline{b}e^{\underline{x}_t} = \underline{c}t + \underline{b},$$

*it holds that*

$$\underline{x}_t \leq \frac{1}{m}\sum_{r=1}^{m} \overline{\rho}^{(t)}_{y_i,r,i} \leq \overline{x}_t + \overline{c}/(1+\overline{b}), \quad \frac{1}{1+\overline{b}e^{\overline{x}_t}} \leq -\ell_i'^{(t)} \leq \frac{1}{1+\underline{b}e^{\underline{x}_t}}$$

*for all $r \in [m]$ and $i \in [n]$.*

**Proof** [Proof of Lemma D.6] We use induction to prove this lemma. All conclusions hold naturally when $t = 0$. Now, suppose that there exists $\tilde{t} \leq T$ such that five conditions hold for any $0 \leq t \leq \tilde{t} - 1$, we prove that these conditions also hold for $t = \tilde{t}$.

We prove conclusion 1 first. By Lemma D.5, we easily see that

$$\left| y_i \cdot f\big(\mathbf{W}^{(t)}, \mathbf{x}_i\big) - y_k \cdot f\big(\mathbf{W}^{(t)}, \mathbf{x}_k\big) - \frac{1}{m}\sum_{r=1}^{m}\big[\overline{\rho}^{(t)}_{y_i,r,i} - \overline{\rho}^{(t)}_{y_k,r,k}\big] \right| \leq \kappa. \qquad \text{(D.7)}$$

Recall the update rule for $\overline{\rho}^{(t)}_{j,r,i}$

$$\overline{\rho}^{(t+1)}_{j,r,i} = \overline{\rho}^{(t)}_{j,r,i} - \frac{\eta}{nm} \cdot \ell_i'^{(t)} \cdot \mathbf{1}\big(\big\langle \mathbf{w}^{(t)}_{j,r}, \boldsymbol{\xi}_i \big\rangle \geq 0\big) \cdot \mathbf{1}\,(y_i = j)\, \|\boldsymbol{\xi}_i\|_2^2$$

Hence we have

$$\frac{1}{m}\sum_{r=1}^{m} \overline{\rho}^{(t+1)}_{j,r,i} = \frac{1}{m}\sum_{r=1}^{m} \overline{\rho}^{(t)}_{j,r,i} - \frac{\eta}{nm} \cdot \ell_i'^{(t)} \cdot \frac{1}{m}\sum_{r=1}^{m}\mathbf{1}\big(\big\langle \mathbf{w}^{(t)}_{j,r}, \boldsymbol{\xi}_i \big\rangle \geq 0\big) \cdot \mathbf{1}\,(y_i = j)\, \|\boldsymbol{\xi}_i\|_2^2$$

for all $j \in \{\pm 1\}, r \in [m], i \in [n]$ and $t \in [T^*]$. Also note that $S_i^{(t)} := \big\{r \in [m] : \big\langle \mathbf{w}^{(t)}_{y_i,r}, \boldsymbol{\xi}_i \big\rangle > 0\big\}$, we have

$$\frac{1}{m}\sum_{r=1}^{m}\big[\overline{\rho}^{(t+1)}_{y_i,r,i} - \overline{\rho}^{(t+1)}_{y_k,r,k}\big] = \frac{1}{m}\sum_{r=1}^{m}\big[\overline{\rho}^{(t)}_{y_i,r,i} - \overline{\rho}^{(t)}_{y_k,r,k}\big] - \frac{\eta}{nm^2} \cdot \big(|S_i^{(t)}|\ell_i'^{(t)} \cdot \|\boldsymbol{\xi}_i\|_2^2 - |S_k^{(t)}|\ell_k'^{(t)} \cdot \|\boldsymbol{\xi}_k\|_2^2\big).$$

We prove condition 1 in two cases: $1/m\sum_{r=1}^{m}\big[\overline{\rho}^{(\tilde{t}-1)}_{y_i,r,i} - \overline{\rho}^{(\tilde{t}-1)}_{y_k,r,k}\big] \leq \log(2) + 2\kappa + 3\sqrt{\log(2n/\delta)/m}$ and $1/m\sum_{r=1}^{m}\big[\overline{\rho}^{(\tilde{t}-1)}_{y_i,r,i} - \overline{\rho}^{(\tilde{t}-1)}_{y_k,r,k}\big] \geq \log(2) + 2\kappa + 3\sqrt{\log(2n/\delta)/m}$.

When $1/m\sum_{r=1}^{m}\big[\overline{\rho}^{(\tilde{t}-1)}_{y_i,r,i} - \overline{\rho}^{(\tilde{t}-1)}_{y_k,r,k}\big] \leq \log(2) + 2\kappa + 3\sqrt{\log(2n/\delta)/m}$, we have

$$\frac{1}{m}\sum_{r=1}^{m}\big[\overline{\rho}^{(\tilde{t})}_{y_i,r,i} - \overline{\rho}^{(\tilde{t})}_{y_k,r,k}\big] = \frac{1}{m}\sum_{r=1}^{m}\big[\overline{\rho}^{(\tilde{t}-1)}_{y_i,r,i} - \overline{\rho}^{(\tilde{t}-1)}_{y_k,r,k}\big]$$
$$- \frac{\eta}{nm^2} \cdot \big(|S_i^{(\tilde{t}-1)}|\ell_i'^{(\tilde{t}-1)} \cdot \|\boldsymbol{\xi}_i\|_2^2 - |S_k^{(\tilde{t}-1)}|\ell_k'^{(\tilde{t}-1)} \cdot \|\boldsymbol{\xi}_k\|_2^2\big)$$

$$\leq \frac{1}{m}\sum_{r=1}^{m}\big[\overline{\rho}_{y_i,r,i}^{(\tilde{t}-1)} - \overline{\rho}_{y_k,r,k}^{(\tilde{t}-1)}\big] + \frac{\eta}{nm}\|\boldsymbol{\xi}_i\|_2^2$$

$$\leq \log(2) + 3\sqrt{\log(2n/\delta)/m} + 2\kappa + \sqrt{\log(2n/\delta)/m}$$

$$\leq \log(2) + 4\sqrt{\log(2n/\delta)/m} + 2\kappa.$$

Here, the first inequality is by $\big|S_i^{(\tilde{t}-1)}\big| \leq m$ and $-\ell_i'^{(t)} \leq 1$, and the second inequality is by the condition of $\eta$ in Condition 3.1.

For when $1/m\sum_{r=1}^{m}\big[\overline{\rho}_{y_i,r,i}^{(\tilde{t}-1)} - \overline{\rho}_{y_k,r,k}^{(\tilde{t}-1)}\big] \geq \log(2) + 2\kappa + 3\sqrt{\log(2n/\delta)/m}$, from equation D.7 we have

$$y_i \cdot f\big(\mathbf{W}^{(\tilde{t}-1)}, \mathbf{x}_i\big) - y_k \cdot f\big(\mathbf{W}^{(\tilde{t}-1)}, \mathbf{x}_k\big) \geq \log(2) + 3\sqrt{\log(2n/\delta)/m},$$

hence

$$\frac{\ell_i'^{(\tilde{t}-1)}}{\ell_k'^{(\tilde{t}-1)}} \leq \exp\big(y_k \cdot f\big(\mathbf{W}^{(\tilde{t}-1)}, \mathbf{x}_k\big) - y_i \cdot f\big(\mathbf{W}^{(\tilde{t}-1)}, \mathbf{x}_i\big)\big) \leq \exp\big(-\kappa - 3\sqrt{\log(2n/\delta)/m}\big)/2.$$

$$(D.8)$$

Also from condition 2 we have $\big|S_i^{(\tilde{t}-1)}\big| = \big|S_i^{(0)}\big|$ and $\big|S_k^{(\tilde{t}-1)}\big| = \big|S_k^{(0)}\big|$, we have

$$\frac{\big|S_i^{(\tilde{t}-1)}\big|\big|\ell_i'^{(\tilde{t}-1)}\big|\|\boldsymbol{\xi}_i\|^2}{\big|S_k^{(\tilde{t}-1)}\big|\big|\ell_k'^{(\tilde{t}-1)}\big|\|\boldsymbol{\xi}_i\|^2} \leq \frac{\big|S_i^{(t)}\big|\big|\ell_i'^{(\tilde{t}-1)}\big|\|\boldsymbol{\xi}_i\|^2}{\big|S_k^{(0)}\big|\big|\ell_k'^{(\tilde{t}-1)}\big|\|\boldsymbol{\xi}_i\|^2}$$

$$\leq \frac{m}{(0.5m - \sqrt{2\log(2n/\delta)m})} \cdot \frac{e^{-3\sqrt{\log(2n/\delta)/m}}}{2} \cdot e^{-\kappa} \cdot \frac{1 + C\sqrt{\log(4n/\delta)/d}}{1 - C\sqrt{\log(4n/\delta)/d}}$$

$$< 1 * 1 = 1.$$

Here, the first inequality is by Lemma C.1, equation D.8 and Lemma C.3; the second inequality is by

$$\frac{1}{1 - \sqrt{2x}} \cdot e^{-3x} < 1 \quad \text{when } 0 < x < 0.1, \quad \kappa \gg 2C\sqrt{\log(4n/\delta)/d}.$$

By Lemma C.3, under event $\mathcal{E}$, we have

$$\big|\|\boldsymbol{\xi}_i\|_2^2 - d \cdot \sigma_p^2\big| \leq C_0\sigma_p^2 \cdot \sqrt{d\log(4n/\delta)}, \forall i \in [n].$$

Note that $d = \Omega(\log(4n/\delta))$ from Condition 3.1, it follows that

$$\big|S_i^{(\tilde{t}-1)}\big|\big(-\ell_i'^{(\tilde{t}-1)}\big) \cdot \|\boldsymbol{\xi}_i\|_2^2 < \big|S_k^{(\tilde{t}-1)}\big|\big(-\ell_k'^{(\tilde{t}-1)}\big) \cdot \|\boldsymbol{\xi}_k\|_2^2.$$

We conclude that

$$\frac{1}{m}\sum_{r=1}^{m}\big[\overline{\rho}_{y_i,r,i}^{(\tilde{t})} - \overline{\rho}_{y_k,r,k}^{(\tilde{t})}\big] \leq \frac{1}{m}\sum_{r=1}^{m}\big[\overline{\rho}_{y_i,r,i}^{(\tilde{t}-1)} - \overline{\rho}_{y_k,r,k}^{(\tilde{t}-1)}\big] \leq \log(2) + 2\kappa + 4\sqrt{\log(2n/\delta)/m}.$$

Hence conclusion 1 holds for $t = \tilde{t}$.

To prove conclusion 2, we prove $S_i^{(0)} \subseteq S_i^{(t)}$, and it is quite similar to prove $S_{j,r}^{(0)} = S_{j,r}^{(t)}$. Recall the update rule of $\langle\mathbf{w}_{j,r}^{(t)}, \boldsymbol{\xi}_i\rangle$, for $r \in S_i^{(0)}$ we have

$$\langle\mathbf{w}_{y_i,r}^{(\tilde{t})}, \boldsymbol{\xi}_i\rangle = \langle\mathbf{w}_{y_i,r}^{(\tilde{t}-1)}, \boldsymbol{\xi}_i\rangle - \frac{\eta}{nm} \cdot \underbrace{\ell_i'^{(\tilde{t}-1)} \cdot \|\boldsymbol{\xi}_i\|_2^2}_{I_3} - \frac{\eta}{nm} \cdot \underbrace{\sum_{i' \neq i} \ell_{i'}'^{(\tilde{t}-1)} \cdot \sigma'\big(\langle\mathbf{w}_{y_i,r}^{(\tilde{t}-1)}, \boldsymbol{\xi}_{i'}\rangle\big) \cdot \langle\boldsymbol{\xi}_{i'}, \boldsymbol{\xi}_i\rangle}_{I_4} \cdot$$

Lemma C.3 shows that

$$-I_3 \geq \big|\ell_i'^{(\tilde{t}-1)}\big|\sigma_p^2 d/2,$$

and

$$
\begin{aligned}
|I_4| &\leq \sum_{i' \neq i} \left| \ell_{i'}^{(\tilde{t}-1)} \right| \cdot \sigma' \left( \left\langle \mathbf{w}_{y_i,r}^{(\tilde{t}-1)}, \boldsymbol{\xi}_{i'} \right\rangle \right) \cdot \left| \left\langle \boldsymbol{\xi}_{i'}, \boldsymbol{\xi}_i \right\rangle \right| \\
&\leq \sum_{i' \neq i} \left| \ell_{i'}^{(\tilde{t}-1)} \right| \cdot 2\sigma_p^2 \cdot \sqrt{d \log(4n^2/\delta)} \leq 2n \left| \ell_i^{(\tilde{t}-1)} \right| \cdot 2\sigma_p^2 \cdot \sqrt{d \log(4n^2/\delta)},
\end{aligned}
$$

where the first inequality is by triangle inequality, the second inequality is by Lemma C.3 and the last inequality is by induction hypothesis of condition 3 at $t = \tilde{t} - 1$. By Condition 3.1, we can see $-I_3 \geq |I_4|$, hence we have

$$
\left\langle \mathbf{w}_{y_i,r}^{(\tilde{t})}, \boldsymbol{\xi}_i \right\rangle \geq \left\langle \mathbf{w}_{y_i,r}^{(\tilde{t}-1)}, \boldsymbol{\xi}_i \right\rangle > 0,
$$

which indicates that

$$
S_i^{(0)} \subseteq S_i^{(\tilde{t}-1)} \subseteq S_i^{(\tilde{t})}.
$$

We prove that

$$
S_i^{(0)} \subseteq S_i^{(\tilde{t})}.
$$

As for the last conclusion, recall that

$$
\frac{1}{m} \sum_{r=1}^m \overline{\rho}_{y_i,r,i}^{(t+1)} = \frac{1}{m} \sum_{r=1}^m \overline{\rho}_{y_i,r,i}^{(t)} - \frac{\eta}{nm} \cdot \ell_i'^{(t)} \cdot \frac{1}{m} \sum_{r=1}^m \mathbf{1}\left( \left\langle \mathbf{w}_{y_i,r}^{(t)}, \boldsymbol{\xi}_i \right\rangle \geq 0 \right) \cdot \mathbf{1}\left( y_i = j \right) \|\boldsymbol{\xi}_i\|_2^2,
$$

by condition 2 for $t \in [\tilde{t} - 1]$, we have

$$
\frac{1}{m} \sum_{r=1}^m \overline{\rho}_{y_i,r,i}^{(\tilde{t})} = \frac{1}{m} \sum_{r=1}^m \overline{\rho}_{y_i,r,i}^{(\tilde{t}-1)} - \frac{\eta}{nm} \cdot \frac{1}{1 + \exp\left( y_i f(\mathbf{W}^{(\tilde{t})}, \mathbf{x}_i) \right)} \cdot \frac{|S_i^{(0)}|}{m} \cdot \|\boldsymbol{\xi}_i\|_2^2,
$$

then Lemma C.1, Lemma C.3 and and Lemma D.5 give that

$$
\begin{aligned}
\frac{1}{m} \sum_{r=1}^m \overline{\rho}_{y_i,r,i}^{(\tilde{t})} &\leq \frac{1}{m} \sum_{r=1}^m \overline{\rho}_{y_i,r,i}^{(\tilde{t}-1)} + \frac{\eta}{nm} \cdot \frac{1}{1 + e^{-\kappa} \cdot e^{\frac{1}{m} \sum_{r=1}^m \overline{\rho}_{y_i,r,i}^{(\tilde{t}-1)}}} \cdot \|\boldsymbol{\xi}_i\|_2^2 \\
&\leq \frac{1}{m} \sum_{r=1}^m \overline{\rho}_{y_i,r,i}^{(\tilde{t}-1)} + \frac{\overline{c}}{1 + \overline{b} e^{\frac{1}{m} \sum_{r=1}^m \overline{\rho}_{y_i,r,i}^{(\tilde{t}-1)}}}, \\
\frac{1}{m} \sum_{r=1}^m \overline{\rho}_{y_i,r,i}^{(\tilde{t})} &\geq \frac{1}{m} \sum_{r=1}^m \overline{\rho}_{y_i,r,i}^{(\tilde{t}-1)} + \frac{\eta}{nm} \cdot \frac{1/2 - \sqrt{2 \log(2n/\delta)/m}}{1 + e^{\kappa} \cdot e^{\frac{1}{m} \sum_{r=1}^m \overline{\rho}_{y_i,r,i}^{(\tilde{t}-1)}}} \cdot \|\boldsymbol{\xi}_i\|_2^2 \\
&\geq \frac{1}{m} \sum_{r=1}^m \underline{\rho}_{y_i,r,i}^{(\tilde{t}-1)} + \frac{\underline{c}}{1 + \underline{b} e^{\frac{1}{m} \sum_{r=1}^m \underline{\rho}_{y_i,r,i}^{(\tilde{t}-1)}}}.
\end{aligned}
$$

Combined the two inequalities with Lemma D.1 completes the first result in the last conclusion. As for the second result, by Lemma D.5, it directly holds from

$$
\frac{1}{m} \sum_{r=1}^m \overline{\rho}_{y_i,r,i}^{(t)} - \kappa/2 \leq y_i f(\mathbf{W}^{(t)}, \mathbf{x}_i) \leq \frac{1}{m} \sum_{r=1}^m \overline{\rho}_{y_i,r,i}^{(t)} + \kappa/2,
$$
$$
\overline{c} \leq \kappa/2.
$$

This completes the proof of Lemma D.6. ∎

**Lemma D.7 (Restatement of Lemma 4.1)** *Under Condition 3.1, suppose equation D.2 and equation D.3 hold at iteration t. Then, for all $i, k \in [n]$, it holds that*

$$
\ell_i'^{(t)} / \ell_k'^{(t)} \leq 2 \cdot e^{3\kappa + 4\sqrt{\log(2n/\delta)/m}} = 2 + o(1).
$$

**Proof** [Proof of Lemma D.7] By Lemma D.6, we have $1/m \cdot \sum_{r=1}^{m} \left[ \overline{\rho}_{y_i,r,i}^{(t)} - \overline{\rho}_{y_k,r,k}^{(t)} \right] \leq \log(2) + 2\kappa + 4\sqrt{\log(2n/\delta)/m}$. The conclusion follows directly from Lemma D.5 and $\ell_i'^{(t)}/\ell_k'^{(t)} \leq \exp\{y_i f(\mathbf{W}^{(t)}, \mathbf{x}_i) - y_k f(\mathbf{W}^{(t)}, \mathbf{x}_k)\}$. $\blacksquare$

We are now ready to prove Proposition D.2.

**Proof** [Proof of Proposition D.2] Our proof is based on induction. The results are obvious at $t = 0$. Suppose that there exists $\tilde{T} \leq T^*$ such that the results in Proposition D.2 hold for all $0 \leq t \leq \tilde{T} - 1$. We target to prove that the results hold at $t = \tilde{T}$.

We first prove that equation D.3 holds. For $\underline{\rho}_{j,r,i}^{(t)}$, recall that $\underline{\rho}_{j,r,i}^{(t)} = 0$ when $j = y_i$, hence we only need to consider the case $j \neq y_i$. Easy to see $\underline{\rho}_{j,r,i}^{(t)} \leq 0$. When $\underline{\rho}_{j,r,i}^{(\tilde{T}-1)} \leq -2\sqrt{\log(12mn/\delta)} \cdot \sigma_0 \sigma_p \sqrt{d} - 16\sqrt{\frac{\log(4n^2/\delta)}{d}} n \log(T^*)$, by Lemma D.3 we can see

$$\langle \mathbf{w}_{j,r}^{(\tilde{T}-1)}, \boldsymbol{\xi}_i \rangle \leq \underline{\rho}_{j,r,i}^{(\tilde{T}-1)} + \langle \mathbf{w}_{j,r}^{(0)}, \boldsymbol{\xi}_i \rangle + 16\sqrt{\frac{\log(6n^2/\delta)}{d}} n \log(T^*) \leq 0,$$

hence

$$\begin{aligned}
\underline{\rho}_{j,r,i}^{(\tilde{T})} &= \underline{\rho}_{j,r,i}^{(\tilde{T}-1)} + \frac{\eta}{nm} \cdot \ell_i'^{(\tilde{T}-1)} \cdot \mathbf{1}(\langle \mathbf{w}_{j,r}^{(\tilde{T}-1)}, \boldsymbol{\xi}_i \rangle \geq 0) \cdot \mathbf{1}(y_i = -j) \|\boldsymbol{\xi}_i\|_2^2 \\
&= \underline{\rho}_{j,r,i}^{(\tilde{T}-1)} \geq -2\sqrt{\log(12mn/\delta)} \cdot \sigma_0 \sigma_p \sqrt{d} - 32\sqrt{\frac{\log(4n^2/\delta)}{d}} n \log(T^*).
\end{aligned}$$

Here the last inequality is by induction hypothesis. When $\underline{\rho}_{j,r,i}^{(\tilde{T}-1)} \geq -2\sqrt{\log(12mn/\delta)} \cdot \sigma_0 \sigma_p \sqrt{d} - 16\sqrt{\frac{\log(4n^2/\delta)}{d}} n \log(T^*)$, we have

$$\begin{aligned}
\underline{\rho}_{j,r,i}^{(\tilde{T})} &= \underline{\rho}_{j,r,i}^{(\tilde{T}-1)} + \frac{\eta}{nm} \cdot \ell_i'^{(\tilde{T}-1)} \cdot \mathbf{1}(\langle \mathbf{w}_{j,r}^{(\tilde{T}-1)}, \boldsymbol{\xi}_i \rangle \geq 0) \cdot \mathbf{1}(y_i = -j) \|\boldsymbol{\xi}_i\|_2^2 \\
&\geq -2\sqrt{\log(12mn/\delta)} \cdot \sigma_0 \sigma_p \sqrt{d} - 16\sqrt{\frac{\log(4n^2/\delta)}{d}} n \log(T^*) - \frac{3\eta \sigma_p^2 d}{2nm} \\
&\geq -2\sqrt{\log(12mn/\delta)} \cdot \sigma_0 \sigma_p \sqrt{d} - 32\sqrt{\frac{\log(4n^2/\delta)}{d}} n \log(T^*),
\end{aligned}$$

where the first inequality is by $0 < -\ell_i'^{(\tilde{T}-1)} \leq 1$ and $\|\boldsymbol{\xi}_i\|_2^2 \leq 3\sigma_p^2 d/2$, the second inequality is by $16\sqrt{\log(4n^2/\delta)/d} \cdot n \log(T^*) \geq 3\eta \sigma_n^2 d/2nm$ by the condition for $\eta$ in Condition 3.1. We complete the proof that $\underline{\rho}_{j,r,i}^{(t)} \geq -2\sqrt{\log(12mn/\delta)} \cdot \sigma_0 \sigma_p \sqrt{d} - 32\sqrt{\frac{\log(4n^2/\delta)}{d}} n \log(T^*)$. For $\overline{\rho}_{j,r,i}^{(t)}$, it is easy to see $\overline{\rho}_{j,r,i}^{(t)} = 0$ when $j \neq y_i$, hence we only consider the case $j = y_i$. Recall the update rule

$$\overline{\rho}_{j,r,i}^{(t+1)} = \overline{\rho}_{j,r,i}^{(t)} - \frac{\eta}{nm} \cdot \ell_i'^{(t)} \cdot \mathbf{1}(\langle \mathbf{w}_{j,r}^{(t)}, \boldsymbol{\xi}_i \rangle \geq 0) \cdot \mathbf{1}(y_i = j) \|\boldsymbol{\xi}_i\|_2^2.$$

When $j = y_i$, we can easily see that $\overline{\rho}_{j,r,i}^{(t)}$ increases when $t$ increases. Assume that $t_{j,r,i}$ be the last time such that $\overline{\rho}_{j,r,i}^{(t)} \leq 2\log(T^*)$, then for $\overline{\rho}_{j,r,i}^{(\tilde{T})}$ we have

$$\begin{aligned}
\overline{\rho}_{j,r,i}^{(\tilde{T})} &= \overline{\rho}_{j,r,i}^{(t_{j,r,i})} - \frac{\eta}{nm} \cdot \ell_i'^{(t_{j,r,i})} \cdot \mathbf{1}(\langle \mathbf{w}_{j,r}^{(t_{j,r,i})}, \boldsymbol{\xi}_i \rangle \geq 0) \cdot \mathbf{1}(y_i = j) \|\boldsymbol{\xi}_i\|_2^2 \\
&\quad - \sum_{t_{j,r,i} < t < \tilde{T}} \frac{\eta}{nm} \cdot \ell_i'^{(t)} \cdot \mathbf{1}(\langle \mathbf{w}_{j,r}^{(t)}, \boldsymbol{\xi}_i \rangle \geq 0) \cdot \mathbf{1}(y_i = j) \|\boldsymbol{\xi}_i\|_2^2 \\
&\leq 2\log(T^*) + \frac{3\eta \sigma_p^2 d}{2nm} - \sum_{t_{j,r,i} < t < \tilde{T}} \frac{3\eta \sigma_p^2 d}{2nm} \cdot \ell_i'^{(t)} \cdot \mathbf{1}(\langle \mathbf{w}_{j,r}^{(t)}, \boldsymbol{\xi}_i \rangle \geq 0) \cdot \mathbf{1}(y_i = j) \\
&\leq 3\log(T^*) - \sum_{t_{j,r,i} < t < \tilde{T}} \frac{3\eta \sigma_p^2 d}{2nm} \cdot \ell_i'^{(t)} \cdot \mathbf{1}(\langle \mathbf{w}_{j,r}^{(t)}, \boldsymbol{\xi}_i \rangle \geq 0) \cdot \mathbf{1}(y_i = j).
\end{aligned}$$

Here, the first inequality is by $-\ell_i'^{(t_{j,r,i})} \leq 1$ and $\|\boldsymbol{\xi}_i\|_2^2 \leq 3\sigma_p^2 d/2$ in Lemma C.3; the second inequality is by $\frac{3\eta\sigma_p^2 d}{2nm} \leq 2\log(T^*)$ which comes directly from the condition for $\eta$ in Condition 3.1. The only thing remained is to prove that

$$-\sum_{t_{j,r,i}<t<\widetilde{T}} \frac{3\eta\sigma_p^2 d}{2nm} \cdot \ell_i'^{(t)} \cdot \mathbf{1}\big(\langle \mathbf{w}_{j,r}^{(t)}, \boldsymbol{\xi}_i \rangle \geq 0\big) \cdot \mathbf{1}\big(y_i = j\big) \leq \log(T^*).$$

For $j = y_i$, we have that

$$\langle \mathbf{w}_{j,r}^{(t)}, \boldsymbol{\xi}_i \rangle \geq \langle \mathbf{w}_{j,r}^{(0)}, \boldsymbol{\xi}_i \rangle + \overline{\rho}_{j,r,i}^{(t)} - 16\sqrt{\frac{\log(4n^2/\delta)}{d}} n\log(T^*)$$

$$\geq -2\sqrt{\log(12mn/\delta)} \cdot \sigma_0\sigma_p\sqrt{d} + 2\log(T^*) - 16\sqrt{\frac{\log(4n^2/\delta)}{d}} n\log(T^*)$$

$$\geq 1.6\log(T^*),$$

where the first inequality is by Lemma D.3, and the second inequality is by $\overline{\rho}_{j,r,i}^{(t)} > 2\log(T^*)$ and Lemma C.4, and the third inequality is by $2\sqrt{\log(12mn/\delta)} \cdot \sigma_0\sigma_p\sqrt{d} + \sqrt{\log(16n^2/\delta)/d}n\log(T^*) \ll 0.4\log(T^*)$. Then it holds that

$$\big|\ell_i'^{(t)}\big| = \frac{1}{1 + \exp\big\{y_i \cdot \big[F_{+1}\big(\mathbf{W}_{+1}^{(t)}, \mathbf{x}_i\big) - F_{-1}\big(\mathbf{W}_{-1}^{(t)}, \mathbf{x}_i\big)\big]\big\}}$$

$$\leq \exp\big(-y_i F_{y_i}\big(\mathbf{W}_{y_i}^{(t)}, \mathbf{x}_i\big) + 0.1\big)$$

$$\leq \exp\big(-1/m \cdot \sum_{r=1}^{m} \sigma\big(\langle \mathbf{w}_{y_i,r}^{(t)}, \boldsymbol{\xi}_i \rangle\big) + 0.1\big) \leq 2\exp(-1.6\log(T^*)). \tag{D.9}$$

Here, the first inequality is by Lemma D.4 that $F_{-y_i}\big(\mathbf{W}_{-y_i}^{(t)}, \mathbf{x}_i\big) \leq 0.1$; the last inequality is by $\langle \mathbf{w}_{y_i,r}^{(t)}, \boldsymbol{\xi}_i \rangle \geq 1.6\log(T^*)$. By equation D.9, we have that

$$-\sum_{t_{j,r,i}<t<\widetilde{T}} \frac{3\eta\sigma_p^2 d}{2nm} \cdot \ell_i'^{(t)} \cdot \mathbf{1}\big(\langle \mathbf{w}_{j,r}^{(t)}, \boldsymbol{\xi}_i \rangle \geq 0\big) \cdot \mathbf{1}\big(y_i = j\big)$$

$$\leq \sum_{t_{j,r,i}<t<\widetilde{T}} \frac{3\eta\sigma_p^2 d}{2nm} \cdot \exp(-1.6\log(T^*)) \cdot \mathbf{1}\big(\langle \mathbf{w}_{j,r}^{(t)}, \boldsymbol{\xi}_i \rangle \geq 0\big) \cdot \mathbf{1}\big(y_i = j\big)$$

$$\leq \widetilde{T} \cdot \frac{3\eta\sigma_p^2 d}{2nm} \cdot \exp(-1.6\log(T^*)) \cdot \leq \frac{T^*}{(T^*)^{1.6}} \cdot \frac{3\eta\sigma_p^2 d}{2nm} \leq 1 \leq \log(T^*).$$

The last second inequality is by the condition for $\eta$ in Condition 3.1. We complete the proof that $0 \leq \overline{\rho}_{j,r,i}^{(t)} \leq 4\log(T^*)$.

We then prove that equation D.2 holds. To get equation D.2, we prove a stronger conclusion that there exists a $i^* \in [n]$ with $y_{i^*} = j$, such that for any $0 \leq t \leq T^*$,

$$|\langle \mathbf{w}_{j,r}^{(t)}, \mathbf{u} \rangle|/\overline{\rho}_{j,r,i^*}^{(t)} \leq 8n\mathrm{SNR}^2, \tag{D.10}$$

and $i^*$ can be taken as any sample from $S_{j,r}^{(0)}$. It is obviously true when $t = 1$. Suppose that it is true for $0 \leq t \leq \widetilde{T} - 1$, we aim to prove that equation D.10 holds at $t = \widetilde{T}$. Recall that

$$\langle \mathbf{w}_{j,r}^{(t+1)}, \mathbf{u} \rangle = \langle \mathbf{w}_{j,r}^{(t)}, \mathbf{u} \rangle - \frac{\eta j}{nm} \sum_{i \in S_{+\mathbf{u},+1} \cup S_{-\mathbf{u},-1}} \ell_i'^{(t)} \cdot \mathbf{1}\{\langle \mathbf{w}_{j,r}^{(t)}, \boldsymbol{\mu}_i \rangle > 0\}\|\boldsymbol{\mu}\|_2^2$$

$$+ \frac{\eta j}{nm} \sum_{i \in S_{-\mathbf{u},+1} \cup S_{+\mathbf{u},-1}} \ell_i'^{(t)} \cdot \mathbf{1}\{\langle \mathbf{w}_{j,r}^{(t)}, \boldsymbol{\mu}_i \rangle > 0\}\|\boldsymbol{\mu}\|_2^2$$

$$+ \frac{\eta j}{nm} \sum_{i \in S_{+\mathbf{v},-1} \cup S_{-\mathbf{v},+1}} \ell_i'^{(t)} \cdot \mathbf{1}\{\langle \mathbf{w}_{j,r}^{(t)}, \boldsymbol{\mu}_i \rangle > 0\}\|\boldsymbol{\mu}\|_2^2 \cos\theta$$

$$- \frac{\eta j}{nm} \sum_{i \in S_{-\mathbf{v},-1} \cup S_{+\mathbf{v},+1}} \ell_i'^{(t)} \cdot \mathbf{1}\{\langle \mathbf{w}_{j,r}^{(t)}, \boldsymbol{\mu}_i \rangle > 0\} \|\boldsymbol{\mu}\|_2^2 \cos\theta.$$

We have $|S_{+\mathbf{u},+1} \cup S_{-\mathbf{u},-1}| + |S_{-\mathbf{v},-1} \cup S_{+\mathbf{v},+1}| \leq n/2 + \widetilde{O}(\sqrt{n})$, hence

$$\left| \langle \mathbf{w}_{j,r}^{(\tilde{T})}, \mathbf{u} \rangle \right| \leq \left| \langle \mathbf{w}_{j,r}^{(\tilde{T}-1)}, \mathbf{u} \rangle \right| + 2(1 + o(1)) \cdot \left| \ell_{i^*}'^{(\tilde{T}-1)} \right| \cdot \frac{\eta \|\boldsymbol{\mu}\|_2^2 (1 + \cos\theta)}{2m}. \tag{D.11}$$

Here, we utilize the third conclusion in the induction hypothesis. Moreover, for $\overline{\rho}_{j,r,i}^{(\tilde{T})}$ we have that $\langle \mathbf{w}_{y_{i^*},r}^{\tilde{T}-1}, \boldsymbol{\xi}_{i^*} \rangle \geq 0$ by the condition in Lemma D.6, therefore it holds that

$$\overline{\rho}_{j,r,i^*}^{(\tilde{T})} = \overline{\rho}_{j,r,i^*}^{(\tilde{T}-1)} - \frac{\eta}{nm} \cdot \ell_{i^*}'^{(\tilde{T}-1)} \cdot \|\boldsymbol{\xi}_{i^*}\|_2^2 \geq \overline{\rho}_{j,r,i^*}^{(\tilde{T}-1)} - \frac{\eta}{nm} \cdot \ell_{i^*}'^{(\tilde{T}-1)} \cdot \sigma_p^2 d/2. \tag{D.12}$$

Hence we have

$$\frac{\left| \langle \mathbf{w}_{j,r}^{(\tilde{T})}, \mathbf{u} \rangle \right|}{\overline{\rho}_{j,r,i^*}^{(\tilde{T})}} \leq \frac{\left| \langle \mathbf{w}_{j,r}^{(\tilde{T}-1)}, \mathbf{u} \rangle \right| + 2 \cdot \left| \ell_{i^*}'^{(\tilde{T}-1)} \right| \cdot \frac{2\eta \|\boldsymbol{\mu}\|_2^2}{m}}{\overline{\rho}_{j,r,i^*}^{(\tilde{T}-1)} - \frac{\eta}{nm} \cdot \ell_{i^*}'^{(\tilde{T}-1)} \cdot \sigma_p^2 d/2}$$

$$\leq \max \left\{ \frac{\left| \langle \mathbf{w}_{j,r}^{(\tilde{T}-1)}, \mathbf{u} \rangle \right|}{\overline{\rho}_{j,r,i^*}^{(\tilde{T}-1)}}, \frac{4 \left| \ell_{i^*}'^{(\tilde{T}-1)} \right| \|\boldsymbol{\mu}\|_2^2}{\left| \ell_{i^*}'^{(\tilde{T}-1)} \right| \sigma_p^2 d/(2n)} \right\} \leq 8n\text{SNR}^2,$$

where the first inequality is by equation D.11 and equation D.12, the second inequality is by condition 3 in Lemma D.6 and $(a_1 + a_2)/(b_1 + b_2) \leq \max(a_1/b_1, a_2/b_2)$ for $a_1, a_2, b_1, b_2 > 0$, and the third inequality is by induction hypothesis. We hence completes the proof that $\left| \langle \mathbf{w}_{j,r}^{(\tilde{T})}, \mathbf{u} \rangle \right| \leq 32n \cdot \text{SNR}^2 \log(T^*)$. The proof of $\left| \langle \mathbf{w}_{j,r}^{(\tilde{T})}, \mathbf{v} \rangle \right| \leq 32n \cdot \text{SNR}^2 \log(T^*)$ is exactly the same, and we thus omit it. ∎

We summarize the conclusions above and thus have the following proposition:

**Proposition D.8** *If Condition 3.1 holds, then for any $0 \leq t \leq T^*$, $j \in \{\pm 1\}$, $r \in [m]$ and $i \in [n]$, it holds that*

$$0 \leq |\langle \mathbf{w}_{j,r}^{(t)}, \mathbf{u} \rangle|, |\langle \mathbf{w}_{j,r}^{(t)}, \mathbf{v} \rangle| \leq 32n \cdot \text{SNR}^2 \log(T^*),$$

$$0 \leq \overline{\rho}_{j,r,i}^{(t)} \leq 4\log(T^*), \quad 0 \geq \underline{\rho}_{j,r,i}^{(t)} \geq -2\sqrt{\log(12mn/\delta)} \cdot \sigma_0 \sigma_p \sqrt{d} - 32\sqrt{\frac{\log(4n^2/\delta)}{d}} n \log(T^*),$$

*and for any $i^* \in S_{j,r}^{(0)}$ it holds that*

$$|\langle \mathbf{w}_{j,r}^{(t)}, \mathbf{u} \rangle|/\overline{\rho}_{j,r,i^*}^{(t)} \leq 8n \cdot \text{SNR}^2, \quad |\langle \mathbf{w}_{j,r}^{(t)}, \mathbf{v} \rangle|/\overline{\rho}_{j,r,i^*}^{(t)} \leq 8n \cdot \text{SNR}^2.$$

*Moreover, the following conclusions hold:*

1. *$1/m \cdot \sum_{r=1}^m \left[ \overline{\rho}_{y_i,r,i}^{(t)} - \overline{\rho}_{y_k,r,k}^{(t)} \right] \leq \log(2) + 2\kappa + 4\sqrt{\log(2n/\delta)/m}$ for all $i, k \in [n]$.*
2. *$-\frac{\kappa}{2} + \frac{1}{m} \sum_{r=1}^m \overline{\rho}_{y_i,r,i}^{(t)} \leq y_i f(\mathbf{W}^{(t)}, \mathbf{x}_i) \leq \frac{\kappa}{2} + \frac{1}{m} \sum_{r=1}^m \overline{\rho}_{y_i,r,i}^{(t)}$ for any $i \in [n]$. For any $i, k \in [n]$, it holds that $\ell_i'^{(t)}/\ell_k'^{(t)} \leq 2 + o(1)$.*
3. *Define $S_i^{(t)} := \{r \in [m] : \langle \mathbf{w}_{y_i,r}^{(t)}, \boldsymbol{\xi}_i \rangle > 0\}$ and $S_{j,r}^{(t)} := \{i \in [n] : y_i = j, \langle \mathbf{w}_{j,r}^{(t)}, \boldsymbol{\xi}_i \rangle > 0\}$. For all $i \in [n]$, $r \in [m]$ and $j \in \{\pm 1\}$, $S_i^{(0)} \subseteq S_i^{(t)}$, $S_{j,r}^{(0)} \subseteq S_{j,r}^{(t)}$.*
4. *Define $\overline{c} = \frac{2\eta \sigma_p^2 d}{nm}$, $\underline{c} = \frac{\eta \sigma_p^2 d}{3nm}$, $\overline{b} = e^{-\kappa}$ and $\underline{b} = e^\kappa$, and let $\overline{x}_t$, $\underline{x}_t$ be the unique solution of*

$$\overline{x}_t + \overline{b}e^{\overline{x}_t} = \overline{c}t + \overline{b},$$
$$\underline{x}_t + \underline{b}e^{\underline{x}_t} = \underline{c}t + \underline{b},$$

*it holds that*

$$\underline{x}_t \leq \frac{1}{m} \sum_{r=1}^m \overline{\rho}_{y_i,r,i}^{(t)} \leq \overline{x}_t + \overline{c}/(1 + \overline{b}), \quad \frac{1}{1 + \overline{b}e^{\overline{x}_t}} \leq -\ell_i'^{(t)} \leq \frac{1}{1 + \underline{b}e^{\underline{x}_t}}$$

*for all $r \in [m]$ and $i \in [n]$.*

The results in Proposition D.8 are deemed adequate for demonstrating the convergence of the training loss, and we shall proceed to establish in Section F. It is worthy noting that the gap between $\underline{x}_t$ and $\overline{x}_t$ is small. Indeed, we have the following lemma:

**Lemma D.9** *It is easy to check that*

$$\log\left(\frac{\eta\sigma_p^2 d}{8nm}t + \frac{2}{3}\right) \leq \overline{x}_t \leq \log\left(\frac{2\eta\sigma_p^2 d}{nm}t + 1\right),$$

$$\log\left(\frac{\eta\sigma_p^2 d}{8nm}t + \frac{2}{3}\right) \leq \underline{x}_t \leq \log\left(\frac{2\eta\sigma_p^2 d}{nm}t + 1\right).$$

**Proof** [Proof of Lemma D.9] We can easily obtain the inequality by

$$\overline{b}e^{\overline{x}_t} \leq \overline{x}_t + \overline{b}e^{\overline{x}_t} \leq 1.5\overline{b}e^{\overline{x}_t}, \quad \underline{b}e^{\underline{x}_t} \leq \underline{x}_t + \underline{b}e^{\underline{x}_t} \leq 1.5\underline{b}e^{\underline{x}_t},$$

and $\eta\sigma_p^2 d/(8nm) \leq \overline{c}/\overline{b} \leq 2\eta\sigma_p^2 d/(nm)$, $\eta\sigma_p^2 d/(8nm) \leq \underline{c}/\underline{b} \leq 2\eta\sigma_p^2 d/(nm)$. ∎

# E   SIGNAL LEARNING AND NOISE MEMORIZATION ANALYSIS

In this section, we apply the precise results obtained from previous sections into the analysis of signal learning and noise memorization.

## E.1   SIGNAL LEARNING

We start the analysis of signal learning. we have following lemma:

**Lemma E.1 (Conclusion 1 in Proposition A.1)** *Under Condition 3.1, the following conclusions hold:*

1. *If $\langle \mathbf{w}_{+1,r}^{(0)}, \mathbf{u} \rangle > 0(< 0)$, then $\langle \mathbf{w}_{+1,r}^{(t)}, \mathbf{u} \rangle$ strictly increases (decreases) with $t \in [T^*]$;*
2. *If $\langle \mathbf{w}_{-1,r}^{(0)}, \mathbf{v} \rangle > 0(< 0)$, then $\langle \mathbf{w}_{-1,r}^{(t)}, \mathbf{v} \rangle$ strictly increases (decreases) with $t \in [T^*]$;*
3. *$|\langle \mathbf{w}_{+1,r}^{(t)}, \mathbf{v} \rangle| \leq |\langle \mathbf{w}_{+1,r}^{(0)}, \mathbf{v} \rangle| + \eta\|\boldsymbol{\mu}\|_2^2/m$, $|\langle \mathbf{w}_{-1,r}^{(t)}, \mathbf{u} \rangle| \leq |\langle \mathbf{w}_{-1,r}^{(0)}, \mathbf{u} \rangle| + \eta\|\boldsymbol{\mu}\|_2^2/m$ for all $t \in [T^*]$ and $r \in [m]$.*

*Moreover, it holds that*

$$\langle \mathbf{w}_{+1,r}^{(t+1)}, \mathbf{u} \rangle \geq \langle \mathbf{w}_{+1,r}^{(t)}, \mathbf{u} \rangle - \frac{c\eta\|\boldsymbol{\mu}\|_2^2}{nm} \sum_{i \in S_{+\mathbf{u},+1}} \ell_i'^{(t)}, \quad \langle \mathbf{w}_{+1,r}^{(0)}, \mathbf{u} \rangle > 0;$$

$$\langle \mathbf{w}_{+1,r}^{(t+1)}, \mathbf{u} \rangle \leq \langle \mathbf{w}_{+1,r}^{(t)}, \mathbf{u} \rangle + \frac{c\eta\|\boldsymbol{\mu}\|_2^2}{nm} \cdot \sum_{i \in S_{-\mathbf{u},+1}} \ell_i'^{(t)}, \quad \langle \mathbf{w}_{+1,r}^{(0)}, \mathbf{u} \rangle < 0;$$

$$\langle \mathbf{w}_{-1,r}^{(t+1)}, \mathbf{v} \rangle \geq \langle \mathbf{w}_{-1,r}^{(t)}, \mathbf{v} \rangle - \frac{c\eta\|\boldsymbol{\mu}\|_2^2}{nm} \cdot \sum_{i \in S_{+\mathbf{v},-1}} \ell_i'^{(t)}, \quad \langle \mathbf{w}_{-1,r}^{(0)}, \mathbf{v} \rangle > 0;$$

$$\langle \mathbf{w}_{-1,r}^{(t+1)}, \mathbf{v} \rangle \leq \langle \mathbf{w}_{-1,r}^{(t)}, \mathbf{v} \rangle + \frac{c\eta\|\boldsymbol{\mu}\|_2^2}{nm} \cdot \sum_{i \in S_{-\mathbf{v},-1}} \ell_i'^{(t)}, \quad \langle \mathbf{w}_{-1,r}^{(0)}, \mathbf{v} \rangle < 0$$

*for some constant $c > 0$. Similarly, it also holds that*

$$\langle \mathbf{w}_{+1,r}^{(t+1)}, \mathbf{u} \rangle \leq \langle \mathbf{w}_{+1,r}^{(t)}, \mathbf{u} \rangle - \frac{C\eta\|\boldsymbol{\mu}\|_2^2}{nm} \sum_{i \in [n]} \ell_i'^{(t)}, \quad \langle \mathbf{w}_{+1,r}^{(0)}, \mathbf{u} \rangle > 0;$$

$$\langle \mathbf{w}_{+1,r}^{(t+1)}, \mathbf{u} \rangle \leq \langle \mathbf{w}_{+1,r}^{(t)}, \mathbf{u} \rangle + \frac{C\eta\|\boldsymbol{\mu}\|_2^2}{nm} \sum_{i \in [n]} \ell_i'^{(t)}, \quad \langle \mathbf{w}_{+1,r}^{(0)}, \mathbf{u} \rangle < 0;$$

$$\langle \mathbf{w}_{-1,r}^{(t+1)}, \mathbf{v} \rangle \geq \langle \mathbf{w}_{-1,r}^{(t)}, \mathbf{v} \rangle - \frac{C\eta\|\boldsymbol{\mu}\|_2^2}{nm} \sum_{i \in [n]} \ell_i'^{(t)}, \quad \langle \mathbf{w}_{-1,r}^{(0)}, \mathbf{v} \rangle > 0;$$

$$\langle \mathbf{w}_{-1,r}^{(t+1)}, \mathbf{v}\rangle \le \langle \mathbf{w}_{-1,r}^{(t)}, \mathbf{v}\rangle + \frac{C\eta\|\boldsymbol{\mu}\|_2^2}{nm}\sum_{i\in[n]}\ell_i'^{(t)}, \quad \langle \mathbf{w}_{-1,r}^{(0)}, \mathbf{v}\rangle < 0,$$

*for some constant $C > 0$.*

**Proof** [Proof of Lemma E.1] Recall that the update rule for inner product can be written as

$$\langle \mathbf{w}_{j,r}^{(t+1)}, \mathbf{u}\rangle = \langle \mathbf{w}_{j,r}^{(t)}, \mathbf{u}\rangle - \frac{\eta j}{nm}\sum_{i\in S_{+\mathbf{u},+1}\cup S_{-\mathbf{u},-1}}\ell_i'^{(t)}\cdot\mathbf{1}\{\langle \mathbf{w}_{j,r}^{(t)}, \boldsymbol{\mu}_i\rangle > 0\}\|\boldsymbol{\mu}\|_2^2$$
$$+ \frac{\eta j}{nm}\sum_{i\in S_{-\mathbf{u},+1}\cup S_{+\mathbf{u},-1}}\ell_i'^{(t)}\cdot\mathbf{1}\{\langle \mathbf{w}_{j,r}^{(t)}, \boldsymbol{\mu}_i\rangle > 0\}\|\boldsymbol{\mu}\|_2^2$$
$$+ \frac{\eta j}{nm}\sum_{i\in S_{+\mathbf{v},-1}\cup S_{-\mathbf{v},+1}}\ell_i'^{(t)}\cdot\mathbf{1}\{\langle \mathbf{w}_{j,r}^{(t)}, \boldsymbol{\mu}_i\rangle > 0\}\|\boldsymbol{\mu}\|_2^2\cos\theta$$
$$- \frac{\eta j}{nm}\sum_{i\in S_{-\mathbf{v},-1}\cup S_{+\mathbf{v},+1}}\ell_i'^{(t)}\cdot\mathbf{1}\{\langle \mathbf{w}_{j,r}^{(t)}, \boldsymbol{\mu}_i\rangle > 0\}\|\boldsymbol{\mu}\|_2^2\cos\theta,$$

and

$$\langle \mathbf{w}_{j,r}^{(t+1)}, \mathbf{v}\rangle = \langle \mathbf{w}_{j,r}^{(t)}, \mathbf{v}\rangle - \frac{\eta j}{nm}\sum_{i\in S_{+\mathbf{u},+1}\cup S_{-\mathbf{u},-1}}\ell_i'^{(t)}\cdot\mathbf{1}\{\langle \mathbf{w}_{j,r}^{(t)}, \boldsymbol{\mu}_i\rangle > 0\}\|\boldsymbol{\mu}\|_2^2\cos\theta$$
$$+ \frac{\eta j}{nm}\sum_{i\in S_{-\mathbf{u},+1}\cup S_{+\mathbf{u},-1}}\ell_i'^{(t)}\cdot\mathbf{1}\{\langle \mathbf{w}_{j,r}^{(t)}, \boldsymbol{\mu}_i\rangle > 0\}\|\boldsymbol{\mu}\|_2^2\cos\theta$$
$$+ \frac{\eta j}{nm}\sum_{i\in S_{+\mathbf{v},-1}\cup S_{-\mathbf{v},+1}}\ell_i'^{(t)}\cdot\mathbf{1}\{\langle \mathbf{w}_{j,r}^{(t)}, \boldsymbol{\mu}_i\rangle > 0\}\|\boldsymbol{\mu}\|_2^2$$
$$- \frac{\eta j}{nm}\sum_{i\in S_{-\mathbf{v},-1}\cup S_{+\mathbf{v},+1}}\ell_i'^{(t)}\cdot\mathbf{1}\{\langle \mathbf{w}_{j,r}^{(t)}, \boldsymbol{\mu}_i\rangle > 0\}\|\boldsymbol{\mu}\|_2^2.$$

When $\langle \mathbf{w}_{+1,r}^{(0)}, \mathbf{u}\rangle > 0$, assume that for any $0 \le t \le \tilde{T} - 1$ such that $\langle \mathbf{w}_{+1,r}^{(t)}, \mathbf{u}\rangle > 0$, there are two cases for $\langle \mathbf{w}_{+1,r}^{(t)}, \mathbf{v}\rangle$. When $\langle \mathbf{w}_{+1,r}^{(\tilde{T}-1)}, \mathbf{v}\rangle < 0$, the simplified update rule of $\langle \mathbf{w}_{+1,r}^{(t)}, \mathbf{u}\rangle$ have the four terms below, which have the relation:

$$\frac{\eta}{nm}\left(\sum_{i\in S_{-\mathbf{v},+1}}\ell_i'^{(\tilde{T}-1)} - \sum_{i\in S_{-\mathbf{v},-1}}\ell_i'^{(\tilde{T}-1)}\right)\cdot\|\boldsymbol{\mu}\|_2^2\cos\theta > 0,$$

$$\frac{\eta}{nm}\left(-\sum_{i\in S_{+\mathbf{u},+1}}\ell_i'^{(\tilde{T}-1)} + \sum_{i\in S_{+\mathbf{u},-1}}\ell_i'^{(\tilde{T}-1)}\right)\cdot\|\boldsymbol{\mu}\|_2^2 > 0$$

from Lemma C.2 and Proposition D.8. Hence we have

$$\langle \mathbf{w}_{+1,r}^{(\tilde{T})}, \mathbf{u}\rangle \ge \langle \mathbf{w}_{+1,r}^{(\tilde{T}-1)}, \mathbf{u}\rangle + \frac{\eta}{nm}\left(-\sum_{i\in S_{+\mathbf{u},+1}}\ell_i'^{(\tilde{T}-1)} + \sum_{i\in S_{+\mathbf{u},-1}}\ell_i'^{(\tilde{T}-1)}\right)\cdot\|\boldsymbol{\mu}\|_2^2$$
$$\ge \langle \mathbf{w}_{+1,r}^{(\tilde{T}-1)}, \mathbf{u}\rangle - \frac{\eta\|\boldsymbol{\mu}\|_2^2}{nm}\cdot\sum_{i\in S_{+\mathbf{u},+1}}\ell_i'^{(\tilde{T}-1)}\left(1 - \frac{2pn(1+o(1))}{(1-p)n(1-o(1))}\right)$$
$$\ge \langle \mathbf{w}_{+1,r}^{(\tilde{T}-1)}, \mathbf{u}\rangle - \frac{c\eta\|\boldsymbol{\mu}\|_2^2}{nm}\cdot\sum_{i\in S_{+\mathbf{u},+1}}\ell_i'^{(\tilde{T}-1)}.$$

Here, the second inequality is by Proposition D.8, and the third inequality is by $p < 1/C$ fixed for large $C$ in Condition 3.1. We prove the case when $\langle \mathbf{w}_{+1,r}^{(\tilde{T}-1)}, \mathbf{v}\rangle < 0$.

When $\langle \mathbf{w}_{+1,r}^{(\tilde{T}-1)}, \mathbf{v}\rangle > 0$, the update rule can be simplified as

$$\langle \mathbf{w}_{+1,r}^{(\tilde{T})}, \mathbf{u}\rangle = \langle \mathbf{w}_{+1,r}^{(\tilde{T}-1)}, \mathbf{u}\rangle$$

$$- \frac{\eta\|\boldsymbol{\mu}\|_2^2}{nm}\left(\sum_{i \in S_{+\mathbf{u},+1}} \ell_i^{\prime(\tilde{T}-1)} - \sum_{i \in S_{+\mathbf{u},-1}} \ell_i^{\prime(\tilde{T}-1)} - \sum_{i \in S_{+\mathbf{v},-1}} \ell_i^{\prime(\tilde{T}-1)} \cos\theta + \sum_{i \in S_{+\mathbf{v},+1}} \ell_i^{\prime(\tilde{T}-1)} \cos\theta\right).$$

$$(E.1)$$

We can select $i^*$ such that

$$- \sum_{i \in S_{+\mathbf{u},-1}} \ell_i^{\prime(\tilde{T}-1)} \le pn/4 \cdot \ell_{i^*}^{\prime(\tilde{T}-1)} \cdot (1 + o(1)),$$

by the union bound in Proposition D.8, we have that

$$\sum_{i \in S_{+\mathbf{u},-1}} \ell_i^{\prime(\tilde{T}-1)} \Big/ \sum_{i \in S_{+\mathbf{u},+1}} \ell_i^{\prime(\tilde{T}-1)} \le \frac{pn/4 \cdot \ell_{i^*}^{\tilde{T}-1}(1+o(1))}{\sum_{i \in S_{+\mathbf{u},+1}} \ell_i^{\prime(\tilde{T}-1)}} \le \frac{2pn(1+o(1))}{(1-p)n(1-o(1))}.$$

Applying similar conclusions above, equation E.1 shows that

$$\langle \mathbf{w}_{+1,r}^{(\tilde{T})}, \mathbf{u}\rangle \ge \langle \mathbf{w}_{+1,r}^{(\tilde{T}-1)}, \mathbf{u}\rangle$$
$$- \frac{\eta\|\boldsymbol{\mu}\|_2^2}{nm} \sum_{i \in S_{+\mathbf{u},+1}} \ell_i^{\prime(\tilde{T}-1)} \cdot \left(1 - \frac{2pn(1+o(1))}{(1-p)n(1-o(1))} - \frac{2\cos\theta(1+o(1))}{1-o(1)} + \frac{p\cos\theta(1-o(1))}{2(1-p)(1+o(1))}\right).$$

Note that $\cos\theta < 1/2$ is fixed, and $p < 1/C$ for sufficient large $C$, we conclude that when $\langle \mathbf{w}_{+1,r}^{(\tilde{T}-1)}, \mathbf{v}\rangle > 0$,

$$\langle \mathbf{w}_{+1,r}^{(\tilde{T})}, \mathbf{u}\rangle \ge \langle \mathbf{w}_{+1,r}^{(\tilde{T}-1)}, \mathbf{u}\rangle - \frac{c\eta\|\boldsymbol{\mu}\|_2^2}{nm} \cdot \sum_{i \in S_{+\mathbf{u},+1}} \ell_i^{\prime(\tilde{T}-1)}$$

for some constant $c > 0$. Similarly, we have

$$\langle \mathbf{w}_{+1,r}^{(t+1)}, \mathbf{u}\rangle \le \langle \mathbf{w}_{+1,r}^{(t)}, \mathbf{u}\rangle + \frac{c\eta\|\boldsymbol{\mu}\|_2^2}{nm} \cdot \sum_{i \in S_{-\mathbf{u},+1}} \ell_i^{\prime(t)}, \quad \langle \mathbf{w}_{+1,r}^{(0)}, \mathbf{u}\rangle < 0;$$

$$\langle \mathbf{w}_{-1,r}^{(t+1)}, \mathbf{v}\rangle \ge \langle \mathbf{w}_{-1,r}^{(t)}, \mathbf{v}\rangle - \frac{c\eta\|\boldsymbol{\mu}\|_2^2}{nm} \cdot \sum_{i \in S_{+\mathbf{v},-1}} \ell_i^{\prime(t)} \quad \langle \mathbf{w}_{-1,r}^{(0)}, \mathbf{v}\rangle > 0;$$

$$\langle \mathbf{w}_{-1,r}^{(t+1)}, \mathbf{v}\rangle \le \langle \mathbf{w}_{-1,r}^{(t)}, \mathbf{v}\rangle + \frac{c\eta\|\boldsymbol{\mu}\|_2^2}{nm} \cdot \sum_{i \in S_{-\mathbf{v},-1}} \ell_i^{\prime(t)}, \quad \langle \mathbf{w}_{-1,r}^{(0)}, \mathbf{v}\rangle < 0.$$

This completes the conclusion 1 and 2.

To prove result 3, it is easy to verify that $|\langle \mathbf{w}_{+1,r}^{(0)}, \mathbf{v}\rangle| \le |\langle \mathbf{w}_{+1,r}^{(0)}, \mathbf{v}\rangle| + \eta\|\boldsymbol{\mu}\|_2^2/(2m)$, assume that $|\langle \mathbf{w}_{+1,r}^{(t)}, \mathbf{v}\rangle| \le |\langle \mathbf{w}_{+1,r}^{(0)}, \mathbf{v}\rangle| + \eta\|\boldsymbol{\mu}\|_2^2/(2m)$, we then prove $|\langle \mathbf{w}_{+1,r}^{(t+1)}, \mathbf{v}\rangle| \le |\langle \mathbf{w}_{+1,r}^{(0)}, \mathbf{v}\rangle| + \eta\|\boldsymbol{\mu}\|_2^2/(2m)$. Recall the update rule, we have

$$\langle \mathbf{w}_{+1,r}^{(t+1)}, \mathbf{v}\rangle = \langle \mathbf{w}_{+1,r}^{(t)}, \mathbf{v}\rangle - \frac{\eta}{nm} \sum_{i \in S_{+\mathbf{u},+1} \cup S_{-\mathbf{u},-1}} \ell_i^{\prime(t)} \cdot \mathbf{1}\{\langle \mathbf{w}_{+1,r}^{(t)}, \boldsymbol{\mu}_i\rangle > 0\}\|\boldsymbol{\mu}\|_2^2 \cos\theta$$

$$+ \frac{\eta}{nm} \sum_{i \in S_{-\mathbf{u},+1} \cup S_{+\mathbf{u},-1}} \ell_i^{\prime(t)} \cdot \mathbf{1}\{\langle \mathbf{w}_{+1,r}^{(t)}, \boldsymbol{\mu}_i\rangle > 0\}\|\boldsymbol{\mu}\|_2^2 \cos\theta$$

$$+ \frac{\eta}{nm} \sum_{i \in S_{+\mathbf{v},-1} \cup S_{-\mathbf{v},+1}} \ell_i^{\prime(t)} \cdot \mathbf{1}\{\langle \mathbf{w}_{+1,r}^{(t)}, \boldsymbol{\mu}_i\rangle > 0\}\|\boldsymbol{\mu}\|_2^2$$

$$- \frac{\eta}{nm} \sum_{i \in S_{-\mathbf{v},-1} \cup S_{+\mathbf{v},+1}} \ell_i^{\prime(t)} \cdot \mathbf{1}\{\langle \mathbf{w}_{+1,r}^{(t)}, \boldsymbol{\mu}_i\rangle > 0\}\|\boldsymbol{\mu}\|_2^2$$

We prove the case for $\langle \mathbf{w}_{+1,r}^{(t)}, \mathbf{v}\rangle > 0$. The opposite case for $\langle \mathbf{w}_{+1,r}^{(t)}, \mathbf{v}\rangle < 0$ is similar and we thus omit it. If $\langle \mathbf{w}_{+1,r}^{(t)}, \mathbf{v}\rangle > 0$, we immediately get that

$$\langle \mathbf{w}_{+1,r}^{(t+1)}, \mathbf{v}\rangle \ge \frac{\eta}{nm} \sum_{i \in S_{-\mathbf{u},+1} \cup S_{+\mathbf{u},-1}} \ell_i^{\prime(t)} \cdot \mathbf{1}\{\langle \mathbf{w}_{+1,r}^{(t)}, \boldsymbol{\mu}_i\rangle > 0\}\|\boldsymbol{\mu}\|_2^2 \cos\theta$$

$$+ \frac{\eta}{nm} \sum_{i \in S_{+\mathbf{v},-1} \cup S_{-\mathbf{v},+1}} \ell_i'^{(t)} \cdot \mathbf{1}\{\langle \mathbf{w}_{+1,r}^{(t)}, \boldsymbol{\mu}_i \rangle > 0\} \|\boldsymbol{\mu}\|_2^2$$

$$\geq -\eta \|\boldsymbol{\mu}\|_2^2/m,$$

where the second inequality is by $-\ell'^{(t)} \leq 1$ and Lemma C.2. As for the upper bound, if $\langle \mathbf{w}_{j,r}^{(t)}, \mathbf{u} \rangle < 0$, we have

$$\langle \mathbf{w}_{+1,r}^{(t+1)}, \mathbf{v} \rangle = \langle \mathbf{w}_{+1,r}^{(t)}, \mathbf{v} \rangle - \frac{\eta \|\boldsymbol{\mu}\|_2^2}{nm} \bigg( \sum_{i \in S_{-\mathbf{u},-1}} \ell_i'^{(t)} \cos\theta - \sum_{i \in S_{-\mathbf{u},+1}} \ell_i'^{(t)} \cos\theta \bigg)$$

$$- \frac{\eta \|\boldsymbol{\mu}\|_2^2}{nm} \bigg( - \sum_{i \in S_{+\mathbf{v},-1}} \ell_i'^{(t)} + \sum_{i \in S_{+\mathbf{v},+1}} \ell_i'^{(t)} \bigg)$$

$$\leq \langle \mathbf{w}_{+1,r}^{(t)}, \mathbf{v} \rangle,$$

where the inequality is by

$$\sum_{i \in S_{-\mathbf{u},-1}} \ell_i'^{(t)} \cos\theta - \sum_{i \in S_{-\mathbf{u},+1}} \ell_i'^{(t)} \cos\theta > 0,$$

$$- \sum_{i \in S_{+\mathbf{v},-1}} \ell_i'^{(t)} + \sum_{i \in S_{+\mathbf{v},+1}} \ell_i'^{(t)} > 0$$

from Lemma C.2 and Proposition D.8. If $\langle \mathbf{w}_{j,r}^{(t)}, \mathbf{u} \rangle > 0$, the update rule can be written as

$$\langle \mathbf{w}_{+1,r}^{(t+1)}, \mathbf{v} \rangle = \langle \mathbf{w}_{+1,r}^{(t)}, \mathbf{v} \rangle - \frac{\eta \|\boldsymbol{\mu}\|_2^2}{nm} \bigg( \sum_{i \in S_{+\mathbf{u},+1}} \ell_i'^{(t)} \cos\theta - \sum_{i \in S_{+\mathbf{u},-1}} \ell_i'^{(t)} \cos\theta \bigg)$$

$$- \frac{\eta \|\boldsymbol{\mu}\|_2^2}{nm} \bigg( - \sum_{i \in S_{+\mathbf{v},-1}} \ell_i'^{(t)} + \sum_{i \in S_{+\mathbf{v},+1}} \ell_i'^{(t)} \bigg)$$

$$= \langle \mathbf{w}_{+1,r}^{(t)}, \mathbf{v} \rangle - \frac{\eta \|\boldsymbol{\mu}\|_2^2 \sum_{i \in S_{+\mathbf{v},-1}} \ell_i'^{(t)}}{nm} \bigg( \frac{\sum_{i \in S_{+\mathbf{u},+1}} \ell_i'^{(t)}}{\sum_{i \in S_{+\mathbf{v},-1}} \ell_i'^{(t)}} \cdot \cos\theta - \frac{\sum_{i \in S_{+\mathbf{u},-1}} \ell_i'^{(t)}}{\sum_{i \in S_{+\mathbf{v},-1}} \ell_i'^{(t)}} \cdot \cos\theta \bigg)$$

$$- \frac{\eta \|\boldsymbol{\mu}\|_2^2 \sum_{i \in S_{+\mathbf{v},-1}} \ell_i'^{(t)}}{nm} \bigg( -1 + \frac{\sum_{i \in S_{+\mathbf{v},+1}} \ell_i'^{(t)}}{\sum_{i \in S_{+\mathbf{v},-1}} \ell_i'^{(t)}} \bigg)$$

$$\leq \langle \mathbf{w}_{+1,r}^{(t)}, \mathbf{v} \rangle$$

$$- \frac{\eta \|\boldsymbol{\mu}\|_2^2}{nm} \cdot \sum_{i \in S_{+\mathbf{v},-1}} \ell_i'^{(t)} \cdot \bigg( 2\cos\theta(1 + o(1)) - \frac{p\cos\theta(1 - o(1))}{2(1-p)(1+o(1))} - 1 + \frac{2p(1 + o(1))}{(1-p)(1 - o(1))} \bigg)$$

$$\leq \langle \mathbf{w}_{+1,r}^{(t)}, \mathbf{v} \rangle.$$

Here, the first inequality is by Proposition D.8, the second inequality is by the condition of $\theta$ and $p$ in Condition 3.1. We conclude that when $\langle \mathbf{w}_{+1,r}^{(t)}, \mathbf{v} \rangle > 0$, $|\langle \mathbf{w}_{+1,r}^{(t+1)}, \mathbf{v} \rangle| \leq |\langle \mathbf{w}_{+1,r}^{(0)}, \mathbf{v} \rangle| + \eta \|\boldsymbol{\mu}\|_2^2/m$. The proof for $\langle \mathbf{w}_{+1,r}^{(t)}, \mathbf{v} \rangle < 0$ is quite similar and we thus omit it. Similarly, we can also prove that $|\langle \mathbf{w}_{-1,r}^{(t)}, \mathbf{u} \rangle| \leq |\langle \mathbf{w}_{-1,r}^{(0)}, \mathbf{u} \rangle| + \eta \|\boldsymbol{\mu}\|_2^2/m$. As for the precise conclusions for upper bound, by the update rule, we can easily conclude from $1 + \cos\theta \leq 2$ that

$$\langle \mathbf{w}_{j,r}^{(t+1)}, \mathbf{u} \rangle \leq \langle \mathbf{w}_{j,r}^{(t)}, \mathbf{u} \rangle - \frac{\eta j}{nm} \sum_{i \in S_{+\mathbf{u},+1} \cup S_{-\mathbf{u},-1}} \ell_i'^{(t)} \cdot \mathbf{1}\{\langle \mathbf{w}_{j,r}^{(t)}, \boldsymbol{\mu}_i \rangle > 0\} \|\boldsymbol{\mu}\|_2^2$$

$$- \frac{\eta j}{nm} \sum_{i \in S_{-\mathbf{v},-1} \cup S_{+\mathbf{v},+1}} \ell_i'^{(t)} \cdot \mathbf{1}\{\langle \mathbf{w}_{j,r}^{(t)}, \boldsymbol{\mu}_i \rangle > 0\} \|\boldsymbol{\mu}\|_2^2 \cos\theta$$

$$\leq \langle \mathbf{w}_{+1,r}^{(t)}, \mathbf{u} \rangle - \frac{2\eta \|\boldsymbol{\mu}\|_2^2}{nm} \sum_{i \in [n]} \ell_i'^{(t)}.$$

Similarly, we can get the other conclusions. This completes the proof of Lemma E.1. ∎

In Lemma E.1, it becomes apparent that the direction of signal learning in XOR is determined by the initial state. For instance, if $j = +1$ and the signal vector is $\mathbf{u}$, the monotonicity in the value of inner product between the weight and $\mathbf{u}$ is solely dependent on the initialization, whether the inner product is less than or greater than 0. Conversely, for $\mathbf{u}$ and the weight vectors where $j = -1$, the inner product between them will be relatively small. We give the growth rate of signal learning in the next proposition.

**Proposition E.2 (Conclusion 2 in Proposition A.1)** *For* $\langle \mathbf{w}_{+1,r}^{(0)}, \mathbf{u} \rangle, \langle \mathbf{w}_{-1,r}^{(0)}, \mathbf{v} \rangle > 0$ *it holds that*

$$\langle \mathbf{w}_{+1,r}^{(t)}, \mathbf{u} \rangle, \langle \mathbf{w}_{-1,r}^{(t)}, \mathbf{v} \rangle$$
$$\geq -2\sqrt{\log(12m/\delta)} \cdot \sigma_0 \|\boldsymbol{\mu}\|_2 + \frac{cn\|\boldsymbol{\mu}\|_2^2}{\sigma_p^2 d} \cdot \log\left(\frac{\eta \sigma_p^2 d}{12nm}(t-1) + \frac{2}{3}\right) - \eta \cdot \|\boldsymbol{\mu}\|_2^2/m.$$

*For* $\langle \mathbf{w}_{+1,r}^{(0)}, \mathbf{u} \rangle, \langle \mathbf{w}_{-1,r}^{(0)}, \mathbf{v} \rangle < 0$, *it holds that*

$$\langle \mathbf{w}_{+1,r}^{(t)}, \mathbf{u} \rangle, \langle \mathbf{w}_{-1,r}^{(t)}, \mathbf{v} \rangle$$
$$\leq 2\sqrt{\log(12m/\delta)} \cdot \sigma_0 \|\boldsymbol{\mu}\|_2 - \frac{cn\|\boldsymbol{\mu}\|_2^2}{\sigma_p^2 d} \cdot \log\left(\frac{\eta \sigma_p^2 d}{12nm}(t-1) + \frac{2}{3}\right) + \eta \cdot \|\boldsymbol{\mu}\|_2^2/m.$$

*Here* $c > 0$ *is some absolute constant.*

**Proof** [Proof of Proposition E.2] We prove the first case, the other cases are all similar and we thus omit it. By Lemma E.1, when $\langle \mathbf{w}_{+1,r}^{(0)}, \mathbf{u} \rangle > 0$,

$$\langle \mathbf{w}_{+1,r}^{(t+1)}, \mathbf{u} \rangle \geq \langle \mathbf{w}_{+1,r}^{(t)}, \mathbf{u} \rangle - \frac{c\eta\|\boldsymbol{\mu}\|_2^2}{nm} \cdot \sum_{i \in S_{+\mathbf{u},+1}} \ell_i'^{(t)}$$

$$\geq \langle \mathbf{w}_{+1,r}^{(t)}, \mathbf{u} \rangle + \frac{c\eta\|\boldsymbol{\mu}\|_2^2}{nm} \cdot \sum_{i \in S_{+\mathbf{u},+1}} \frac{1}{1 + e^{\frac{1}{m}\sum_{r=1}^{m} \overline{\rho}_{y_i,r,i}^{(t)} + \kappa/2}}$$

$$\geq \langle \mathbf{w}_{+1,r}^{(t)}, \mathbf{u} \rangle + \frac{c\eta\|\boldsymbol{\mu}\|_2^2}{m} \frac{1}{1 + \overline{b}e^{\overline{x}_t}}$$

$$\geq \langle \mathbf{w}_{+1,r}^{(0)}, \mathbf{u} \rangle + \frac{c\eta\|\boldsymbol{\mu}\|_2^2}{m} \cdot \sum_{\tau=0}^{t} \frac{1}{1 + \overline{b}e^{\overline{x}_\tau}}$$

$$\geq \langle \mathbf{w}_{+1,r}^{(0)}, \mathbf{u} \rangle + \frac{c\eta\|\boldsymbol{\mu}\|_2^2}{m} \int_1^{t-1} \frac{1}{1 + \overline{b}e^{\overline{x}_\tau}} d\tau$$

$$\geq \langle \mathbf{w}_{+1,r}^{(0)}, \mathbf{u} \rangle + \frac{c\eta\|\boldsymbol{\mu}\|_2^2}{m} \int_1^{t-1} \frac{1}{\overline{c}} d\overline{x}_\tau$$

$$\geq \langle \mathbf{w}_{+1,r}^{(0)}, \mathbf{u} \rangle + \frac{cn\|\boldsymbol{\mu}\|_2^2}{\sigma_p^2 d} \overline{x}_{t-1} - \frac{c \cdot \overline{c}n\|\boldsymbol{\mu}\|_2^2}{\sigma_p^2 d}$$

$$\geq -\sqrt{2\log(12m/\delta)} \cdot \sigma_0\|\mathbf{u}\|_2 + \frac{cn\|\boldsymbol{\mu}\|_2^2}{\sigma_p^2 d} \cdot \overline{x}_{t-1} - \eta \cdot \|\boldsymbol{\mu}\|_2^2/m.$$

Note that the $c$ here is not equal, and we write $c$ here for the reason that all the $c$ are absolute constant. Here, the second inequality is by Lemma D.5, the third inequality is by $p \leq 1/C$ for sufficient large $C$ in Condition 3.1, $|S_{+\mathbf{u},+1}| \geq (1-p)n/4.5$ in Lemma C.2 and Proposition D.8 which gives the bound for the summation over $r$ of $\overline{\rho}_{y_i,r,i}$, the fifth inequality is by the definition of $\overline{x}_t$, the sixth inequality is by $\overline{x}_1 \leq 2\overline{c}$ and the last inequality is by the definition $\overline{c}$ in Proposition D.8 and Lemma C.4. By the definition of $\overline{x}_t$ in Proposition D.8 and results in Lemma D.9, we can easily see that

$$\langle \mathbf{w}_{+1,r}^{(t+1)}, \mathbf{u} \rangle \geq -\sqrt{2\log(12m/\delta)} \cdot \sigma_0\|\mathbf{u}\|_2 + \frac{cn\|\boldsymbol{\mu}\|_2^2}{\sigma_p^2 d} \cdot \log\left(\frac{\eta\sigma_p^2 d}{12nm}(t-1) + \frac{2}{3}\right) - \eta \cdot \|\boldsymbol{\mu}\|_2^2/m$$

The proof of Proposition E.2 completes. ∎

**Proposition E.3** *For $\langle \mathbf{w}^{(0)}_{+1,r}, \mathbf{u} \rangle, \langle \mathbf{w}^{(0)}_{-1,r}, \mathbf{v} \rangle > 0$ it holds that*

$$\langle \mathbf{w}^{(t)}_{+1,r}, \mathbf{u} \rangle, \langle \mathbf{w}^{(t)}_{-1,r}, \mathbf{v} \rangle \le 2\sqrt{\log(12m/\delta)} \cdot \sigma_0 \|\boldsymbol{\mu}\|_2 + \frac{Cn\|\boldsymbol{\mu}\|_2^2}{\sigma_p^2 d} \cdot \log\left(\frac{2\eta\sigma_p^2 d}{nm}t + 1\right).$$

*For $\langle \mathbf{w}^{(0)}_{+1,r}, \mathbf{u} \rangle, \langle \mathbf{w}^{(0)}_{-1,r}, \mathbf{v} \rangle < 0$ it holds that*

$$\langle \mathbf{w}^{(t)}_{+1,r}, \mathbf{u} \rangle, \langle \mathbf{w}^{(t)}_{-1,r}, \mathbf{v} \rangle \ge -2\sqrt{\log(12m/\delta)} \cdot \sigma_0 \|\boldsymbol{\mu}\|_2 - \frac{Cn\|\boldsymbol{\mu}\|_2^2}{\sigma_p^2 d} \cdot \log\left(\frac{2\eta\sigma_p^2 d}{nm}t + 1\right).$$

**Proof** [Proof of Proposition E.3] We prove the case when $\langle \mathbf{w}^{(0)}_{+1,r}, \mathbf{u} \rangle > 0$, the other cases are all similar and we thus omit it. By Lemma E.1, when $\langle \mathbf{w}^{(0)}_{+1,r}, \mathbf{u} \rangle > 0$,

$$
\begin{aligned}
\langle \mathbf{w}^{(t+1)}_{+1,r}, \mathbf{u} \rangle &\le \langle \mathbf{w}^{(t)}_{+1,r}, \mathbf{u} \rangle - \frac{2\eta\|\boldsymbol{\mu}\|_2^2}{nm} \cdot \sum_{i \in [n]} \ell_i'^{(t)} \\
&\le \langle \mathbf{w}^{(t)}_{+1,r}, \mathbf{u} \rangle + \frac{2\eta\|\boldsymbol{\mu}\|_2^2}{nm} \cdot \sum_{i \in [n]} \frac{1}{1 + e^{\frac{1}{m}\sum_{r=1}^m \overline{\rho}_{y_i,r,i} - \kappa/2}} \\
&\le \langle \mathbf{w}^{(0)}_{+1,r}, \mathbf{u} \rangle + \frac{2\eta\|\boldsymbol{\mu}\|_2^2}{m} \cdot \sum_{\tau=0}^{t} \frac{1}{1 + \underline{b}e^{\underline{x}_\tau}} \\
&\le \langle \mathbf{w}^{(0)}_{+1,r}, \mathbf{u} \rangle + \frac{2\eta\|\boldsymbol{\mu}\|_2^2}{m} \int_0^t \frac{1}{1 + \underline{b}e^{\underline{x}_\tau}} d\tau \\
&\le \langle \mathbf{w}^{(0)}_{+1,r}, \mathbf{u} \rangle + \frac{2\eta\|\boldsymbol{\mu}\|_2^2}{m} \int_0^t \frac{1}{\underline{c}} d\underline{x}_\tau \\
&\le \sqrt{2\log(12m/\delta)} \cdot \sigma_0 \|\boldsymbol{\mu}\|_2 + \frac{Cn\|\boldsymbol{\mu}\|_2^2}{\sigma_p^2 d} \cdot \underline{x}_t.
\end{aligned}
$$

Here, the second inequality is by the output bound in Lemma D.5, the third inequality is by Proposition D.8, the fifth inequality is by the definition of $\underline{x}_t$, and the last inequality is by the definition of $\underline{c}$ in Proposition D.8 and Lemma C.4. By the results in Lemma D.9, we have

$$\underline{x}_t \le \log\left(\frac{2\eta\sigma_p^2 d}{nm}t + 1\right).$$

The proof of Proposition E.3 completes. ∎

### E.2 Noise Memorization

In this section, we give the analysis of noise memorization.

**Proposition E.4** *Let $\overline{c}$ and $\overline{x}_t$ be defined in Proposition D.8, then it holds that*

$$3n\underline{x}_t \ge \sum_{i=1}^n \overline{\rho}_{j,r,i}^{(t)} \ge \frac{n}{5} \cdot (\overline{x}_{t-1} - \overline{x}_1)$$

*for all $t \in [T^*]$ and $r \in [m]$.*

**Proof** [Proof of Proposition E.4] Recall the update rule for $\overline{\rho}_{j,r,i}^{(t+1)}$, that we have

$$\overline{\rho}_{j,r,i}^{(t+1)} = \overline{\rho}_{j,r,i}^{(t)} - \frac{\eta}{nm} \cdot \ell_i'^{(t)} \cdot \mathbf{1}\left(\langle \mathbf{w}_{j,r}^{(t)}, \boldsymbol{\xi}_i \rangle \ge 0\right) \cdot \mathbf{1}\left(y_i = j\right) \|\boldsymbol{\xi}_i\|_2^2.$$

Hence

$$\sum_i \overline{\rho}_{y_i,r,i}^{(t)} \ge \sum_i \overline{\rho}_{y_i,r,i}^{(t-1)} + \frac{\eta}{nm} \cdot \frac{1}{1 + \overline{b}\overline{x}_{t-1}} \cdot |S_{j,r}^{(0)}| \cdot \|\boldsymbol{\xi}_i\|_2^2$$

$$
\begin{aligned}
&= \sum_{\tau=1}^{t-1} \frac{\eta}{nm} \cdot \frac{1}{1 + \overline{b}\overline{x}_{\tau-1}} \cdot |S_{j,r}^{(0)}| \cdot \|\boldsymbol{\xi}_i\|_2^2 \\
&\geq \frac{\eta |S_{j,r}^{(0)}| \cdot \|\boldsymbol{\xi}_i\|_2^2}{nm} \int_2^t \frac{1}{1 + \overline{b}\overline{x}_{\tau-1}} d\tau = \frac{\eta |S_{j,r}^{(0)}| \cdot \|\boldsymbol{\xi}_i\|_2^2}{\overline{c} nm} \cdot (\overline{x}_{t-1} - \overline{x}_1) \\
&\geq \frac{n}{5} \cdot (\overline{x}_{t-1} - \overline{x}_1).
\end{aligned}
$$

Here, the first equality is by $|S_{j,r}^{(t)}| = |S_{j,r}^{(0)}|$, the first inequality is by Proposition D.8 and the last inequality is by the definition of $\overline{c}$. Similarly, we have

$$
\begin{aligned}
\sum_i \overline{\rho}_{y_i,r,i}^{(t)} &\leq \sum_i \overline{\rho}_{y_i,r,i}^{(t-1)} + \frac{\eta}{nm} \cdot \frac{1}{1 + \underline{b} \cdot \underline{x}_{t-1}} \cdot m \cdot \|\boldsymbol{\xi}_i\|_2^2 \\
&\leq \frac{\eta m \cdot \|\boldsymbol{\xi}_i\|_2^2}{nm} \int_2^t \frac{1}{1 + \underline{b} \cdot \underline{x}_{\tau-1}} d\tau \\
&\leq 3n\underline{x}_t.
\end{aligned}
$$

This completes the proof of Proposition E.4. ■

The next proposition gives the $L_2$ norm of $\mathbf{w}_{j,r}^{(t)}$.

**Proposition E.5 (Conclusion 3 in Proposition A.1)** *Under Condition 3.1, for $t = \Omega(nm/(\eta\sigma_p^2 d)$ it holds that*

$$
\Theta\left(\sigma_p^{-1} d^{-1/2} n^{1/2}\right) \cdot \log\left(\frac{\eta\sigma_p^2 d}{12nm}(t-1) + \frac{2}{3}\right) \leq \|\mathbf{w}_{j,r}^{(t)}\|_2 \leq \Theta\left(\sigma_p^{-1} d^{-1/2} n^{1/2}\right) \cdot \log\left(\frac{2\eta\sigma_p^2 d}{nm} t + 1\right).
$$

**Proof** [Proof of Proposition E.5] We first have the following inequalities:

$$
\frac{\left|\langle \mathbf{w}_{j,r}^{(t)}, \mathbf{u} \rangle\right| \cdot \|\boldsymbol{\mu}\|_2^{-1}}{\Theta\left(\sigma_p^{-1} d^{-1/2} n^{-1/2}\right) \cdot \sum_{i=1}^n \overline{\rho}_{j,r,i}^{(t)}} = \Theta\left(\sigma_p d^{1/2} n^{1/2} \|\boldsymbol{\mu}\|_2^{-1} \mathrm{SNR}^2\right) = \Theta\left(\sigma_p^{-1} d^{-1/2} n^{1/2} \|\mathbf{u}\|_2\right) = o(1),
$$

based on the coefficient order $\sum_{i=1}^n \overline{\rho}_{j,r,i}^{(t)} / |\langle \mathbf{w}_{j,r}^{(t)}, \mathbf{u} \rangle| = \Theta\left(\mathrm{SNR}^{-2}\right)$, the definition $\mathrm{SNR} = \|\boldsymbol{\mu}\|_2 / (\sigma_p \sqrt{d})$ and the condition for $d$ in Condition 3.1; and also

$$
\frac{\left\|\mathbf{w}_{j,r}^{(0)}\right\|_2}{\Theta\left(\sigma_p^{-1} d^{-1/2} n^{-1/2}\right) \cdot \sum_{i=1}^n \overline{\rho}_{j,r,i}^{(t)}} = \frac{\Theta\left(\sigma_0 \sqrt{d}\right)}{\Theta\left(\sigma_p^{-1} d^{-1/2} n^{-1/2}\right) \cdot \sum_{i=1}^n \overline{\rho}_{j,r,i}^{(t)}} = O\left(\sigma_0 \sigma_p d n^{-1/2}\right) = o(1).
$$

Here, we use Proposition E.4 that when $t = \Omega(nm/(\eta\sigma_p^2 d)$, $\overline{x}_t \geq C > 0$ for some constant $C$, and hence

$$
\sum_{i=1}^n \overline{\rho}_{j,r,i}^{(t)} \geq \frac{n}{5} \cdot (\overline{x}_{t-1} - \overline{x}_1) \geq \Omega(n).
$$

We also have the following estimation for the norm $\|\sum_{i=1}^n \rho_{j,r,i}^{(t)} \cdot \|\boldsymbol{\xi}_i\|_2^{-2} \cdot \boldsymbol{\xi}_i\|_2^2$:

$$
\begin{aligned}
&\left\|\sum_{i=1}^n \rho_{j,r,i}^{(t)} \cdot \|\boldsymbol{\xi}_i\|_2^{-2} \cdot \boldsymbol{\xi}_i\right\|_2^2 \\
&= \sum_{i=1}^n \rho_{j,r,i}^{(t)} \cdot \|\boldsymbol{\xi}_i\|_2^{-2} + 2 \sum_{1 \leq i_1 < i_2 \leq n} \rho_{j,r,i_1}^{(t)} \rho_{j,r,i_2}^{(t)} \cdot \|\boldsymbol{\xi}_{i_1}\|_2^{-2} \cdot \|\boldsymbol{\xi}_{i_2}\|_2^{-2} \cdot \langle \boldsymbol{\xi}_{i_1}, \boldsymbol{\xi}_{i_2} \rangle \\
&= 4\Theta\left(\sigma_p^{-2} d^{-1} \sum_{i=1}^n \rho_{j,r,i}^{(t)}{}^2 + 2 \sum_{1 \leq i_1 < i_2 \leq n} \rho_{j,r,i_1}^{(t)} \rho_{j,r,i_2}^{(t)} \cdot \left(16\sigma_p^{-4} d^{-2}\right) \cdot \left(2\sigma_p^2 \sqrt{d \log\left(6n^2/\delta\right)}\right)\right) \\
&= \Theta\left(\sigma_p^{-2} d^{-1}\right) \sum_{i=1}^n \rho_{j,r,i}^{(t)} + \widetilde{\Theta}\left(\sigma_p^{-2} d^{-3/2}\right) \left(\sum_{i=1}^n \rho_{j,r,i}^{(t)}\right)^2 \\
&= \Theta\left(\sigma_p^{-2} d^{-1} n^{-1}\right) \left(\sum_{i=1}^n \overline{\rho}_{j,r,i}^{(t)}\right)^2,
\end{aligned}
$$

where the first quality is by Lemma C.3; for the second to last equation we plugged in coefficient orders. We can thus upper bound the norm of $\mathbf{w}_{j,r}^{(t)}$ as:

$$\left\|\mathbf{w}_{j,r}^{(t)}\right\|_2 \le \left\|\mathbf{w}_{j,r}^{(0)}\right\|_2 + \left|\langle \mathbf{w}_{j,r}^{(t)} - \mathbf{w}_{j,r}^{(0)}, \mathbf{u}\rangle\right| \cdot \|\mathbf{u}\|_2^{-1} + \left|\langle \mathbf{w}_{j,r}^{(t)} - \mathbf{w}_{j,r}^{(0)}, \mathbf{v}\rangle\right| \cdot \|\mathbf{v}\|_2^{-1} + \left\|\sum_{i=1}^{n} \rho_{j,r,i}^{(t)} \cdot \|\boldsymbol{\xi}_i\|_2^{-2} \cdot \boldsymbol{\xi}_i\right\|_2$$

$$\le 4\left\|\mathbf{w}_{j,r}^{(0)}\right\|_2 + \left|\langle \mathbf{w}_{j,r}^{(t)}, \mathbf{u}\rangle\right| \cdot \|\mathbf{u}\|_2^{-1} + \left|\langle \mathbf{w}_{j,r}^{(t)}, \mathbf{v}\rangle\right| \cdot \|\mathbf{v}\|_2^{-1} + \Theta\left(\sigma_p^{-1} d^{-1/2} n^{-1/2}\right) \cdot \sum_{i=1}^{n} \overline{\rho}_{j,r,i}^{(t)}$$

$$= \Theta\left(\sigma_p^{-1} d^{-1/2} n^{-1/2}\right) \cdot \sum_{i=1}^{n} \overline{\rho}_{j,r,i}^{(t)}. \tag{E.2}$$

Here, the first inequality is due to triangle inequality and the second inequality is due to $\left|\langle \mathbf{w}_{j,r}^{(0)}, \mathbf{u}\rangle / \|\boldsymbol{\mu}\|_2\right| \le \|\mathbf{w}_{j,r}^{(0)}\|_2$. equation E.2 holds from the above analysis. Moreover, we also have that

$$\|\mathbf{w}_{j,r}^{(t)}\|_2 = \Theta\left(\sigma_p^{-1} d^{-1/2} n^{-1/2}\right) \cdot \sum_{i=1}^{n} \overline{\rho}_{j,r,i}^{(t)} \cdot (1 - o(1)) = \Theta\left(\sigma_p^{-1} d^{-1/2} n^{-1/2}\right) \cdot \sum_{i=1}^{n} \overline{\rho}_{j,r,i}^{(t)}.$$

Combined the equation above with Proposition E.4 and Lemma D.9 completes the proof of Proposition E.5. ∎

## F  PROOF OF THEOREM 3.2

We prove Theorem 3.2 in this section. We first prove a slightly different version as follows:

**Theorem F.1** *For any $\varepsilon, \delta > 0$, if Condition 3.1 holds, then there exist constants $C_1, C_2, C_3, C_4 > 0$, such that with probability at least $1 - 2\delta$, the following results hold at $T = \Omega(nm/(\eta\varepsilon\sigma_p^2 d))$:*

1. *The training loss converges below $\varepsilon$, i.e., $L(\mathbf{W}^{(T)}) \le \varepsilon$.*
2. *If $n\|\boldsymbol{\mu}\|_2^4 \ge C_1\sigma_p^4 d$, then the CNN trained by gradient descent can achieve near Bayes-optimal test error: $R(\mathbf{W}^{(T)}) \le p + \exp(-C_2 n\|\boldsymbol{\mu}\|_2^4/(\sigma_p^4 d))$.*
3. *If $n\|\boldsymbol{\mu}\|_2^4 \le C_3\sigma_p^4 d$, then the CNN trained by gradient descent can only achieve sub-optimal error rate: $R(\mathbf{W}^{(T)}) \ge p + C_4$.*

Clearly, Theorem F.1 is almost the same as Theorem 3.2 except some differences in absolute constants. Below we first prove this version, and then show that Theorem F.1 can easily lead to Theorem 3.2.

**Proof** [Proof of Theorem F.1] First of all, we prove the convergence of training. For the training convergence, Lemma D.5 and Proposition D.8 show that

$$y_i f(\mathbf{W}^{(t)}, \mathbf{x}_i) \ge -\frac{\kappa}{2} + \frac{1}{m}\sum_{r=1}^{m} \overline{\rho}_{y_i,r,i}^{(t)}$$

$$\ge -\frac{\kappa}{2} + \underline{x}_t$$

$$\ge -\frac{\kappa}{2} + \log\left(\frac{\eta\sigma_p^2 d}{12nm}t + \frac{2}{3}\right)$$

$$\ge -\kappa + \log\left(\frac{\eta\sigma_p^2 d}{12nm}t + \frac{2}{3}\right).$$

$\kappa$ is defined in equation D.4. Here, the first inequality is by the conclusion in Lemma D.5 and the second inequality is by Proposition D.8, and third inequality are by Lemma D.9, the last inequality is by the definition of $\kappa$ in equation D.4. Therefore we have

$$L(\mathbf{W}^{(t)}) \le \log\left(1 + \exp\{\kappa\}/\left(\frac{\eta\sigma_p^2 d}{12nm}t + \frac{2}{3}\right)\right) \le \frac{e^\kappa}{\frac{\eta\sigma_p^2 d}{12nm}t + \frac{2}{3}}.$$

The last inequality is by $\log(1+x) \le x$. When $t \ge \Omega(\frac{nm}{\eta \sigma_p^2 d \varepsilon})$, we conclude that

$$L(\mathbf{W}^{(t)}) \le \frac{e^\kappa}{\frac{\eta \sigma_p^2 d}{12nm}t + \frac{2}{3}} \le \frac{e^\kappa}{2/\varepsilon + \frac{2}{3}} \le \varepsilon.$$

Here, the last inequality is by $e^\kappa \le 1.5$. This completes the proof for the convergence of training loss.

As for the second conclusion, it is easy to see that

$$P(yf(\mathbf{W}^{(t)}, \mathbf{x}) > 0) = \sum_{\boldsymbol{\mu} \in \{\pm\mathbf{u}, \pm\mathbf{v}\}} P(yf(\mathbf{W}^{(t)}, \mathbf{x}) > 0 | \mathbf{x}_{\text{signal part}} = \boldsymbol{\mu}) \cdot \frac{1}{4}, \tag{F.1}$$

without loss of generality we can assume that the test data $\mathbf{x} = (\mathbf{u}^\top, \boldsymbol{\xi}^\top)^\top$, for $\mathbf{x} = (-\mathbf{u}^\top, \boldsymbol{\xi}^\top)^\top$, $\mathbf{x} = (\mathbf{v}^\top, \boldsymbol{\xi}^\top)^\top$ and $\mathbf{x} = (-\mathbf{v}^\top, \boldsymbol{\xi}^\top)^\top$ the proof is all similar and we omit it. We investigate

$$P(yf(\mathbf{W}^{(t)}, \mathbf{x}) > 0 | \mathbf{x}_{\text{signal part}} = \mathbf{u}).$$

When $\mathbf{x} = (\mathbf{u}^\top, \boldsymbol{\xi}^\top)^\top$, the true label $\overline{y} = +1$. We remind that the true label for $\mathbf{x}$ is $\overline{y}$, and the observed label is $y$. Therefore we have

$$P(yf(\mathbf{W}^{(t)}, \mathbf{x}) < 0 | \mathbf{x}_{\text{signal part}} = \mathbf{u}) = P(\overline{y}f(\mathbf{W}^{(t)}, \mathbf{x}) < 0, \overline{y} = y | \mathbf{x}_{\text{signal part}} = \mathbf{u})$$
$$+ P(\overline{y}f(\mathbf{W}^{(t)}, \mathbf{x}) > 0, \overline{y} \ne y | \mathbf{x}_{\text{signal part}} = \mathbf{u})$$
$$\le p + P(\overline{y}f(\mathbf{W}^{(t)}, \mathbf{x}) < 0 | \mathbf{x}_{\text{signal part}} = \mathbf{u}).$$

It therefore suffices to provide an upper bound for $P(\overline{y}f(\mathbf{W}^{(t)}, \mathbf{x}) < 0 | \mathbf{x}_{\text{signal part}} = \mathbf{u})$. Note that for any test data with the conditioned event $\mathbf{x}_{\text{signal part}} = \mathbf{u}$, it holds that

$$\overline{y}f(\mathbf{W}^{(t)}, \mathbf{x}) = \frac{1}{m}\sum_{r=1}^m F_{+1,r}(\mathbf{W}^{(t)}, \mathbf{u}) + F_{+1,r}(\mathbf{W}^{(t)}, \boldsymbol{\xi}) - \frac{1}{m}\sum_{r=1}^m \left(F_{-1,r}(\mathbf{W}^{(t)}, \mathbf{u}) + F_{-1,r}(\mathbf{W}^{(t)}, \boldsymbol{\xi})\right).$$

We have for $t \ge \Omega(nm/(\eta\sigma_p^2 d))$, $\overline{x}_t \ge C > 0$, and

$$\overline{y}f(\mathbf{W}^{(t)}, \mathbf{x}) \ge \frac{1}{m}\sum_{r=1}^m \text{ReLU}(\langle \mathbf{w}_{+1,r}^{(t)}, \mathbf{u}\rangle) - \frac{1}{m}\sum_{r=1}^m \text{ReLU}(\langle \mathbf{w}_{-1,r}^{(t)}, \boldsymbol{\xi}\rangle) - \frac{1}{m}\sum_{r=1}^m \text{ReLU}(\langle \mathbf{w}_{-1,r}^{(t)}, \mathbf{u}\rangle)$$

$$\ge -2\sqrt{\log(12m/\delta)} \cdot \sigma_0 \|\boldsymbol{\mu}\|_2 + \frac{cn\|\boldsymbol{\mu}\|_2^2}{\sigma_p^2 d} \cdot \overline{x}_{t-1} - \frac{\eta\|\boldsymbol{\mu}\|_2^2}{m}$$

$$- \frac{1}{m}\sum_{r=1}^m \text{ReLU}(\langle \mathbf{w}_{-1,r}^{(t)}, \boldsymbol{\xi}\rangle) - \frac{1}{m}\sum_{r=1}^m \text{ReLU}(\langle \mathbf{w}_{-1,r}^{(t)}, \mathbf{u}\rangle),$$

where the first inequality is by $F_{y,r}(\mathbf{W}^{(t)}, \boldsymbol{\xi}) \ge 0$, and the second inequality is by Proposition E.2 and $|\{r \in [m], \langle \mathbf{w}_{+1,r}^{(0)}, \mathbf{u}\rangle > 0\}|/m \ge 1/3$. Then for $t \ge T = \Omega(nm/(\eta\sigma_p^2 d))$, $\overline{x}_t \ge \underline{x}_t \ge C > 0$, it holds that

$$\overline{y}f(\mathbf{W}^{(t)}, \mathbf{x}) \ge \frac{cn\|\boldsymbol{\mu}\|_2^2}{\sigma_p^2 d} \cdot \overline{x}_{t-1} - \frac{1}{m}\sum_{r=1}^m \text{ReLU}(\langle \mathbf{w}_{-1,r}^{(t)}, \boldsymbol{\xi}\rangle) - \frac{1}{m}\sum_{r=1}^m \text{ReLU}(\langle \mathbf{w}_{-1,r}^{(t)}, \mathbf{u}\rangle)$$

$$- 2\sqrt{\log(12mn/\delta)} \cdot \sigma_0 \|\boldsymbol{\mu}\|_2$$

$$\ge \frac{cn\|\boldsymbol{\mu}\|_2^2}{\sigma_p^2 d} \cdot \overline{x}_{t-1} - \frac{1}{m}\sum_{r=1}^m \text{ReLU}(\langle \mathbf{w}_{-1,r}^{(t)}, \boldsymbol{\xi}\rangle) - 4\sqrt{\log(12mn/\delta)} \cdot \sigma_0 \|\boldsymbol{\mu}\|_2 - 2\eta\|\boldsymbol{\mu}\|_2^2/m$$

$$\ge \frac{cn\|\boldsymbol{\mu}\|_2^2}{\sigma_p^2 d} \cdot \overline{x}_{t-1} - \frac{1}{m}\sum_{r=1}^m \text{ReLU}(\langle \mathbf{w}_{-1,r}^{(t)}, \boldsymbol{\xi}\rangle). \tag{F.2}$$

Here, the first inequality is by the condition of $\sigma_0$, $\eta$ in Condition 3.1, and $\overline{x}_{t-1} \ge C$, the second inequality is by the third conclusion in Lemma E.1 and the third inequality is still by the condition

of $\sigma_0$ , $\eta$ in Condition 3.1. We denote by $h(\boldsymbol{\xi}) = \frac{1}{m}\sum_{r=1}^{m} \text{ReLU}(\langle \mathbf{w}_{-1,r}^{(t)}, \boldsymbol{\xi}\rangle)$. By Theorem 5.2.2 in Vershynin (2018), we have

$$P(h(\boldsymbol{\xi}) - \mathbb{E}h(\boldsymbol{\xi}) \geq x) \leq \exp\left(-\frac{cx^2}{\sigma_p^2 \|h\|_{\text{Lip}}^2}\right). \tag{F.3}$$

By $n\|\boldsymbol{\mu}\|_2^4/(\sigma_p^4 d) \geq C_1$ for some sufficient large $C_1$ and Proposition E.5, we directly have

$$\frac{Cn\|\boldsymbol{\mu}\|_2^2}{\sigma_p^2 d}\cdot \overline{x}_{t-1} \geq \mathbb{E}h(\boldsymbol{\xi}) = \frac{\sigma_p}{\sqrt{2\pi}m}\sum_{r=1}^{m}\|\mathbf{w}_{-\widehat{y},r}^{(t)}\|_2.$$

Now using methods in equation F.3 we get that

$$P(\overline{y}f(\mathbf{W}^{(t)},\mathbf{x}) < 0 | \mathbf{x}_{\text{signal part}} = \mathbf{u}) \leq P\left(h(\boldsymbol{\xi}) - \mathbb{E}h(\boldsymbol{\xi}) \geq \sum_{r}\sigma(\langle \mathbf{w}_{+1,r}^{(t)}, \mathbf{u}\rangle) - \frac{\sigma_p}{\sqrt{2\pi}}\sum_{r=1}^{m}\|\mathbf{w}_{-1,r}^{(t)}\|_2\right)$$

$$\leq \exp\left[-\frac{c\left(\sum_r \sigma(\langle \mathbf{w}_{+1,r}^{(t)}, \mathbf{u}\rangle) - (\sigma_p/\sqrt{2\pi})\sum_{r=1}^{m}\|\mathbf{w}_{-1,r}^{(t)}\|_2\right)^2}{\sigma_p^2\left(\sum_{r=1}^{m}\|\mathbf{w}_{-1,r}^{(t)}\|_2\right)^2}\right]$$

$$\leq \exp\{-C_2 n\|\boldsymbol{\mu}\|_2^4/(\sigma_p^4 d)\}.$$

Here, $C_2 = O(1)$ is some constant. The first inequality is directly by equation F.2, the second inequality is by equation F.3 and the last inequality is by Proposition E.2 which directly gives the lower bound of signal learning and Proposition E.5 which directly gives the scale of $\|\mathbf{w}_{-1,r}^{(t)}\|_2$. We can similarly get the inequality on the condition $\mathbf{x}_{\text{signal part}} = -\mathbf{u}$ and $\mathbf{x}_{\text{signal part}} = \pm\mathbf{v}$. Combined the results with equation F.1, we have

$$P(yf(\mathbf{W}^{(t)},\mathbf{x}) < 0) \leq p + \exp\{-C_2 n\|\boldsymbol{\mu}\|_2^4/(\sigma_p^4 d)\}$$

for all $t \geq \Omega(nm/(\eta\sigma_p^2 d))$. Here, constant $C$ in different inequalities is different, and the inequality above is by $n\|\boldsymbol{\mu}\|_2^4(1 - \cos\theta)^2/(\sigma_p^4 d) \geq C_1$ for some constant $C_1$.

For the proof of the third conclusion in Theorem F.1, we have

$$P(yf(\mathbf{W}^{(t)},\mathbf{x}) < 0) = P(\overline{y}f(\mathbf{W}^{(t)},\mathbf{x}) < 0, \overline{y} = y) + P(\overline{y}f(\mathbf{W}^{(t)},\mathbf{x}) > 0, \overline{y} \neq y)$$

$$= p + (1 - 2p)P(\overline{y}f(\mathbf{W}^{(t)},\mathbf{x}) \leq 0). \tag{F.4}$$

We investigate the probability $P(\overline{y}f(\mathbf{W}^{(t)},\mathbf{x}) \leq 0)$, and have

$$P(\overline{y}f(\boldsymbol{W}^{(t)},\mathbf{x}) \leq 0)$$
$$= P\left(\sum_r \sigma(\langle \mathbf{w}_{-\overline{y},r}^{(t)}, \boldsymbol{\xi}\rangle) - \sum_r \sigma(\langle \mathbf{w}_{\overline{y},r}^{(t)}, \boldsymbol{\xi}\rangle) \geq \sum_r \sigma(\langle \mathbf{w}_{\overline{y},r}^{(t)}, \boldsymbol{\mu}\rangle) - \sum_r \sigma(\langle \mathbf{w}_{-\overline{y},r}^{(t)}, \boldsymbol{\mu}\rangle)\right).$$

Here, $\boldsymbol{\mu}$ is the signal part of the test data. Define $g(\boldsymbol{\xi}) = \left(\frac{1}{m}\sum_{r=1}^{m}\text{ReLU}(\langle \mathbf{w}_{+1,r}^{(t)}, \boldsymbol{\xi}\rangle) - \frac{1}{m}\sum_{r=1}^{m}\text{ReLU}(\langle \mathbf{w}_{-1,r}^{(t)}, \boldsymbol{\xi}\rangle)\right)$, it is easy to see that

$$P(\overline{y}f(\boldsymbol{W}^{(t)},\mathbf{x}) \leq 0) \geq 0.5P\left(|g(\boldsymbol{\xi})| \geq \max\left\{\sum_r \sigma(\langle \mathbf{w}_{\overline{y},r}^{(t)}, \boldsymbol{\mu}\rangle)/m, \sum_r \sigma(\langle \mathbf{w}_{-\overline{y},r}^{(t)}, \boldsymbol{\mu}\rangle)/m\right\}\right), \tag{F.5}$$

since if $|g(\boldsymbol{\xi})|$ is large, we can always select $\overline{y}$ given $\boldsymbol{\xi}$ to make a wrong prediction. Define the set

$$\boldsymbol{\Omega} = \left\{\boldsymbol{\xi} : |g(\boldsymbol{\xi})| \geq \max\left\{\sum_r \sigma(\langle \mathbf{w}_{+1,r}^{(t)}, \boldsymbol{\mu}\rangle)/m, \sum_r \sigma(\langle \mathbf{w}_{-1,r}^{(t)}, \boldsymbol{\mu}\rangle)/m\right\}\right\},$$

it remains for us to proceed $P(\boldsymbol{\Omega})$. To further proceed, we prove that there exists a fixed vector $\boldsymbol{\zeta}$ with $\|\boldsymbol{\zeta}\|_2 \leq 0.02\sigma_p$, such that

$$\sum_{j'\in\{\pm 1\}}\left[g(j'\boldsymbol{\xi} + \boldsymbol{\zeta}) - g(j'\boldsymbol{\xi})\right] \geq 4\max\left\{\sum_r \sigma(\langle \mathbf{w}_{+1,r}^{(t)}, \boldsymbol{\mu}\rangle)/m, \sum_r \sigma(\langle \mathbf{w}_{-1,r}^{(t)}, \boldsymbol{\mu}\rangle)/m\right\}.$$

Here, $\boldsymbol{\mu} \in \{\pm\mathbf{u}, \pm\mathbf{v}\}$ is the signal part of test data. If so, we can see that there must exist at least one of $\boldsymbol{\xi}, \boldsymbol{\xi} + \boldsymbol{\zeta}, -\boldsymbol{\xi} + \boldsymbol{\zeta}$ and $-\boldsymbol{\xi}$ that belongs to $\boldsymbol{\Omega}$. We immediately get that

$$P\big(\boldsymbol{\Omega} \cup (-\boldsymbol{\Omega}) \cup (\boldsymbol{\Omega} - \boldsymbol{\zeta}) \cup (-\boldsymbol{\Omega} - \boldsymbol{\zeta})\big) = 1.$$

Then we can see that there exists at least one of $\boldsymbol{\Omega}, -\boldsymbol{\Omega}, \boldsymbol{\Omega} - \boldsymbol{\zeta}, -\boldsymbol{\Omega} - \boldsymbol{\zeta}$ such that the probability is larger than 0.25. Also note that $\|\boldsymbol{\zeta}\|_2 \leq 0.02\sigma_p$, we have

$$|P(\boldsymbol{\Omega}) - P(\boldsymbol{\Omega} - \boldsymbol{\zeta})| = \big|P_{\boldsymbol{\xi}\sim\mathcal{N}(0,\sigma_p^2\mathbf{I}_d)}(\boldsymbol{\xi} \in \boldsymbol{\Omega}) - P_{\boldsymbol{\xi}\sim\mathcal{N}(\boldsymbol{\zeta},\sigma_p^2\mathbf{I}_d)}(\boldsymbol{\xi} \in \boldsymbol{\Omega})\big|$$

$$\leq \frac{\|\boldsymbol{\zeta}\|_2}{2\sigma_p} \leq 0.01.$$

Here, the first inequality is by Proposition 2.1 in Devroye et al. (2018) and the second inequality is by the condition $\|\boldsymbol{\zeta}\|_2 \leq 0.02\sigma_p$. Combined with $P(\boldsymbol{\Omega}) = P(-\boldsymbol{\Omega})$, we conclude that $P(\boldsymbol{\Omega}) \geq 0.24$. We can obtain from equation F.4 that

$$P(yf(\mathbf{W}^{(t)}, \mathbf{x}) < 0) = p + (1 - 2p)P\big(\overline{y}f(\mathbf{W}^{(t)}, \mathbf{x}) \leq 0\big)$$

$$\geq p + (1 - 2p) * 0.12 \geq 0.76p + 0.12 \geq p + C_4$$

if we find the existence of $\boldsymbol{\zeta}$. Here, the last inequality is by $p < 1/2$ is a fixed value.

The only thing remained for us is to prove the existence of $\boldsymbol{\zeta}$ with $\|\boldsymbol{\zeta}\|_2 \leq 0.02\sigma_p$ such that

$$\sum_{j'\in\{\pm1\}} \big[g\big(j'\boldsymbol{\xi} + \boldsymbol{\zeta}\big) - g\big(j'\boldsymbol{\xi}\big)\big] \geq 4\max\bigg\{\sum_r \sigma(\langle\mathbf{w}_{+1,r}^{(t)}, \boldsymbol{\mu}\rangle)/m, \sum_r \sigma(\langle\mathbf{w}_{-1,r}^{(t)}, \boldsymbol{\mu}\rangle)/m\bigg\}.$$

The existence is proved by the following construction:

$$\boldsymbol{\zeta} = \lambda \cdot \sum_i \mathbf{1}(y_i = 1)\boldsymbol{\xi}_i,$$

where $\lambda = C_4\|\boldsymbol{\mu}\|_2^2/(\sigma_p^2 d)$ for some sufficiently large constant $C_4$. It is worth noting that

$$\|\boldsymbol{\zeta}\|_2 = \Theta\bigg(\frac{\|\boldsymbol{\mu}\|_2^2}{\sigma_p^2 d} \cdot \sqrt{n \cdot \sigma_p^2 d}\bigg) = \Theta\bigg(\sqrt{\frac{n\|\boldsymbol{\mu}\|_2^4}{\sigma_p^2 d}}\bigg) \leq 0.02\sigma_p,$$

where the last inequality is by the condition $n\|\boldsymbol{\mu}\|_2^4 \leq C_3\sigma_p^4 d$ for some sufficient small $C_3$. By the construction of $\boldsymbol{\zeta}$, we have almost surely that

$$\sigma(\langle\mathbf{w}_{+1,r}^{(t)}, \boldsymbol{\xi} + \boldsymbol{\zeta}\rangle) - \sigma(\langle\mathbf{w}_{+1,r}^{(t)}, \boldsymbol{\xi}\rangle) + \sigma(\langle\mathbf{w}_{+1,r}^{(t)}, -\boldsymbol{\xi} + \boldsymbol{\zeta}\rangle) - \sigma(\langle\mathbf{w}_{+1,r}^{(t)}, -\boldsymbol{\xi}\rangle)$$

$$\geq \langle\mathbf{w}_{+1,r}^{(t)}, \boldsymbol{\zeta}\rangle$$

$$\geq \lambda\bigg[\sum_{y_i=1} \overline{\rho}_{+1,r,i}^{(t)} - 2n\sqrt{\log(12mn/\delta)} \cdot \sigma_0\sigma_p\sqrt{d} - 16n^2\log(T^*)\sqrt{\log\left(4n^2/\delta\right)/d}\bigg], \quad \text{(F.6)}$$

where the first inequality is by the convexity of ReLU, and the second inequality is by Lemma C.4 and D.3. For the inequality with $j = -1$, we have

$$\sigma(\langle\mathbf{w}_{-1,r}^{(t)}, \boldsymbol{\xi} + \boldsymbol{\zeta}\rangle) - \sigma(\langle\mathbf{w}_{-1,r}^{(t)}, \boldsymbol{\xi}\rangle) + \sigma(\langle\mathbf{w}_{-1,r}^{(t)}, -\boldsymbol{\xi} + \boldsymbol{\zeta}\rangle) - \sigma(\langle\mathbf{w}_{-1,r}^{(t)}, -\boldsymbol{\xi}\rangle)$$

$$\leq 2\big|\langle\mathbf{w}_{-1,r}^{(t)}, \boldsymbol{\zeta}\rangle\big|$$

$$\leq 2\lambda\bigg[\sum_{y_i=1} \rho_{-1,r,i}^{(t)} - 2n\sqrt{\log(12mn/\delta)} \cdot \sigma_0\sigma_p\sqrt{d} - 16n^2\log(T^*)\sqrt{\log\left(4n^2/\delta\right)/d}\bigg], \quad \text{(F.7)}$$

where the first inequality is by the Liptchitz continuous of ReLU, and the second inequality is by Lemma C.4 and D.3. Combing equation F.6 with equation F.7, we have

$$g(\boldsymbol{\xi} + \boldsymbol{\zeta}) - g(\boldsymbol{\xi}) + g(-\boldsymbol{\xi} + \boldsymbol{\zeta}) - g(-\boldsymbol{\xi})$$

$$\geq \lambda\bigg[\sum_r\sum_{y_i=1} \overline{\rho}_{1,r,i}^{(t)}/m - 6n\sqrt{\log(12mn/\delta)} \cdot \sigma_0\sigma_p\sqrt{d} - 48n^2\log(T^*)\sqrt{\log(4n^2/\delta)/d}\bigg]$$

$$\geq (\lambda/2) \cdot \sum_r \sum_{y_i=1} \overline{\rho}_{1,r,i}^{(t)}/m$$

$$\geq (\lambda/2) \cdot \mathrm{SNR}^{-2} \cdot \sum_r \sigma(\langle \mathbf{w}_{+1,r}^{(t)}, \boldsymbol{\mu} \rangle)/m \geq 4 \sum_r \sigma(\langle \mathbf{w}_{+1,r}^{(t)}, \boldsymbol{\mu} \rangle)/m.$$

Here, the first inequality is by Proposition D.8 and Condition 3.1; the second inequality is by the scale of summation over $\overline{\rho}$ in Proposition D.8, and when $t \geq \Omega(nm/(\eta\sigma_p^2 d))$, $\underline{x}_t \geq C$ for some constant $C$; the third inequality is by the upper bound of signal learning in Proposition E.3, and the last inequality is by Condition 3.1. Wrapping all together, the proof for the existence of $\boldsymbol{\zeta}$ completes the proof of Theorem F.1. ∎

We can simply adapt the results in Theorem F.1 to prove Theorem 3.2.

**Proof** [Proof of Theorem 3.2] Here we denote $C_1', C_2', C_3', C_4'$ the constants $C_1, C_2, C_3, C_4$ in Theorem F.1, and then Theorem 3.2 can hold by letting $C_1 = C_3'$, $C_2 = \min\{-\log(1-p)/C_3', C_2'\}$ and $C_3 = C_4'$. We identify three different cases by Theorem F.1: (1) $n\|\boldsymbol{\mu}\|_2^4/\sigma_p^4 d \geq C_1'$, (2) $C_3' \leq n\|\boldsymbol{\mu}\|_2^4/\sigma_p^4 d \leq C_1'$, and (3) $n\|\boldsymbol{\mu}\|_2^4/\sigma_p^4 d \leq C_3'$. We will prove Theorem 3.2 as follows:

1. When $n\|\boldsymbol{\mu}\|_2^4 \geq C_1'\sigma_p^4 d$, we have $R(\mathbf{W}^{(T)}) \leq p + \exp\{-C_2'n\|\boldsymbol{\mu}\|_2^4/\sigma_p^4 d\} \leq p + \exp\{-C_2 n\|\boldsymbol{\mu}\|_2^4/\sigma_p^4 d\}$ due to $C_2 \leq C_2'$;
2. When $C_3'\sigma_p^4 d \leq n\|\boldsymbol{\mu}\|_2^4 \leq C_1'\sigma_p^4 d$, we can bound $R(\mathbf{W}^{(T)}) \leq 1$, and by our choice of $C_2$, we have $p + \exp\{-C_2 n\|\boldsymbol{\mu}\|_2^4/\sigma_p^4 d\} \geq 1 \geq R(\mathbf{W}^{(T)})$ due to $C_2 \leq -\log(1-p)/C_3'$;
3. When $n\|\boldsymbol{\mu}\|_2^4 \leq C_3'\sigma_p^4 d = C_1\sigma_p^4 d$, we have $R(\mathbf{W}^{(T)}) \geq p + C_4' = p + C_3$.

Therefore, from 1 and 2 above we can conclude that when $n\|\boldsymbol{\mu}\|_2^4 \geq C_3'\sigma_p^4 d = C_1\sigma_p^4 d$, $R(\mathbf{W}^{(T)}) \leq p + \exp\{-C_2 n\|\boldsymbol{\mu}\|_2^4/\sigma_p^4 d\}$ with $C_2 = \min\{-\log(1-p)/C_3', C_2'\}$; from 3 above we have $n\|\boldsymbol{\mu}\|_2^4 \leq C_3'\sigma_p^4 d = C_1\sigma_p^4 d$, $R(\mathbf{W}^{(T)}) \geq p + C_4' = p + C_3$. This completes the proof of Theorem 3.2. ∎

# G  ANALYSIS FOR SMALL ANGLE

**In this section, we begin the proof for the case when $\theta$ can be small. It is important to note that in this section and the following section, we only rely on the results presented in Appendix B, C, and Lemma D.1.** The analysis for $\cos\theta \geq 1/2$ is much more complicated. For instance, it is unclear to see the main term of the dynamic on the inner product of $\langle \mathbf{w}_{+1,r}^{(t)}, \mathbf{u} \rangle$, hence additional technique is required.

## G.1  SCALE DIFFERENCE IN TRAINING PROCEDURE

In this section, we start our analysis of the scale in the coefficients during the training procedure.

We give the following proposition. Here, we remind the readers that $\mathrm{SNR} = \|\boldsymbol{\mu}\|_2/(\sigma_p\sqrt{d})$.

**Proposition G.1** *For any $0 \leq t \leq T^*$, it holds that*

$$0 \leq |\langle \mathbf{w}_{j,r}^{(t)}, \mathbf{u} \rangle|, |\langle \mathbf{w}_{j,r}^{(t)}, \mathbf{v} \rangle| \leq 8n \cdot \mathrm{SNR}^2 \log(T^*), \tag{G.1}$$

$$0 \leq \overline{\rho}_{j,r,i}^{(t)} \leq 4\log(T^*), \quad 0 \geq \underline{\rho}_{j,r,i}^{(t)} \geq -2\sqrt{\log(12mn/\delta)} \cdot \sigma_0\sigma_p\sqrt{d} - 32\sqrt{\frac{\log(4n^2/\delta)}{d}}n\log(T^*) \tag{G.2}$$

*for all $j \in \{\pm 1\}$, $r \in [m]$ and $i \in [n]$.*

We use induction to prove Proposition G.1. We introduce several technical lemmas which are applied into the inductive proof of Proposition G.1.

**Lemma G.2** *Under Condition 3.3, suppose equation G.1 and equation G.2 hold at iteration $t$. Then, for all $r \in [m], j \in \{\pm 1\}$ and $i \in [n]$, it holds that*

$$\left|\langle \mathbf{w}_{j,r}^{(t)} - \mathbf{w}_{j,r}^{(0)}, \boldsymbol{\xi}_i \rangle - \underline{\rho}_{j,r,i}^{(t)}\right| \le 16\sqrt{\frac{\log(4n^2/\delta)}{d}} n \log(T^*), \quad j \ne y_i,$$

$$\left|\langle \mathbf{w}_{j,r}^{(t)} - \mathbf{w}_{j,r}^{(0)}, \boldsymbol{\xi}_i \rangle - \overline{\rho}_{j,r,i}^{(t)}\right| \le 16\sqrt{\frac{\log(4n^2/\delta)}{d}} n \log(T^*), \quad j = y_i.$$

**Proof** [Proof of Lemma G.2] We omit the proof due to the similarity in the proof of Lemma D.3. ∎

Now, we define

$$\kappa = 56\sqrt{\frac{\log(4n^2/\delta)}{d}} n \log(T^*) + 10\sqrt{\log(12mn/\delta)} \cdot \sigma_0 \sigma_p \sqrt{d} + 16n \cdot \text{SNR}^2 \log(T^*). \quad \text{(G.3)}$$

By Condition 3.3, it is easy to verify that $\kappa$ is a negligible term. The lemma below gives us a direct characterization of the neural networks output with respect to the time $t$.

**Lemma G.3** *Under Condition 3.3, suppose equation G.1 and equation G.2 hold at iteration $t$. Then, for all $r \in [m]$ it holds that*

$$F_{-y_i}(W_{-y_i}^{(t)}, \mathbf{x}_i) \le \frac{\kappa}{2}, \quad -\frac{\kappa}{2} + \frac{1}{m}\sum_{r=1}^{m} \overline{\rho}_{y_i,r,i}^{(t)} \le F_{y_i}(\mathbf{W}_{y_i}^{(t)}, \mathbf{x}_i) \le \frac{1}{m}\sum_{r=1}^{m} \overline{\rho}_{y_i,r,i}^{(t)} + \frac{\kappa}{2}.$$

*Here, $\kappa$ is defined in equation G.3.*

**Proof** [Proof of Lemma G.3] The proof is similar to the proof of Lemma D.4, we thus omit it. ∎

**Lemma G.4** *Under Condition 3.3, suppose equation G.1 and equation G.2 hold at iteration $t$. Then, for all $i \in [n]$, it holds that*

$$-\frac{\kappa}{2} + \frac{1}{m}\sum_{r=1}^{m} \overline{\rho}_{y_i,r,i}^{(t)} \le y_i f(\mathbf{W}^{(t)}, \mathbf{x}_i) \le \frac{\kappa}{2} + \frac{1}{m}\sum_{r=1}^{m} \overline{\rho}_{y_i,r,i}^{(t)}.$$

*Here, $\kappa$ is defined in equation G.3.*

**Proof** [Proof of Lemma G.4] Note that

$$y_i f(\mathbf{W}^{(t)}, \mathbf{x}_i) = F_{y_i}(\mathbf{W}_{y_i}^{(t)}, \mathbf{x}_i) - F_{-y_i}(\mathbf{W}_{-y_i}^{(t)}, \mathbf{x}_i),$$

the conclusion directly holds from Lemma G.3 ∎

**Lemma G.5** *Under Condition 3.3, suppose equation G.1 and equation G.2 hold for any iteration $t \le T$. Then for any $t \le T$, it holds that:*

1. $1/m \cdot \sum_{r=1}^{m} \left[\overline{\rho}_{y_i,r,i}^{(t)} - \overline{\rho}_{y_k,r,k}^{(t)}\right] \le 3\kappa + 4\sqrt{\log(2n/\delta)/m}$ *for all $i, k \in [n]$.*

2. *Define $S_i^{(t)} := \left\{r \in [m] : \langle \mathbf{w}_{y_i,r}^{(t)}, \boldsymbol{\xi}_i \rangle > 0\right\}$ and $S_{j,r}^{(t)} := \left\{i \in [n] : y_i = j, \langle \mathbf{w}_{j,r}^{(t)}, \boldsymbol{\xi}_i \rangle > 0\right\}$. For all $i \in [n], r \in [m]$ and $j \in \{\pm 1\}$, $S_i^{(0)} = S_i^{(t)}, S_{j,r}^{(0)} = S_{j,r}^{(t)}$.*

3. *Define $\overline{c} = \frac{\eta \sigma_p^2 d}{2nm}(1 + \sqrt{2\log(6n/\delta)/m})(1 + C_0\sqrt{\log(4n/\delta)/d})$, $\underline{c} = \frac{\eta \sigma_p^2 d}{2nm}(1 - \sqrt{2\log(6n/\delta)/m})(1 - C_0\sqrt{\log(4n/\delta)/d})$, $\overline{b} = e^{-\kappa}$ and $\underline{b} = e^{\kappa}$, and let $\overline{x}_t, \underline{x}_t$ be the unique solution of*

$$\overline{x}_t + \overline{b}e^{\overline{x}_t} = \overline{c}t + \overline{b},$$

$$\underline{x}_t + \underline{b}e^{\underline{x}_t} = \underline{c}t + \underline{b},$$

*it holds that*

$$\underline{x}_t \le \frac{1}{m} \sum_{r=1}^{m} \overline{\rho}_{y_i,r,i}^{(t)} \le \overline{x}_t + \overline{c}/(1+\overline{b}), \quad \frac{1}{1+\overline{b}e^{\overline{x}_t}} \le -\ell_i^{\prime(t)} \le \frac{1}{1+\underline{b}e^{\underline{x}_t}}$$

*for all $r \in [m]$ and $i \in [n]$.*

**Proof** [Proof of Lemma G.5] We use induction to prove this lemma. All conclusions hold naturally when $t = 0$. Now, suppose that there exists $\tilde{t} \le T$ such that five conditions hold for any $0 \le t \le \tilde{t}-1$, we prove that these conditions also hold for $t = \tilde{t}$.

We prove conclusion 1 first. By Lemma G.4, we easily see that

$$\left| y_i \cdot f(\mathbf{W}^{(t)}, \mathbf{x}_i) - y_k \cdot f(\mathbf{W}^{(t)}, \mathbf{x}_k) - \frac{1}{m} \sum_{r=1}^{m} \left[ \overline{\rho}_{y_i,r,i}^{(t)} - \overline{\rho}_{y_k,r,k}^{(t)} \right] \right| \le \kappa. \tag{G.4}$$

Recall the update rule for $\overline{\rho}_{j,r,i}^{(t)}$

$$\overline{\rho}_{j,r,i}^{(t+1)} = \overline{\rho}_{j,r,i}^{(t)} - \frac{\eta}{nm} \cdot \ell_i^{\prime(t)} \cdot \mathbf{1}(\langle \mathbf{w}_{j,r}^{(t)}, \boldsymbol{\xi}_i \rangle \ge 0) \cdot \mathbf{1}(y_i = j) \|\boldsymbol{\xi}_i\|_2^2$$

Hence we have

$$\frac{1}{m} \sum_{r=1}^{m} \overline{\rho}_{j,r,i}^{(t+1)} = \frac{1}{m} \sum_{r=1}^{m} \overline{\rho}_{j,r,i}^{(t)} - \frac{\eta}{nm} \cdot \ell_i^{\prime(t)} \cdot \frac{1}{m} \sum_{r=1}^{m} \mathbf{1}(\langle \mathbf{w}_{j,r}^{(t)}, \boldsymbol{\xi}_i \rangle \ge 0) \cdot \mathbf{1}(y_i = j) \|\boldsymbol{\xi}_i\|_2^2$$

for all $j \in \{\pm 1\}, r \in [m], i \in [n]$ and $t \in [T^*]$. Also note that $S_i^{(t)} := \{r \in [m] : \langle \mathbf{w}_{y_i,r}^{(t)}, \boldsymbol{\xi}_i \rangle > 0\}$, we have

$$\frac{1}{m} \sum_{r=1}^{m} \left[ \overline{\rho}_{y_i,r,i}^{(t+1)} - \overline{\rho}_{y_k,r,k}^{(t+1)} \right] = \frac{1}{m} \sum_{r=1}^{m} \left[ \overline{\rho}_{y_i,r,i}^{(t)} - \overline{\rho}_{y_k,r,k}^{(t)} \right] - \frac{\eta}{nm^2} \cdot \left( |S_i^{(t)}| \ell_i^{\prime(t)} \cdot \|\boldsymbol{\xi}_i\|_2^2 - |S_k^{(t)}| \ell_k^{\prime(t)} \cdot \|\boldsymbol{\xi}_k\|_2^2 \right).$$

We prove condition 1 in two cases: $1/m \sum_{r=1}^{m} \left[ \overline{\rho}_{y_i,r,i}^{(\tilde{t}-1)} - \overline{\rho}_{y_k,r,k}^{(\tilde{t}-1)} \right] \le 2\kappa + 3\sqrt{\log(2n/\delta)/m}$ and $1/m \sum_{r=1}^{m} \left[ \overline{\rho}_{y_i,r,i}^{(\tilde{t}-1)} - \overline{\rho}_{y_k,r,k}^{(\tilde{t}-1)} \right] \ge 2\kappa + 3\sqrt{\log(2n/\delta)/m}$.

When $1/m \sum_{r=1}^{m} \left[ \overline{\rho}_{y_i,r,i}^{(\tilde{t}-1)} - \overline{\rho}_{y_k,r,k}^{(\tilde{t}-1)} \right] \le 2\kappa + 3\sqrt{\log(2n/\delta)/m}$, we have

$$\begin{aligned}
\frac{1}{m} \sum_{r=1}^{m} \left[ \overline{\rho}_{y_i,r,i}^{(\tilde{t})} - \overline{\rho}_{y_k,r,k}^{(\tilde{t})} \right] &= \frac{1}{m} \sum_{r=1}^{m} \left[ \overline{\rho}_{y_i,r,i}^{(\tilde{t}-1)} - \overline{\rho}_{y_k,r,k}^{(\tilde{t}-1)} \right] \\
&\quad - \frac{\eta}{nm^2} \cdot \left( |S_i^{(\tilde{t}-1)}| \ell_i^{\prime(\tilde{t}-1)} \cdot \|\boldsymbol{\xi}_i\|_2^2 - |S_k^{(\tilde{t}-1)}| \ell_k^{\prime(\tilde{t}-1)} \cdot \|\boldsymbol{\xi}_k\|_2^2 \right) \\
&\le \frac{1}{m} \sum_{r=1}^{m} \left[ \overline{\rho}_{y_i,r,i}^{(\tilde{t}-1)} - \overline{\rho}_{y_k,r,k}^{(\tilde{t}-1)} \right] + \frac{\eta}{nm} \|\boldsymbol{\xi}_i\|_2^2 \\
&\le 2\kappa + 3\sqrt{\log(2n/\delta)/m} + \kappa + \sqrt{\log(2n/\delta)/m} \\
&\le 3\kappa + 4\sqrt{\log(2n/\delta)/m}.
\end{aligned}$$

Here, the first inequality is by $|S_i^{(\tilde{t}-1)}| \le m$ and $-\ell_i^{\prime(t)} \le 1$, and the second inequality is by the condition of $\eta$ in Condition 3.3.

For when $1/m \sum_{r=1}^{m} \left[ \overline{\rho}_{y_i,r,i}^{(\tilde{t}-1)} - \overline{\rho}_{y_k,r,k}^{(\tilde{t}-1)} \right] \ge 2\kappa + 3\sqrt{\log(2n/\delta)/m}$, from equation G.4 we have

$$y_i \cdot f(\mathbf{W}^{(\tilde{t}-1)}, \mathbf{x}_i) - y_k \cdot f(\mathbf{W}^{(\tilde{t}-1)}, \mathbf{x}_k) \ge \kappa + 3\sqrt{\log(2n/\delta)/m},$$

hence

$$\frac{\ell_i^{\prime(\tilde{t}-1)}}{\ell_k^{\prime(\tilde{t}-1)}} \le \exp\left( y_k \cdot f(\mathbf{W}^{(\tilde{t}-1)}, \mathbf{x}_k) - y_i \cdot f(\mathbf{W}^{(\tilde{t}-1)}, \mathbf{x}_i) \right) \le \exp\left( -\kappa - 3\sqrt{\log(2n/\delta)/m} \right).$$

$$\tag{G.5}$$

Also from condition 2 we have $\left|S_i^{(\tilde{t}-1)}\right| = \left|S_i^{(0)}\right|$ and $\left|S_k^{(\tilde{t}-1)}\right| = \left|S_k^{(0)}\right|$, we have

$$
\begin{aligned}
\frac{\left|S_i^{(\tilde{t}-1)}\right|\left|\ell_i'^{(\tilde{t}-1)}\right|\left\|\boldsymbol{\xi}_i\right\|^2}{\left|S_k^{(\tilde{t}-1)}\right|\left|\ell_k'^{(\tilde{t}-1)}\right|\left\|\boldsymbol{\xi}_i\right\|^2} &= \frac{\left|S_i^{(0)}\right|\left|\ell_i'^{(\tilde{t}-1)}\right|\left\|\boldsymbol{\xi}_i\right\|^2}{\left|S_k^{(0)}\right|\left|\ell_k'^{(\tilde{t}-1)}\right|\left\|\boldsymbol{\xi}_i\right\|^2} \\
&\leq \frac{(1+\sqrt{2\log(2n/\delta)/m})}{(1-\sqrt{2\log(2n/\delta)/m})} \cdot e^{-3\sqrt{\log(2n/\delta)/m}} \cdot e^{-\kappa} \cdot \frac{1+C\sqrt{\log(4n/\delta)/d}}{1-C\sqrt{\log(4n/\delta)/d}} \\
&< 1*1 = 1.
\end{aligned}
$$

Here, the first inequality is by Lemma C.1, equation G.5 and Lemma C.3; the second inequality is by

$$
\frac{1+\sqrt{2}x}{1-\sqrt{2}x} \cdot e^{-3x} < 1 \quad \text{when } 0 < x < 0.1, \quad \kappa \gg 2C\sqrt{\log(4n/\delta)/d}.
$$

By Lemma C.3, under event $\mathcal{E}$, we have

$$
\left|\left\|\boldsymbol{\xi}_i\right\|_2^2 - d \cdot \sigma_p^2\right| \leq C_0\sigma_p^2 \cdot \sqrt{d\log(4n/\delta)}, \forall i \in [n].
$$

Note that $d = \Omega(\log(4n/\delta))$ from Condition 3.3, it follows that

$$
\left|S_i^{(\tilde{t}-1)}\right|\left(-\ell_i'^{(\tilde{t}-1)}\right) \cdot \left\|\boldsymbol{\xi}_i\right\|_2^2 < \left|S_k^{(\tilde{t}-1)}\right|\left(-\ell_k'^{(\tilde{t}-1)}\right) \cdot \left\|\boldsymbol{\xi}_k\right\|_2^2.
$$

We conclude that

$$
\frac{1}{m}\sum_{r=1}^m \left[\overline{\rho}_{y_i,r,i}^{(\tilde{t})} - \overline{\rho}_{y_k,r,k}^{(\tilde{t})}\right] \leq \frac{1}{m}\sum_{r=1}^m \left[\overline{\rho}_{y_i,r,i}^{(\tilde{t}-1)} - \overline{\rho}_{y_k,r,k}^{(\tilde{t}-1)}\right] \leq 3\kappa + 4\sqrt{\log(2n/\delta)/m}.
$$

Hence conclusion 1 holds for $t = \tilde{t}$.

To prove conclusion 2, we prove $S_i^{(0)} = S_i^{(t)}$, and it is quite similar to prove $S_{j,r}^{(0)} = S_{j,r}^{(t)}$. From similar proof in Lemma D.6, we can prove one side that

$$
S_i^{(0)} \subseteq S_i^{(\tilde{t})}.
$$

To prove the other side, we give the update rule for $r \notin S_i^{(0)}$. By induction hypothesis, $r \notin S_i^{(\tilde{t}-1)}$ we have

$$
\begin{aligned}
\left\langle \mathbf{w}_{y_i,r}^{(\tilde{t})}, \boldsymbol{\xi}_i \right\rangle &= \left\langle \mathbf{w}_{y_i,r}^{(\tilde{t}-1)}, \boldsymbol{\xi}_i \right\rangle - \frac{\eta}{nm} \cdot \sum_{i'\neq i} \ell_{i'}^{(\tilde{t}-1)} \cdot \sigma'\left(\left\langle \mathbf{w}_{y_i,r}^{(\tilde{t}-1)}, \boldsymbol{\xi}_{i'} \right\rangle\right) \cdot \left\langle \boldsymbol{\xi}_{i'}, \boldsymbol{\xi}_i \right\rangle \\
&= \left\langle \mathbf{w}_{y_i,r}^{(0)}, \boldsymbol{\xi}_i \right\rangle - \frac{\eta}{nm}\sum_{t'=1}^{\tilde{t}-1} \cdot \sum_{i'\in S_{y_i,r}^{(0)}} \ell_{i'}^{(t')} \cdot \sigma'\left(\left\langle \mathbf{w}_{y_i,r}^{(t')}, \boldsymbol{\xi}_{i'} \right\rangle\right) \cdot \left\langle \boldsymbol{\xi}_{i'}, \boldsymbol{\xi}_i \right\rangle \\
&\leq -\frac{\sigma_p\sigma_0\sqrt{d}\delta}{8m} + \frac{\eta}{nm}\sum_{t'=1}^{\tilde{t}-1}\sum_{i'\in S_{y_i,r}^{(0)}} \frac{|\left\langle \boldsymbol{\xi}_{i'}, \boldsymbol{\xi}_i \right\rangle|}{1+e^{y_{i'}f(\mathbf{W}^{(t')},\mathbf{x}_{i'})}} \\
&\leq -\frac{\sigma_p\sigma_0\sqrt{d}\delta}{8m} + \frac{\eta\sigma_p^2 d}{nm}\sum_{t'=1}^{\tilde{t}-1}\sum_{i'\in S_{y_i,r}^{(0)}} \frac{2 \cdot \sqrt{\log(4n^2/\delta)/d}}{1+e^{y_{i'}f(\mathbf{W}^{(t')},\mathbf{x}_{i'})}} \\
&\leq -\frac{\sigma_p\sigma_0\sqrt{d}\delta}{8m} + \frac{\eta\sigma_p^2 d}{nm}\sum_{t'=1}^{\tilde{t}-1}\sum_{i'\in S_{y_i,r}^{(0)}} \frac{2 \cdot \sqrt{\log(4n^2/\delta)/d}}{1+e^{-\kappa} \cdot e^{\frac{1}{m}\sum_{r=1}^m \overline{\rho}_{y_i,r,i}^{(t')}}} \\
&\leq -\frac{\sigma_p\sigma_0\sqrt{d}\delta}{8m} + \frac{\eta\sigma_p^2 d}{nm}\sum_{t'=1}^{\tilde{t}-1}\sum_{i'\in S_{y_i,r}^{(0)}} \frac{2 \cdot \sqrt{\log(4n^2/\delta)/d}}{1+e^{-\kappa} \cdot e^{\underline{x}_{t'}}}.
\end{aligned}
$$

Here, the first inequality is by Lemma C.4, Lemma C.3; the second inequality is by Lemma G.4; the third inequality is by condition 6 in the induction hypothesis. By the definition of $\underline{x}_{t'}$ that

$$
\underline{x}_t + \underline{b}e^{\underline{x}_t} = \underline{c}t + \underline{b},
$$

we can easy to have

$$\underline{x}_t \leq \log(\underline{c}t/\underline{b} + 1).$$

Then combined the inequality above with $\underline{b} > 1$ and $t \geq 1$, we have

$$\underline{x}_t \geq \log\left(\underline{c}t/\underline{b} + 1 - \log(\underline{c}t/\underline{b} + 1)/\underline{b}\right) \geq \log\left(\underline{c}t/(2\underline{b}) + 1\right).$$

Here, the second inequality is by $\log(x) < x/2$. Therefore, plug the lower bound of $\underline{x}_t$ into the inequality of $\left\langle \mathbf{w}_{y_i,r}^{(\tilde{t})}, \boldsymbol{\xi}_i \right\rangle$ we have that

$$\left\langle \mathbf{w}_{y_i,r}^{(\tilde{t})}, \boldsymbol{\xi}_i \right\rangle \leq -\frac{\sigma_p \sigma_0 \sqrt{d}\delta}{8m} + \frac{\eta \sigma_p^2 d}{nm} \sum_{t'=1}^{\tilde{t}-1} \sum_{i' \in S_{y_i,r}^{(0)}} \frac{2 \cdot \sqrt{\log(4n^2/\delta)/d}}{1 + e^{-\kappa} \cdot e^{\underline{x}_{t'}}}$$

$$\leq -\frac{\sigma_p \sigma_0 \sqrt{d}\delta}{8m} + \frac{\eta \sigma_p^2 d}{nm} \sum_{t'=1}^{\tilde{t}-1} \sum_{i' \in S_{y_i,r}^{(0)}} \frac{2 \cdot \sqrt{\log(4n^2/\delta)/d}}{1 + e^{-\kappa} \cdot (1 + \underline{c}t/(2\underline{b}))}$$

$$\leq -\frac{\sigma_p \sigma_0 \sqrt{d}\delta}{8m} + \frac{\eta \sigma_p^2 d}{m} \int_0^{\tilde{t}} \frac{2 \cdot \sqrt{\log(4n^2/\delta)/d}}{1 + e^{-\kappa} \cdot (1 + \underline{c}t/(2\underline{b}))} \mathrm{d}t$$

$$= -\frac{\sigma_p \sigma_0 \sqrt{d}\delta}{8m} + \frac{\eta \sigma_p^2 d}{m} \cdot 2 \cdot \sqrt{\log(4n^2/\delta)/d} \cdot \frac{2\underline{b}e^{\kappa}}{\underline{c}} \cdot \log\left(1 + \frac{e^{-\kappa}\underline{c}\tilde{t}}{2\underline{b}(1 + e^{-\kappa})}\right)$$

$$\leq -\frac{\sigma_p \sigma_0 \sqrt{d}\delta}{8m} + 5n\sqrt{\log(4n^2/\delta)/d} \cdot \log^2(T^*) < 0.$$

Here, the last inequality is by the selection of $\sigma_0 = nm/(\sigma_p d\delta) \cdot \mathrm{polylog}(d)$. We prove that $r \notin S_i^{(\tilde{t})}$. Hence we have $S_i^{(\tilde{t})} \subseteq S_i^{(0)}$. We conclude that $S_i^{(\tilde{t})} = S_i^{(0)}$. The proof is quite similar for $S_{j,r}^{(t)}$.

As for the last conclusion, recall that

$$\frac{1}{m}\sum_{r=1}^m \overline{\rho}_{y_i,r,i}^{(t+1)} = \frac{1}{m}\sum_{r=1}^m \overline{\rho}_{y_i,r,i}^{(t)} - \frac{\eta}{nm} \cdot \ell_i'^{(t)} \cdot \frac{1}{m}\sum_{r=1}^m \mathbf{1}\left(\left\langle \mathbf{w}_{y_i,r}^{(t)}, \boldsymbol{\xi}_i \right\rangle \geq 0\right) \cdot \mathbf{1}\left(y_i = j\right) \|\boldsymbol{\xi}_i\|_2^2,$$

by condition 2 for $t \in [\tilde{t} - 1]$, we have

$$\frac{1}{m}\sum_{r=1}^m \overline{\rho}_{y_i,r,i}^{(\tilde{t})} = \frac{1}{m}\sum_{r=1}^m \overline{\rho}_{y_i,r,i}^{(\tilde{t}-1)} - \frac{\eta}{nm} \cdot \frac{1}{1 + \exp\left(y_i f(\mathbf{W}^{(\tilde{t})})\right)} \cdot \frac{|S_i^{(0)}|}{m} \cdot \|\boldsymbol{\xi}_i\|_2^2,$$

then Lemma C.1, Lemma C.3 and and Lemma G.4 give that

$$\frac{1}{m}\sum_{r=1}^m \overline{\rho}_{y_i,r,i}^{(\tilde{t})} \leq \frac{1}{m}\sum_{r=1}^m \overline{\rho}_{y_i,r,i}^{(\tilde{t}-1)} + \frac{\eta}{nm} \cdot \frac{1}{1 + e^{-\kappa} \cdot e^{\frac{1}{m}\sum_{r=1}^m \overline{\rho}_{y_i,r,i}^{(\tilde{t}-1)}}} \cdot \left(\frac{1}{2} + \sqrt{2\log(6n/\delta)/m}\right) \cdot \|\boldsymbol{\xi}_i\|_2^2$$

$$\leq \frac{1}{m}\sum_{r=1}^m \overline{\rho}_{y_i,r,i}^{(\tilde{t}-1)} + \frac{\overline{c}}{1 + \overline{b}e^{\frac{1}{m}\sum_{r=1}^m \overline{\rho}_{y_i,r,i}^{(\tilde{t}-1)}}},$$

$$\frac{1}{m}\sum_{r=1}^m \overline{\rho}_{y_i,r,i}^{(\tilde{t})} \geq \frac{1}{m}\sum_{r=1}^m \overline{\rho}_{y_i,r,i}^{(\tilde{t}-1)} + \frac{\eta}{nm} \cdot \frac{1}{1 + e^{\kappa} \cdot e^{\frac{1}{m}\sum_{r=1}^m \overline{\rho}_{y_i,r,i}^{(\tilde{t}-1)}}} \cdot \left(\frac{1}{2} - \sqrt{2\log(6n/\delta)/m}\right) \cdot \|\boldsymbol{\xi}_i\|_2^2$$

$$\geq \frac{1}{m}\sum_{r=1}^m \underline{\rho}_{y_i,r,i}^{(\tilde{t}-1)} + \frac{\underline{c}}{1 + \underline{b}e^{\frac{1}{m}\sum_{r=1}^m \underline{\rho}_{y_i,r,i}^{(\tilde{t}-1)}}}.$$

Combined the two inequalities with Lemma D.1 completes the first result in the last conclusion. As for the second result, by Lemma G.4, it directly holds from

$$\frac{1}{m}\sum_{r=1}^m \overline{\rho}_{y_i,r,i}^{(t)} - \kappa/2 \leq y_i f(\mathbf{W}^{(t)}, \mathbf{x}_i) \leq \frac{1}{m}\sum_{r=1}^m \overline{\rho}_{y_i,r,i}^{(t)} + \kappa/2,$$

$$\overline{c} \leq \kappa/2.$$

This completes the proof of Lemma G.5. ∎

We are now ready to prove Proposition G.1.

**Proof** [Proof of Proposition G.1] With Lemma G.5 above, following the same procedure in the proof of Proposition D.2 completes the proof of Proposition G.5. ∎

We summarize the conclusions above and thus have the following proposition:

**Proposition G.6** *If Condition 3.3 holds, then for any $0 \le t \le T^*$, $j \in \{\pm 1\}$, $r \in [m]$ and $i \in [n]$, it holds that*

$$0 \le |\langle \mathbf{w}_{j,r}^{(t)}, \mathbf{u} \rangle|, |\langle \mathbf{w}_{j,r}^{(t)}, \mathbf{v} \rangle| \le 8n \cdot \mathrm{SNR}^2 \log(T^*),$$

$$0 \le \overline{\rho}_{j,r,i}^{(t)} \le 4\log(T^*), \quad 0 \ge \underline{\rho}_{j,r,i}^{(t)} \ge -2\sqrt{\log(12mn/\delta)} \cdot \sigma_0\sigma_p\sqrt{d} - 32\sqrt{\frac{\log(4n^2/\delta)}{d}} n\log(T^*),$$

*and for any $i^* \in S_{j,r}^{(0)}$ it holds that*

$$|\langle \mathbf{w}_{j,r}^{(t)}, \mathbf{u} \rangle|/\overline{\rho}_{j,r,i^*}^{(t)} \le 2n \cdot \mathrm{SNR}^2, \quad |\langle \mathbf{w}_{j,r}^{(t)}, \mathbf{v} \rangle|/\overline{\rho}_{j,r,i^*}^{(t)} \le 2n \cdot \mathrm{SNR}^2.$$

*Moreover, the following conclusions hold:*

1. $1/m \cdot \sum_{r=1}^m \left[ \overline{\rho}_{y_i,r,i}^{(t)} - \overline{\rho}_{y_k,r,k}^{(t)} \right] \le 3\kappa + 4\sqrt{\log(2n/\delta)/m}$ *for all $i, k \in [n]$.*
2. $-\frac{\kappa}{2} + \frac{1}{m}\sum_{r=1}^m \overline{\rho}_{y_i,r,i}^{(t)} \le y_i f(\mathbf{W}^{(t)}, \mathbf{x}_i) \le \frac{\kappa}{2} + \frac{1}{m}\sum_{r=1}^m \overline{\rho}_{y_i,r,i}^{(t)}$ *for any $i \in [n]$.*
3. *Define $S_i^{(t)} := \left\{ r \in [m] : \langle \mathbf{w}_{y_i,r}^{(t)}, \boldsymbol{\xi}_i \rangle > 0 \right\}$ and $S_{j,r}^{(t)} := \left\{ i \in [n] : y_i = j, \langle \mathbf{w}_{j,r}^{(t)}, \boldsymbol{\xi}_i \rangle > 0 \right\}$. For all $i \in [n]$, $r \in [m]$ and $j \in \{\pm 1\}$, $S_i^{(0)} = S_i^{(t)}$, $S_{j,r}^{(0)} = S_{j,r}^{(t)}$.*
4. *Define $\overline{c} = \frac{\eta\sigma_p^2 d}{2nm}(1 + \sqrt{2\log(6n/\delta)/m})(1 + C_0\sqrt{\log(4n/\delta)/d})$, $\underline{c} = \frac{\eta\sigma_p^2 d}{2nm}(1 - \sqrt{2\log(6n/\delta)/m})(1 - C_0\sqrt{\log(4n/\delta)/d})$, $\overline{b} = e^{-\kappa}$ and $\underline{b} = e^{\kappa}$, and let $\overline{x}_t, \underline{x}_t$ be the unique solution of*

$$\overline{x}_t + \overline{b}e^{\overline{x}_t} = \overline{c}t + \overline{b},$$
$$\underline{x}_t + \underline{b}e^{\underline{x}_t} = \underline{c}t + \underline{b},$$

   *it holds that*

$$\underline{x}_t \le \frac{1}{m}\sum_{r=1}^m \overline{\rho}_{y_i,r,i}^{(t)} \le \overline{x}_t + \overline{c}/(1+\overline{b}), \quad \frac{1}{1+\overline{b}e^{\overline{x}_t}} \le -\ell_i'^{(t)} \le \frac{1}{1+\underline{b}e^{\underline{x}_t}}$$

   *for all $r \in [m]$ and $i \in [n]$.*

The results in Proposition G.6 are adequate for demonstrating the convergence of the training loss, and we shall proceed to establish in Section H. Different from the previous discussion, it is worthy noting that the gap between $\underline{x}_t$ and $\overline{x}_t$ is negligible. It is easy to see that $\overline{x}_t \ge \underline{x}_t > 0$ for $t > 0$. For the other side, combined with Lemma G.8 and G.10, we have

$$\overline{x}_t - \underline{x}_t \le 2(1 - \overline{b}) + (\underline{b} - 1) + 2(\overline{c} - \underline{c})/\underline{c} = o(1).$$

Hence, for any time $t$ which lets $\underline{x}_t \ge C$ for some constant $C > 0$, we conclude that

$$1 \le \overline{x}_t/\underline{x}_t \le 1 + o(1).$$

**Lemma G.7** *It is easy to check that*

$$\log\left(\frac{\eta\sigma_p^2 d}{4nm}t + \frac{2}{3}\right) \le \overline{x}_t \le \log\left(\frac{\eta\sigma_p^2 d}{nm}t + 1\right),$$
$$\log\left(\frac{\eta\sigma_p^2 d}{4nm}t + \frac{2}{3}\right) \le \underline{x}_t \le \log\left(\frac{\eta\sigma_p^2 d}{2nm}t + 1\right).$$

**Proof** [Proof of Lemma G.7] We can easily obtain the inequality by

$$\overline{b}e^{\overline{x}_t} \leq \overline{x}_t + \overline{b}e^{\overline{x}_t} \leq 1.5\overline{b}e^{\overline{x}_t}, \quad \underline{b}e^{\underline{x}_t} \leq \underline{x}_t + \underline{b}e^{\underline{x}_t} \leq 1.5\underline{b}e^{\underline{x}_t},$$

and $3\eta\sigma_p^2 d/(8nm) \leq \overline{c}/\overline{b} \leq \eta\sigma_p^2 d/(nm), 3\eta\sigma_p^2 d/(8nm) \leq \underline{c}/\underline{b} \leq \eta\sigma_p^2 d/(2nm).$ ∎

## G.2 Precise Bound of the Summation of Loss Derivatives

In this section, we give a more precise conclusions on the difference between the summation of loss derivatives. The main idea for virtual sequence comparison is to define a new iterative sequences, then obtain the small difference between the new iterative sequences and the iteration sequences in CNNs. We give several technical lemmas first. The following four lemmas are technical comparison lemmas.

**Lemma G.8** *For any given number $b, c \geq 0$, define two continuous process $x_t$ and $y_t$ with $t \geq 0$ satisfy*

$$x_t + be^{x_t} = ct + x_0 + be^{x_0},$$
$$y_t + e^{y_t} = ct + y_0 + e^{y_0}, \qquad x_0 = y_0.$$

*If $b \geq 0.5$, it holds that*

$$\sup_{t \geq 0} |x_t - y_t| \leq \max\{2(1-b), b-1\}.$$

**Proof** [Proof of Lemma G.8] First, it is easy to see that $x_t, y_t \geq x_0$ and increases with $t$. When $b = 1$, $x_t = y_t$ and the conclusion naturally holds. We prove the case when $b > 1$ and when $0.5 \leq b < 1$.

When $b > 1$, we can see that

$$\begin{aligned}
y_t + b(e^{y_t} - e^{y_0}) &> y_t + (e^{y_t} - e^{y_0}) \\
&= ct + y_0 \\
&= ct + x_0 \\
&= x_t + b(e^{x_t} - e^{x_0}),
\end{aligned}$$

where the first inequality is by $b > 1$ and $y_t > y_0$. By the function $x + b(e^x - e^{x_0})$ is increasing, we have that $y_t > x_t$. We investigate the value $y_t - x_t$. Assume that there exists $t$ such that $y_t - x_t > b - 1$, then we have

$$\begin{aligned}
0 &= x_t + b(e^{x_t} - e^{x_0}) - y_t - (e^{y_t} - e^{y_0}) \\
&< be^{x_t} - e^{y_t} + (1-b) + (1-b)e^{x_0} \\
&= e^{x_t}(b - e^{y_t - x_t}) + (1-b)(1 + e^{x_0}) \\
&< e^{x_t}(b - e^{b-1}) + (1-b)(1 + e^{x_0}) < 0,
\end{aligned}$$

which contradicts to the assumption $y_t - x_t > b - 1$. Here, the first equality is by the definition of $x_t$ and $y_t$, the first and second inequality is by the assumption $y_t - x_t > b - 1$, the last inequality is by the condition $b > 1$. Hence we conclude that when $b > 1$, $y_t > x_t$ and $\sup_{t \geq 0} |x_t - y_t| \leq b - 1$.

When $0.5 \leq b < 1$, we have

$$\begin{aligned}
y_t + b(e^{y_t} - e^{y_0}) &< y_t + (e^{y_t} - e^{y_0}) \\
&= ct + y_0 \\
&= ct + x_0 \\
&= x_t + b(e^{x_t} - e^{x_0}),
\end{aligned}$$

where the first inequality is by $b < 1$ and $y_t > y_0$. Therefore by the function $x + b(e^x - e^{x_0})$ is increasing by $x$, we have that $y_t < x_t$. Assume that there exists $t$ such that $x_t - y_t > 2(1 - b)$, then we have

$$0 = x_t + b(e^{x_t} - e^{x_0}) - y_t - (e^{y_t} - e^{y_0})$$

$$> be^{x_t} - e^{y_t} + 2(1-b) + (1-b)e^{x_0}$$
$$= e^{x_t}(b - e^{y_t - x_t}) + (1-b)(2 + e^{x_0})$$
$$> e^{x_t}(b - e^{2(b-1)}) + (1-b)(2 + e^{x_0}) > 0,$$

which contradicts to the assumption $x_t - y_t > 2(1-b)$. Here, the first equality is by the definition of $x_t$ and $y_t$, the first and second inequality is by the assumption $x_t - y_t > 2(1-b)$, the last inequality is by the condition $0.5 \leq b < 1$ which indicates that $b - e^{2(b-1)} > 0$. Hence we conclude that when $0.5 \leq b < 1$, $y_t < x_t$ and $\sup_{t \geq 0} |x_t - y_t| < 2(1-b)$. The proof of Lemma G.8 completes. ∎

**Lemma G.9** *For any given number $b \geq 0$ and $1 \geq c \geq 0$, define two discrete process $m_t$ and $n_t$ with $t \in \mathbb{N}_+$ satisfy*

$$m_{t+1} = m_t + \frac{c}{1 + be^{m_t}},$$
$$n_{t+1} = n_t + \frac{c}{1 + e^{n_t}}, \qquad m_0 = n_0.$$

*If $b \geq 0.5$, it holds that*

$$\sup_{t \in \mathbb{N}_+} |m_t - n_t| \leq \max\{2(1-b), b-1\} + 2c.$$

**Proof** [Proof of Lemma G.9] Let $x_t$ and $y_t$ be defined in Lemma G.8, we let $m_0 = x_0$ where $x_0$ is defined in Lemma G.8. By Lemma D.1, we have

$$x_t \leq m_t \leq x_t + \frac{c}{1 + be^{m_0}}, \qquad y_t \leq n_t \leq y_t + \frac{c}{1 + e^{n_0}}.$$

We have

$$\sup_{t \in \mathbb{N}_+} |x_t - y_t| \leq \sup_{t \geq 0} |x_t - y_t| \leq \max\{2(1-b), b-1\}, \qquad \sup_{t \in \mathbb{N}_+} |x_t - m_t|, |y_t - n_t| \leq c,$$

by triangle inequality we conclude that

$$\sup_{t \in \mathbb{N}_+} |m_t - n_t| \leq \max\{2(1-b), b-1\} + 2c.$$

This completes the proof. ∎

**Lemma G.10** *Given any number $c_1 \geq c_2 > 0$ and define two continuous process $x_t$, $y_t$ with $t \geq 0$ satisfy*

$$x_t + e^{x_t} = c_1 t + x_0 + e^{x_0},$$
$$y_t + e^{y_t} = c_2 t + y_0 + e^{y_0}, \quad x_0 = y_0.$$

*It holds that*

$$\sup_{t \geq 0} |x_t - y_t| \leq (1 + e^{-x_0}) \cdot \frac{c_1 - c_2}{c_2}.$$

**Proof** [Proof of Lemma G.10] Define $a_0 = x_0 + e^{x_0}$, and the function $g(x)$ with $x \geq 0$ by

$$g(x) + e^{g(x)} - a_0 \equiv x.$$

Easy to see that $x_t = g(c_1 t)$, $y_t = g(c_2 t)$ and $g(x)$ is an strictly increasing function. If

$$g(m_1) - g(m_2) \leq (1 + e^{-x_0}) \cdot \frac{m_1 - m_2}{m_2}$$

holds for any $m_1 \geq m_2 > 0$, we can easily see that

$$x_t - y_t = g(c_1 t) - g(c_2 t) \leq (1 + e^{-x_0}) \cdot \frac{c_1 - c_2}{c_2}.$$

It only remains to prove that

$$g(m_1) - g(m_2) \leq (1 + e^{-x_0}) \cdot \frac{m_1 - m_2}{m_2}$$

holds for any $m_1 \geq m_2 > 0$.

To prove so, define

$$f(m) = g(m) - g(m_2) - (1 + e^{-x_0}) \cdot \frac{m - m_2}{m_2},$$

we can see $f(m_2) = 0$. For $m \geq m_2 > 0$,

$$\begin{aligned}
f'(m) &= g'(m) - (1 + e^{-x_0})/m_2 \\
&= \frac{1}{1 + e^{g(m)}} - \frac{1 + e^{-x_0}}{m_2} \\
&= \frac{m_2 - (1 + e^{x_0})(1 + e^{g(m)})}{(1 + e^{g(m)})m_2} \\
&= \frac{m_2 - (1 + e^{x_0}) \cdot (m + 1 + a_0 - g(m))}{(1 + e^{g(m)})m_2} \\
&= \frac{m_2 - m + (1 + e^{-x_0})g(m) - e^{-x_0}m - (1 + e^{-x_0})(1 + a_0)}{(1 + e^{g(m)})m_2}.
\end{aligned}$$

Here, the second and fourth equality is by the definition of $g(m)$. Define $h(m) = (1 + e^{-x_0})g(m) - e^{-x_0}m - (1 + e^{-x_0})(1 + a_0)$, we have $f'(m) = (m_2 - m + h(m))/(1 + e^{g(m)})m_2$. If $h(m) < 0$, combined this with $m_2 - m \leq 0$ we can conclude that $f'(m) < 0$, which directly prove that $f(m) \leq f(m_2) = 0$.

Now, the only thing remained is to prove that for $m > 0$,

$$h(m) = (1 + e^{-x_0})g(m) - e^{-x_0}m - (1 + e^{-x_0})(1 + a_0) < 0.$$

It is easy to check that $g(0) \leq a_0 = g(0) + e^{g(0)} < a_0 + 1$, therefore $h(0) < 0$. Moreover, it holds that

$$\begin{aligned}
h'(m) &= (1 + e^{-x_0})g'(m) - e^{-x_0} \\
&= \frac{1 + e^{-x_0}}{1 + e^{g(m)}} - e^{-x_0} \\
&\leq e^{-x_0} \cdot \left( \frac{1 + e^{x_0}}{1 + e^{g(0)}} - 1 \right) \leq 0,
\end{aligned}$$

where the second equality is by the definition of $g(m)$, the first inequality is by the property that $g(m)$ is an increasing function and the last inequality is by $x_0 = g(0)$. We have that $h(m) \leq h(0) < 0$, which completes the proof. ∎

**Lemma G.11** *Given any number $1 \geq c_1 \geq c_2 > 0$ and define two discrete process $m_t$, $n_t$ with $t \in \mathbb{N}_+$ satisfy*

$$\begin{aligned}
m_{t+1} &= m_t + \frac{c_1}{1 + e^{m_t}}, \\
n_{t+1} &= n_t + \frac{c_2}{1 + e^{n_t}}, \quad m_0 = n_0.
\end{aligned}$$

*It holds that*

$$\sup_{t \in \mathbb{N}_+} |m_t - n_t| \leq (1 + e^{-m_0}) \cdot \frac{c_1 - c_2}{c_2} + c_1 + c_2.$$

**Proof** [Proof of Lemma G.11] Let $x_t$ and $y_t$ be defined in Lemma G.10, we let $m_0 = x_0$ where $x_0$ is defined in Lemma G.10. By Lemma D.1, we have

$$x_t \leq m_t \leq x_t + \frac{c_1}{1 + e^{m_0}}, \qquad y_t \leq n_t \leq y_t + \frac{c_2}{1 + e^{n_0}}.$$

We have

$$\sup_{t \in \mathbb{N}_+} |x_t - y_t| \le \sup_{t \ge 0} |x_t - y_t| \le (1 + e^{-x_0}) \cdot \frac{c_1 - c_2}{c_2}, \ \sup_{t \in \mathbb{N}_+} |x_t - m_t| \le c_1, \ \sup_{t \in \mathbb{N}_+} |y_t - n_t| \le c_2,$$

by triangle inequality we conclude that

$$\sup_{t \in \mathbb{N}_+} |m_t - n_t| \le (1 + e^{-x_0}) \cdot \frac{c_1 - c_2}{c_2} + c_1 + c_2.$$

This completes the proof. ∎

We now define a new iterative sequences, which are related to the initialization state:

**Definition G.12** *Given the noise vectors $\boldsymbol{\xi}_i$ which are exactly the noise in training samples. Define*

$$\tilde{\ell}_i'^{(t)} = -\frac{1}{1 + \exp\{A_i^{(t)}\}}, \qquad A_i^{(t+1)} = A_i^{(t)} - \frac{\eta}{nm^2} \cdot \tilde{\ell}_i'^{(t)} \cdot |S_i^{(0)}| \cdot \|\boldsymbol{\xi}_i\|_2^2.$$

*Here, $S_i^{(0)} = \{r \in [m] : \langle \mathbf{w}_{y_i,r}^{(0)}, \boldsymbol{\xi}_i \rangle > 0\}$, and $A_i^{(0)} = 0$ for all $i \in [n]$.*

It is worth noting that the Definition G.12 is exactly the same in Lemma 4.2. With Definition G.12, we prove the following lemmas.

**Lemma G.13 (Restatement of Lemma 4.2)** *Let $A_i^{(t)}$ be defined in Definition G.12, it holds that*

$$\left| A_i^{(t)} - \frac{1}{m} \sum_{r=1}^m \overline{\rho}_{y_i,r,i}^{(t)} \right| \le 3\kappa, \quad |\tilde{\ell}_i'^{(t)} - \ell_i'^{(t)}| \le 3\kappa, \quad \ell_i'^{(t)}/\tilde{\ell}_i'^{(t)}, \tilde{\ell}_i'^{(t)}/\ell_i'^{(t)} \le e^{4\kappa},$$

*for all $t \in [T^*]$ and $i \in [n]$.*

**Proof** [Proof of Lemma G.13] By Lemma G.4, we have

$$\frac{1}{1 + e^{\kappa} \exp\{\frac{1}{m} \sum_{r=1}^m \overline{\rho}_{y_i,r,i}^{(t)}\}} \le -\ell_i'^{(t)} \le \frac{1}{1 + e^{-\kappa} \exp\{\frac{1}{m} \sum_{r=1}^m \overline{\rho}_{y_i,r,i}^{(t)}\}}.$$

If the first conclusion holds, we have that

$$\begin{aligned}
\tilde{\ell}_i'^{(t)} - \ell_i'^{(t)} &\le \frac{1}{1 + e^{-\kappa} \exp\{\frac{1}{m} \sum_{r=1}^m \overline{\rho}_{y_i,r,i}^{(t)}\}} - \frac{1}{1 + \exp\{A_i^{(t)}\}} \\
&= \frac{1}{1 + e^{-\kappa} \exp\{\frac{1}{m} \sum_{r=1}^m \overline{\rho}_{y_i,r,i}^{(t)}\}} - \frac{1}{1 + e^{-\kappa} \exp\{A_i^{(t)}\}} \\
&\quad + \frac{1}{1 + e^{-\kappa} \exp\{A_i^{(t)}\}} - \frac{1}{1 + \exp\{A_i^{(t)}\}} \\
&\le \frac{1}{2} \left| A_i^{(t)} - \frac{1}{m} \sum_{r=1}^m \overline{\rho}_{y_i,r,i}^{(t)} \right| + 1 - e^{-\kappa} \le 3\kappa.
\end{aligned}$$

Here, the second inequality is by $(1 + e^{-\kappa}e^x)^{-1} - (1 + e^x)^{-1} \le 1 - e^{-\kappa}$ and $\left| (1 + e^{-\kappa}e^{x_1})^{-1} - (1 + e^{-\kappa}e^{x_2})^{-1} \right| \le \frac{1}{2}|x_1 - x_2|$ for $x \ge 0$ and $\kappa > 0$. The last inequality is by the first conclusion and $1 - e^{-\kappa} \le 1.5\kappa$. Similarly to get that $\ell_i'^{(t)} - \tilde{\ell}_i'^{(t)} \le 3\kappa$. We see that if the first conclusion holds, the second conclusion directly holds.

Simimlarly, we prove if the first conclusion holds, the third conclusion holds. We have

$$\begin{aligned}
\tilde{\ell}_i'^{(t)}/\ell_i'^{(t)} &= \frac{1 + \exp\{A_i^{(t)}\}}{1 + e^{-\kappa} \exp\{\frac{1}{m} \sum_{r=1}^m \overline{\rho}_{y_i,r,i}^{(t)}\}} \\
&\le \exp\left\{ A_i^{(t)} - \frac{1}{m} \sum_{r=1}^m \overline{\rho}_{y_i,r,i}^{(t)} + \kappa \right\}
\end{aligned}$$

$$\leq \exp\{4\kappa\},$$

where the first inequality is by $\ell'(z_1)/\ell'(z_2) \leq \exp(|z_1 - z_2|)$. The proof for $\ell_i'^{(t)}/\tilde{\ell}_i'^{(t)}$ is quite similar and we omit it.

We now prove the first conclusion. Recall the update rule of $\frac{1}{m}\sum_{r=1}^{m}\overline{\rho}_{y_i,r,i}^{(t)}$, we have that

$$\frac{1}{m}\sum_{r=1}^{m}\overline{\rho}_{y_i,r,i}^{(t)} + \frac{\eta|S_i^{(0)}|\|\boldsymbol{\xi}_i\|_2^2/nm^2}{1 + e^{\kappa}e^{\frac{1}{m}\sum_{r=1}^{m}\overline{\rho}_{y_i,r,i}^{(t)}}} \leq \frac{1}{m}\sum_{r=1}^{m}\overline{\rho}_{y_i,r,i}^{(t+1)} = \frac{1}{m}\sum_{r=1}^{m}\overline{\rho}_{y_i,r,i}^{(t)} - \frac{\eta}{nm^2}\cdot\ell_i'^{(t)}\cdot|S_i^{(0)}|\cdot\|\boldsymbol{\xi}_i\|_2^2$$

$$\leq \frac{1}{m}\sum_{r=1}^{m}\overline{\rho}_{y_i,r,i}^{(t)} + \frac{\eta|S_i^{(0)}|\|\boldsymbol{\xi}_i\|_2^2/nm^2}{1 + e^{-\kappa}e^{\frac{1}{m}\sum_{r=1}^{m}\overline{\rho}_{y_i,r,i}^{(t)}}},$$

and

$$A_i^{(t+1)} = A_i^{(t)} + \frac{\eta|S_i^{(0)}|\|\boldsymbol{\xi}_i\|_2^2/nm^2}{1 + \exp\{A_i^{(t)}\}}.$$

Define $B_i^{(t+1)} = B_i^{(t)} + \frac{\eta|S_i^{(0)}|\|\boldsymbol{\xi}_i\|_2^2/nm^2}{1 + e^{-\kappa}e^{B_i^{(t)}}}$, and $B_i^{(t)} = 0$, we have

$$\frac{1}{m}\sum_{r=1}^{m}\overline{\rho}_{y_i,r,i}^{(t)} - A_i^{(t)} \leq B_i^{(t)} - A_i^{(t)}$$

$$\leq 2(1 - e^{-\kappa}) + 2\eta|S_i^{(0)}|\|\boldsymbol{\xi}_i\|_2^2/nm^2 \leq 3\kappa.$$

Here, the first inequality is by $\frac{1}{m}\sum_{r=1}^{m}\overline{\rho}_{y_i,r,i}^{(t)} \leq B_i^{(t)}$, the second inequality is by Lemma G.9 and the third inequality is by the condition of $\eta$ in Condition 3.3 and $\kappa < 0.01$. We can similarly have that

$$A_i^{(t)} - \frac{1}{m}\sum_{r=1}^{m}\overline{\rho}_{y_i,r,i}^{(t)} \leq 3\kappa,$$

which completes the proof. $\blacksquare$

We next give the precise bound of $\sum_{i\in S_+}\tilde{\ell}_i'^{(t)}/\sum_{i\in S_-}\tilde{\ell}_i'^{(t)}$.

**Lemma G.14 (Restatement of Lemma 4.3)** *Let $\tilde{\ell}_i'^{(t)}$, $S_+$ and $S_-$ be defined in Definition G.12, then it holds that*

$$\left|\frac{\sum_{i\in S_+}\tilde{\ell}_i'^{(t)}}{\sum_{i\in S_-}\tilde{\ell}_i'^{(t)}} - \frac{|S_+|}{|S_-|}\right| \leq \frac{2\mathcal{G}_{\text{gap}}(|S_-|\sqrt{|S_+|} + |S_-|\sqrt{|S_+|})}{|S_-|^2}$$

*with probability at least $1 - 2\delta$. Here, $\mathcal{G}_{\text{gap}}$ is defined by*

$$\mathcal{G}_{\text{gap}} = 20\sqrt{\log(2n/\delta)/m}\cdot\sqrt{\log(4/\delta)}.$$

**Proof** [Proof of Lemma G.14] By Lemma G.11, we have

$$\left|A_i^{(t)} - A_k^{(t)}\right| \leq \frac{2\left|\eta|S_i^{(0)}|\|\boldsymbol{\xi}_i\|_2^2/nm^2 - \eta|S_k^{(0)}|\|\boldsymbol{\xi}_k\|_2^2/nm^2\right|}{\min\{\eta|S_i^{(0)}|\|\boldsymbol{\xi}_i\|_2^2/nm^2, \eta|S_k^{(0)}|\|\boldsymbol{\xi}_k\|_2^2/nm^2\}}$$

$$+ \frac{\eta}{4nm^2}(|S_i^{(0)}|\|\boldsymbol{\xi}_i\|_2^2 + |S_k^{(0)}|\|\boldsymbol{\xi}_k\|_2^2).$$

By Lemma C.1 and Lemma C.3, we can see that

$$\frac{m}{2} - \sqrt{\frac{m\log(2n/\delta)}{2}} \leq |S_i^{(0)}|, |S_k^{(0)}| \leq \frac{m}{2} + \sqrt{\frac{m\log(2n/\delta)}{2}},$$

$$\sigma_p^2 d - C_0 \sigma_p^2 \sqrt{d \cdot \log(4n/\delta)} \le \|\boldsymbol{\xi}_i\|_2^2, \|\boldsymbol{\xi}_k\|_2^2 \le \sigma_p^2 d + C_0 \sigma_p^2 \sqrt{d \cdot \log(4n/\delta)}.$$

By the condition of $d$ in Condition 3.3, we easily conclude that

$$\frac{\big||S_i^{(0)}|\|\boldsymbol{\xi}_i\|_2^2 - |S_k^{(0)}|\|\boldsymbol{\xi}_k\|_2^2\big|}{\min\{|S_i^{(0)}|\|\boldsymbol{\xi}_i\|_2^2, |S_k^{(0)}|\|\boldsymbol{\xi}_k\|_2^2\}} \le 2\sqrt{\log(2n/\delta)/m}, \tag{G.6}$$

we conclude that

$$\big|A_i^{(t)} - A_k^{(t)}\big| \le 5\sqrt{\log(2n/\delta)/m}. \tag{G.7}$$

The inequality is by the equation G.6 and the condition of $\eta$ in Condition 3.3. For any $\zeta$ between $A_i^{(t)}$ and $A_k^{(t)}$, we have $|\ell_i''(\zeta)| \le (1 + \exp(\zeta))^{-1}$, and

$$(1 + \exp(\zeta))^{-1} - 2\Big(1 + \exp\big(A_i^{(t)}\big)\Big)^{-1} \le 0$$

by $\big|\zeta - A_i^{(t)}\big| \le 5\sqrt{\log(2n/\delta)/m} \le 0.01$. We have that for any $i \ne k$,

$$\big|\tilde{\ell}_i'^{(t)} - \tilde{\ell}_k'^{(t)}\big| = \left| \frac{1}{1 + \exp\{A_k^{(t)}\}} - \frac{1}{1 + \exp\{A_i^{(t)}\}} \right|$$

$$\le 2\tilde{\ell}_i'^{(t)} \big|A_i^{(t)} - A_k^{(t)}\big|.$$

By equation G.7, we conclude that

$$\big|\tilde{\ell}_k'^{(t)}/\tilde{\ell}_i'^{(t)} - 1\big| \le 10\sqrt{\log(2n/\delta)/m}$$

for all $i, k$ in the index set $S_+$ and $S_-$.

Conditional on the event $\mathcal{E}$, we can see that the bound above holds almost surely. We denote $\tilde{\ell}_0'^{(t)}$ by an independent copy of $\tilde{\ell}_i'^{(t)}$, and note that $\tilde{\ell}_i'^{(t)}$ and $\tilde{\ell}_k'^{(t)}$ are independent and have same distribution, we assume that $C^{(t)} = \mathbb{E}\tilde{\ell}_i'^{(t)}/\tilde{\ell}_k'^{(t)}$, easy to see $0.5 \le C^{(t)} \le 2$. By Hoeffeding inequality we have

$$P\left( \left| \frac{1}{|S_+|} \sum_{i \in S_+} \tilde{\ell}_i'^{(t)}/\tilde{\ell}_0'^{(t)} - C^{(t)} \right| > t_1 \right) \le 2\exp\left\{ -\frac{2|S_+|t_1^2}{(20\sqrt{\log(2n/\delta)/m})^2} \right\},$$

$$P\left( \left| \frac{1}{|S_-|} \sum_{i \in S_+} \tilde{\ell}_i'^{(t)}/\tilde{\ell}_0'^{(t)} - C^{(t)} \right| > t_2 \right) \le 2\exp\left\{ -\frac{2|S_-|t_2^2}{(20\sqrt{\log(2n/\delta)/m})^2} \right\}.$$

Let $t_1 = \big(20\sqrt{\log(2n/\delta)/m}\big)/\sqrt{2|S_+|} \cdot \sqrt{\log(4/\delta)}$ and $t_2 = \big(20\sqrt{\log(2n/\delta)/m}\big)/\sqrt{2|S_-|} \cdot \sqrt{\log(4/\delta)}$, and write $\mathcal{G}_{\text{gap}} = \big(20\sqrt{\log(2n/\delta)/m}\big) \cdot \sqrt{\log(4/\delta)}$ in short hand, we conclude that

$$P\left( |S_+| \cdot C^{(t)} - \mathcal{G}_{\text{gap}} \cdot \sqrt{|S_+|} \le \left| \sum_{i \in S_+} \tilde{\ell}_i'^{(t)}/\tilde{\ell}_0'^{(t)} \right| \le |S_+| \cdot C^{(t)} + \mathcal{G}_{\text{gap}} \cdot \sqrt{|S_+|} \right) \ge 1 - \delta/2,$$

$$P\left( |S_-| \cdot C^{(t)} - \mathcal{G}_{\text{gap}} \cdot \sqrt{|S_-|} \le \left| \sum_{i \in S_-} \tilde{\ell}_i'^{(t)}/\tilde{\ell}_0'^{(t)} \right| \le |S_-| \cdot C^{(t)} + \mathcal{G}_{\text{gap}} \cdot \sqrt{|S_-|} \right) \ge 1 - \delta/2.$$

By the inequality

$$P(A \cap B) \ge P(A) + P(B) - 1,$$

we have that

$$P\left( \frac{|S_+| \cdot C^{(t)} - \mathcal{G}_{\text{gap}} \cdot \sqrt{|S_+|}}{|S_-| \cdot C^{(t)} + \mathcal{G}_{\text{gap}} \cdot \sqrt{|S_-|}} \le \frac{\sum_{i \in S_+} \tilde{\ell}_i'^{(t)}}{\sum_{i \in S_-} \tilde{\ell}_i'^{(t)}} \le \frac{|S_+| \cdot C^{(t)} + \mathcal{G}_{\text{gap}} \cdot \sqrt{|S_+|}}{|S_-| \cdot C^{(t)} - \mathcal{G}_{\text{gap}} \cdot \sqrt{|S_-|}} \right) \ge 1 - \delta.$$

By $0.5 \le C^{(t)} \le 2$, we directly get that

$$P\left( \left| \frac{\sum_{i \in S_+} \tilde{\ell}_i'^{(t)}}{\sum_{i \in S_-} \tilde{\ell}_i'^{(t)}} - \frac{|S_+|}{|S_-|} \right| \le \frac{2\mathcal{G}_{\text{gap}}(|S_-|\sqrt{|S_+|} + |S_-|\sqrt{|S_+|})}{|S_-|^2} \right) \ge 1 - \delta.$$

Let event $\mathcal{E}_1$ be

$$\left|\frac{\sum_{i\in S_+}\tilde{\ell}_i^{\prime(t)}}{\sum_{i\in S_-}\tilde{\ell}_i^{\prime(t)}} - \frac{|S_+|}{|S_-|}\right| \leq \frac{2\mathcal{G}_{\mathrm{gap}}(|S_-|\sqrt{|S_+|}+|S_-|\sqrt{|S_+|})}{|S_-|^2},$$

we have

$$P(\mathcal{E}_1,\mathcal{E}) = P(\mathcal{E}_1|\mathcal{E}) \cdot P(\mathcal{E}) \geq (1-\delta)^2 \geq 1-2\delta.$$

This completes the proof. ∎

We are now ready to give the proposition which characterizes a more precise bound of the summation of loss derivatives.

**Proposition G.15** *If Condition 3.3 holds, then for any $0 \leq t \leq T^*$, it holds with probability at least $1 - 2\delta$ such that*

$$\left|\frac{\sum_{i\in S_+}\ell_i^{\prime(t)}}{\sum_{i\in S_-}\ell_i^{\prime(t)}} - \frac{|S_+|}{|S_-|}\right| \leq \frac{2\mathcal{G}_{\mathrm{gap}}(|S_-|\sqrt{|S_+|}+|S_-|\sqrt{|S_+|})}{|S_-|^2} \cdot e^{8\kappa} + 10\kappa \cdot \frac{|S_+|}{|S_-|}.$$

*Here,*

$$\mathcal{G}_{\mathrm{gap}} = 20\sqrt{\log(2n/\delta)/m} \cdot \sqrt{\log(4/\delta)}.$$

**Proof** [Proof of Proposition G.15] By Lemma G.13, we have

$$|\ell_i^{\prime(t)}/\tilde{\ell}_i^{\prime(t)}|, |\tilde{\ell}_i^{\prime(t)}/\ell_i^{\prime(t)}| \leq e^{4\kappa}.$$

By Lemma G.14, we obtain that

$$\left|\frac{\sum_{i\in S_+}\tilde{\ell}_i^{\prime(t)}}{\sum_{i\in S_-}\tilde{\ell}_i^{\prime(t)}} - \frac{|S_+|}{|S_-|}\right| \leq \frac{2\mathcal{G}_{\mathrm{gap}}(|S_-|\sqrt{|S_+|}+|S_-|\sqrt{|S_+|})}{|S_-|^2}.$$

We conclude that

$$e^{-8\kappa}\frac{\sum_{i\in S_+}\tilde{\ell}_i^{\prime(t)}}{\sum_{i\in S_-}\tilde{\ell}_i^{\prime(t)}} \leq \frac{\sum_{i\in S_+}\ell_i^{\prime(t)}}{\sum_{i\in S_-}\ell_i^{\prime(t)}} \leq e^{8\kappa}\frac{\sum_{i\in S_+}\tilde{\ell}_i^{\prime(t)}}{\sum_{i\in S_-}\tilde{\ell}_i^{\prime(t)}}. \tag{G.8}$$

Combined the results in Lemma G.14 with equation G.8, we conclude that

$$\left|\frac{\sum_{i\in S_+}\ell_i^{\prime(t)}}{\sum_{i\in S_-}\ell_i^{\prime(t)}} - \frac{|S_+|}{|S_-|}\right| \leq \frac{2\mathcal{G}_{\mathrm{gap}}(|S_-|\sqrt{|S_+|}+|S_-|\sqrt{|S_+|})}{|S_-|^2} \cdot e^{8\kappa} + 10\kappa \cdot \frac{|S_+|}{|S_-|}.$$

Here, the inequality is by the fact that $\kappa$ is small. This completes the proof. ∎

From Proposition G.15, we can directly get the following lemma.

**Lemma G.16 (Rstatement of Lemma 4.4)** *Under the same condition of Proposition G.15, if*

$$c_0 n - C\sqrt{n \cdot \log(8n/\delta)} \leq |S_+|, |S_-| \leq c_1 n + C\sqrt{n \cdot \log(8n/\delta)}$$

*holds for some constant $c_0, c_1, C > 0$, then it holds that*

$$\left|\frac{\sum_{i\in S_+}\ell_i^{\prime(t)}}{\sum_{i\in S_-}\ell_i^{\prime(t)}} - \frac{c_1}{c_0}\right| \leq \frac{4c_1 C}{c_0^2} \cdot \sqrt{\frac{\log(8n/\delta)}{n}}.$$

**Proof** [Proof of Lemma G.16] By the condition

$$c_0 n - C\sqrt{n \cdot \log(8n/\delta)} \le |S_+|, |S_-| \le c_1 n + C\sqrt{n \cdot \log(8n/\delta)},$$

it is easy to see that

$$\frac{2\mathcal{G}_{\text{gap}}(|S_-|\sqrt{|S_+|} + |S_-|\sqrt{|S_+|})}{|S_-|^2} = o(1/\sqrt{n}), \quad 10\kappa \cdot \frac{|S_+|}{|S_-|} = o(1/\sqrt{n}),$$

where we utilize $\kappa = o(1/\sqrt{n})$ by Condition 3.3. Hence we have

$$\left| \frac{\sum_{i \in S_+} \ell_i'^{(t)}}{\sum_{i \in S_-} \ell_i'^{(t)}} - \frac{|S_+|}{|S_-|} \right| \le o(1/\sqrt{n}). \tag{G.9}$$

Moreover, we have

$$(c_0 n - C\sqrt{n \cdot \log(8n/\delta)}) \cdot \left( \frac{c_1}{c_0} + \frac{3c_1 C}{c_0^2} \cdot \sqrt{\log(8n/\delta)/n} \right) \ge c_1 n + C\sqrt{n \cdot \log(8n/\delta)},$$

$$(c_0 n - C\sqrt{n \cdot \log(8n/\delta)}) \cdot \left( \frac{c_1}{c_0} - \frac{3c_1 C}{c_0^2} \cdot \sqrt{\log(8n/\delta)/n} \right) \le c_1 n + C\sqrt{n \cdot \log(8n/\delta)}$$

we conclude that

$$\left| \frac{|S_+|}{|S_-|} - \frac{c_1}{c_0} \right| \le \frac{3c_1 C}{c_0^2} \cdot \sqrt{\log(8n/\delta)/n}.$$

Replacing the equation above into equation G.9 completes the proof. ∎

### G.3 SIGNAL LEARNING AND NOISE MEMORIZATION

We first give some lemmas on the inner product of $\mathbf{w}_{j,r}^{(t)}$ and $\mathbf{u}, \mathbf{v}$.

**Lemma G.17** *Under Condition 3.3, the following conclusions hold:*

1. *If $\langle \mathbf{w}_{+1,r}^{(0)}, \mathbf{u} \rangle > 0 (< 0)$, then $\langle \mathbf{w}_{+1,r}^{(t)}, \mathbf{u} \rangle$ strictly increases (decreases) with $t \in [T^*]$;*
2. *If $\langle \mathbf{w}_{-1,r}^{(0)}, \mathbf{v} \rangle > 0 (< 0)$, then $\langle \mathbf{w}_{-1,r}^{(t)}, \mathbf{v} \rangle$ strictly increases (decreases) with $t \in [T^*]$;*
3. *$|\langle \mathbf{w}_{+1,r}^{(t)}, \mathbf{v} \rangle| \le |\langle \mathbf{w}_{+1,r}^{(0)}, \mathbf{v} \rangle| + \eta \|\boldsymbol{\mu}\|_2^2/m, |\langle \mathbf{w}_{-1,r}^{(t)}, \mathbf{u} \rangle| \le |\langle \mathbf{w}_{-1,r}^{(0)}, \mathbf{u} \rangle| + \eta \|\boldsymbol{\mu}\|_2^2/m$ for all $t \in [T^*]$ and $r \in [m]$.*

*Moreover, it holds that*

$$\langle \mathbf{w}_{+1,r}^{(t+1)}, \mathbf{u} \rangle \ge \langle \mathbf{w}_{+1,r}^{(t)}, \mathbf{u} \rangle - \frac{\eta \|\boldsymbol{\mu}\|_2^2}{nm} \sum_{i \in S_{+\mathbf{u},+1}} \ell_i'^{(t)} \cdot \frac{1-2p}{2(1-p)} \cdot (1 - \cos\theta), \quad \langle \mathbf{w}_{+1,r}^{(0)}, \mathbf{u} \rangle > 0;$$

$$\langle \mathbf{w}_{+1,r}^{(t+1)}, \mathbf{u} \rangle \le \langle \mathbf{w}_{+1,r}^{(t)}, \mathbf{u} \rangle + \frac{\eta \|\boldsymbol{\mu}\|_2^2}{nm} \cdot \sum_{i \in S_{-\mathbf{u},+1}} \ell_i'^{(t)} \cdot \frac{1-2p}{2(1-p)} \cdot (1 - \cos\theta), \quad \langle \mathbf{w}_{+1,r}^{(0)}, \mathbf{u} \rangle < 0;$$

$$\langle \mathbf{w}_{-1,r}^{(t+1)}, \mathbf{v} \rangle \ge \langle \mathbf{w}_{-1,r}^{(t)}, \mathbf{v} \rangle - \frac{\eta \|\boldsymbol{\mu}\|_2^2}{nm} \cdot \sum_{i \in S_{+\mathbf{v},-1}} \ell_i'^{(t)} \cdot \frac{1-2p}{2(1-p)} \cdot (1 - \cos\theta), \quad \langle \mathbf{w}_{-1,r}^{(0)}, \mathbf{v} \rangle > 0;$$

$$\langle \mathbf{w}_{-1,r}^{(t+1)}, \mathbf{v} \rangle \le \langle \mathbf{w}_{-1,r}^{(t)}, \mathbf{v} \rangle + \frac{\eta \|\boldsymbol{\mu}\|_2^2}{nm} \cdot \sum_{i \in S_{-\mathbf{v},-1}} \ell_i'^{(t)} \cdot \frac{1-2p}{2(1-p)} \cdot (1 - \cos\theta), \quad \langle \mathbf{w}_{-1,r}^{(0)}, \mathbf{v} \rangle < 0.$$

*Similarly, it also holds that*

$$\langle \mathbf{w}_{+1,r}^{(t+1)}, \mathbf{u} \rangle \le \langle \mathbf{w}_{+1,r}^{(t)}, \mathbf{u} \rangle - \frac{2\eta \|\boldsymbol{\mu}\|_2^2}{nm} \sum_{i \in [n]} \ell_i'^{(t)}, \quad \langle \mathbf{w}_{+1,r}^{(0)}, \mathbf{u} \rangle > 0;$$

$$\langle \mathbf{w}_{+1,r}^{(t+1)}, \mathbf{u} \rangle \le \langle \mathbf{w}_{+1,r}^{(t)}, \mathbf{u} \rangle + \frac{2\eta \|\boldsymbol{\mu}\|_2^2}{nm} \sum_{i \in [n]} \ell_i'^{(t)}, \quad \langle \mathbf{w}_{+1,r}^{(0)}, \mathbf{u} \rangle < 0;$$

$$\langle \mathbf{w}_{-1,r}^{(t+1)}, \mathbf{v}\rangle \geq \langle \mathbf{w}_{-1,r}^{(t)}, \mathbf{v}\rangle - \frac{2\eta\|\boldsymbol{\mu}\|_2^2}{nm}\sum_{i\in[n]}\ell_i'^{(t)}, \quad \langle \mathbf{w}_{-1,r}^{(0)}, \mathbf{v}\rangle > 0;$$

$$\langle \mathbf{w}_{-1,r}^{(t+1)}, \mathbf{v}\rangle \leq \langle \mathbf{w}_{-1,r}^{(t)}, \mathbf{v}\rangle + \frac{2\eta\|\boldsymbol{\mu}\|_2^2}{nm}\sum_{i\in[n]}\ell_i'^{(t)}, \quad \langle \mathbf{w}_{-1,r}^{(0)}, \mathbf{v}\rangle < 0.$$

**Proof** [Proof of Lemma G.17] Recall that the update rule for inner product can be written as

$$\langle \mathbf{w}_{j,r}^{(t+1)}, \mathbf{u}\rangle = \langle \mathbf{w}_{j,r}^{(t)}, \mathbf{u}\rangle - \frac{\eta j}{nm}\sum_{i\in S_{+\mathbf{u},+1}\cup S_{-\mathbf{u},-1}}\ell_i'^{(t)}\cdot\mathbf{1}\{\langle\mathbf{w}_{j,r}^{(t)},\boldsymbol{\mu}_i\rangle > 0\}\|\boldsymbol{\mu}\|_2^2$$

$$+\frac{\eta j}{nm}\sum_{i\in S_{-\mathbf{u},+1}\cup S_{+\mathbf{u},-1}}\ell_i'^{(t)}\cdot\mathbf{1}\{\langle\mathbf{w}_{j,r}^{(t)},\boldsymbol{\mu}_i\rangle > 0\}\|\boldsymbol{\mu}\|_2^2$$

$$+\frac{\eta j}{nm}\sum_{i\in S_{+\mathbf{v},-1}\cup S_{-\mathbf{v},+1}}\ell_i'^{(t)}\cdot\mathbf{1}\{\langle\mathbf{w}_{j,r}^{(t)},\boldsymbol{\mu}_i\rangle > 0\}\|\boldsymbol{\mu}\|_2^2\cos\theta$$

$$-\frac{\eta j}{nm}\sum_{i\in S_{-\mathbf{v},-1}\cup S_{+\mathbf{v},+1}}\ell_i'^{(t)}\cdot\mathbf{1}\{\langle\mathbf{w}_{j,r}^{(t)},\boldsymbol{\mu}_i\rangle > 0\}\|\boldsymbol{\mu}\|_2^2\cos\theta,$$

and

$$\langle \mathbf{w}_{j,r}^{(t+1)}, \mathbf{v}\rangle = \langle \mathbf{w}_{j,r}^{(t)}, \mathbf{v}\rangle - \frac{\eta j}{nm}\sum_{i\in S_{+\mathbf{u},+1}\cup S_{-\mathbf{u},-1}}\ell_i'^{(t)}\cdot\mathbf{1}\{\langle\mathbf{w}_{j,r}^{(t)},\boldsymbol{\mu}_i\rangle > 0\}\|\boldsymbol{\mu}\|_2^2\cos\theta$$

$$+\frac{\eta j}{nm}\sum_{i\in S_{-\mathbf{u},+1}\cup S_{+\mathbf{u},-1}}\ell_i'^{(t)}\cdot\mathbf{1}\{\langle\mathbf{w}_{j,r}^{(t)},\boldsymbol{\mu}_i\rangle > 0\}\|\boldsymbol{\mu}\|_2^2\cos\theta$$

$$+\frac{\eta j}{nm}\sum_{i\in S_{+\mathbf{v},-1}\cup S_{-\mathbf{v},+1}}\ell_i'^{(t)}\cdot\mathbf{1}\{\langle\mathbf{w}_{j,r}^{(t)},\boldsymbol{\mu}_i\rangle > 0\}\|\boldsymbol{\mu}\|_2^2$$

$$-\frac{\eta j}{nm}\sum_{i\in S_{-\mathbf{v},-1}\cup S_{+\mathbf{v},+1}}\ell_i'^{(t)}\cdot\mathbf{1}\{\langle\mathbf{w}_{j,r}^{(t)},\boldsymbol{\mu}_i\rangle > 0\}\|\boldsymbol{\mu}\|_2^2.$$

When $\langle\mathbf{w}_{+1,r}^{(0)}, \mathbf{u}\rangle > 0$, assume that for any $0 \leq t \leq \tilde{T}-1$ such that $\langle\mathbf{w}_{+1,r}^{(t)}, \mathbf{u}\rangle > 0$, there are two cases for $\langle\mathbf{w}_{+1,r}^{(t)}, \mathbf{v}\rangle$. When $\langle\mathbf{w}_{+1,r}^{(\tilde{T}-1)}, \mathbf{v}\rangle < 0$, we can easily see

$$\frac{\eta}{nm}\left(\sum_{i\in S_{-\mathbf{v},+1}}\ell_i'^{(\tilde{T}-1)} - \sum_{i\in S_{-\mathbf{v},-1}}\ell_i'^{(\tilde{T}-1)}\right)\cdot\mathbf{1}\{\langle\mathbf{w}_{+1,r}^{(\tilde{T}-1)},\boldsymbol{\mu}_i\rangle > 0\}\|\boldsymbol{\mu}\|_2^2\cos\theta > 0,$$

$$\frac{\eta}{nm}\left(-\sum_{i\in S_{+\mathbf{u},+1}}\ell_i'^{(\tilde{T}-1)} + \sum_{i\in S_{+\mathbf{u},-1}}\ell_i'^{(\tilde{T}-1)}\right)\cdot\mathbf{1}\{\langle\mathbf{w}_{+1,r}^{(\tilde{T}-1)},\boldsymbol{\mu}_i\rangle > 0\}\|\boldsymbol{\mu}\|_2^2 > 0$$

from Lemma C.2 and G.16. Hence $\langle\mathbf{w}_{+1,r}^{(\tilde{T})}, \mathbf{u}\rangle \geq \langle\mathbf{w}_{+1,r}^{(\tilde{T}-1)}, \mathbf{u}\rangle > 0$. When $\langle\mathbf{w}_{+1,r}^{(\tilde{T}-1)}, \mathbf{v}\rangle > 0$, the update rule can be simplified as

$$\langle\mathbf{w}_{+1,r}^{(\tilde{T})}, \mathbf{u}\rangle = \langle\mathbf{w}_{+1,r}^{(\tilde{T}-1)}, \mathbf{u}\rangle$$
$$-\frac{\eta\|\boldsymbol{\mu}\|_2^2}{nm}\left(\sum_{i\in S_{+\mathbf{u},+1}}\ell_i'^{(\tilde{T}-1)} - \sum_{i\in S_{+\mathbf{u},-1}}\ell_i'^{(\tilde{T}-1)} - \sum_{i\in S_{+\mathbf{v},-1}}\ell_i'^{(\tilde{T}-1)}\cos\theta + \sum_{i\in S_{+\mathbf{v},+1}}\ell_i'^{(\tilde{T}-1)}\cos\theta\right).$$
$$\text{(G.10)}$$

Note that by Lemma C.2 and G.16, it is easy to verify that

$$-\sum_{i\in S_{+\mathbf{v},-1}}\ell_i'^{(\tilde{T}-1)}\cos\theta + \sum_{i\in S_{-\mathbf{v},+1}}\ell_i'^{(\tilde{T}-1)}\cos\theta > 0,$$

thus for both cases $\langle\mathbf{w}_{+1,r}^{(\tilde{T}-1)}, \mathbf{v}\rangle > 0$ and $\langle\mathbf{w}_{+1,r}^{(\tilde{T}-1)}, \mathbf{v}\rangle < 0$, we have that there exists an absolute constant $C$, such that

$$\langle\mathbf{w}_{+1,r}^{(\tilde{T})}, \mathbf{u}\rangle \geq \langle\mathbf{w}_{+1,r}^{(\tilde{T}-1)}, \mathbf{u}\rangle - \frac{\eta\|\boldsymbol{\mu}\|_2^2}{nm}\sum_{i\in S_{+\mathbf{u},+1}}\ell_i'^{(\tilde{T}-1)}\cdot\left(1 - \frac{p}{1-p} - \frac{C}{8}\cdot\sqrt{\frac{\log(8n/\delta)}{n}}\right)$$

$$-\frac{\eta\|\boldsymbol{\mu}\|_2^2}{nm}\sum_{i\in S_{+\mathbf{u},+1}}\ell_i'^{(\tilde{T}-1)}\cdot\left(-1-\frac{C}{4}\cdot\sqrt{\frac{\log(8n/\delta)}{n}}+\frac{p}{1-p}\right)\cdot\cos\theta.$$

Here, the inequality comes from Lemma G.16. By the condition of $\theta$ in Condition 3.3 that

$$1-\cos\theta\geq\widetilde{\Omega}(1/\sqrt{n}),$$

we have

$$1-\frac{(1-2p)/(1-p)+\frac{C}{4}\cdot\sqrt{\frac{\log(8n/\delta)}{n}}}{(1-2p)/(1-p)-\frac{C}{8}\cdot\sqrt{\frac{\log(8n/\delta)}{n}}}\cos\theta=1-\cos\theta-\frac{\frac{3C}{8}\cdot\sqrt{\frac{\log(8n/\delta)}{n}}}{(1-2p)/(1-p)-\frac{C}{8}\cdot\sqrt{\frac{\log(8n/\delta)}{n}}}\cos\theta$$

$$\geq(1-\cos\theta)\cdot\left(1-\frac{\frac{3}{8}(1-2p)}{\frac{15}{16}\cdot\frac{1-2p}{1-p}}\right)=(1-\cos\theta)\cdot\left(1-\frac{2(1-p)}{5}\right)$$

$$\geq\frac{3(1-\cos\theta)}{5}.\qquad\qquad\text{(G.11)}$$

Here, the inequality is by the condition of $\theta$ in Condition 3.3, $\cos\theta\leq 1$ and $2C\cdot\sqrt{\frac{\log(8n/\delta)}{n}}\leq(1-2p)/(1-p)$. Therefore, we can simplify the inequality of $\langle\mathbf{w}_{+1,r}^{(\tilde{T})},\mathbf{u}\rangle$ in equation G.10 and get that

$$\langle\mathbf{w}_{+1,r}^{(\tilde{T})},\mathbf{u}\rangle\geq\langle\mathbf{w}_{+1,r}^{(\tilde{T}-1)},\mathbf{u}\rangle-\frac{\eta\|\boldsymbol{\mu}\|_2^2}{nm}\sum_{i\in S_{+\mathbf{u},+1}}\ell_i'^{(\tilde{T}-1)}\cdot\left(1-\frac{p}{1-p}-\frac{C}{4}\cdot\sqrt{\frac{\log(8n/\delta)}{n}}\right)\cdot\frac{3(1-\cos\theta)}{5}$$

$$\geq\langle\mathbf{w}_{+1,r}^{(\tilde{T}-1)},\mathbf{u}\rangle-\frac{\eta\|\boldsymbol{\mu}\|_2^2}{nm}\sum_{i\in S_{+\mathbf{u},+1}}\ell_i'^{(\tilde{T}-1)}\cdot\frac{1}{2}\left(1-\frac{p}{1-p}\right)\cdot(1-\cos\theta)$$

$$\geq\langle\mathbf{w}_{+1,r}^{(\tilde{T}-1)},\mathbf{u}\rangle-\frac{\eta\|\boldsymbol{\mu}\|_2^2}{nm}\sum_{i\in S_{+\mathbf{u},+1}}\ell_i'^{(\tilde{T}-1)}\cdot\frac{1-2p}{2(1-p)}\cdot(1-\cos\theta).$$

Here, the first inequality is by equation G.11, and the second inequality is still by $2C\cdot\sqrt{\frac{\log(8n/\delta)}{n}}\leq(1-2p)/(2(1-p))$. Therefore we conclude that

$$\langle\mathbf{w}_{+1,r}^{(\tilde{T})},\mathbf{u}\rangle\geq\langle\mathbf{w}_{+1,r}^{(\tilde{T}-1)},\mathbf{u}\rangle-\frac{\eta\|\boldsymbol{\mu}\|_2^2}{nm}\sum_{i\in S_{+\mathbf{u},+1}}\ell_i'^{(\tilde{T}-1)}\cdot\frac{1-2p}{2(1-p)}\cdot(1-\cos\theta).$$

We conclude that when $\langle\mathbf{w}_{+1,r}^{(0)},\mathbf{u}\rangle>0$,

$$\langle\mathbf{w}_{+1,r}^{(t+1)},\mathbf{u}\rangle\geq\langle\mathbf{w}_{+1,r}^{(t)},\mathbf{u}\rangle-\frac{\eta\|\boldsymbol{\mu}\|_2^2}{nm}\sum_{i\in S_{+\mathbf{u},+1}}\ell_i'^{(t)}\cdot\frac{1-2p}{2(1-p)}\cdot(1-\cos\theta).$$

Similarly, we have

$$\langle\mathbf{w}_{+1,r}^{(t+1)},\mathbf{u}\rangle\leq\langle\mathbf{w}_{+1,r}^{(t)},\mathbf{u}\rangle+\frac{\eta\|\boldsymbol{\mu}\|_2^2}{nm}\cdot\sum_{i\in S_{-\mathbf{u},+1}}\ell_i'^{(t)}\cdot\frac{1-2p}{2(1-p)}\cdot(1-\cos\theta),\quad\langle\mathbf{w}_{+1,r}^{(0)},\mathbf{u}\rangle<0;$$

$$\langle\mathbf{w}_{-1,r}^{(t+1)},\mathbf{v}\rangle\geq\langle\mathbf{w}_{-1,r}^{(t)},\mathbf{v}\rangle-\frac{\eta\|\boldsymbol{\mu}\|_2^2}{nm}\cdot\sum_{i\in S_{+\mathbf{v},-1}}\ell_i'^{(t)}\cdot\frac{1-2p}{2(1-p)}\cdot(1-\cos\theta),\quad\langle\mathbf{w}_{-1,r}^{(0)},\mathbf{v}\rangle>0;$$

$$\langle\mathbf{w}_{-1,r}^{(t+1)},\mathbf{v}\rangle\leq\langle\mathbf{w}_{-1,r}^{(t)},\mathbf{v}\rangle+\frac{\eta\|\boldsymbol{\mu}\|_2^2}{nm}\cdot\sum_{i\in S_{-\mathbf{v},-1}}\ell_i'^{(t)}\cdot\frac{1-2p}{2(1-p)}\cdot(1-\cos\theta),\quad\langle\mathbf{w}_{-1,r}^{(0)},\mathbf{v}\rangle<0.$$

The proof for the first, second and the precise conclusions for lower bound completes.

To prove result 3, it is easy to verify that $|\langle \mathbf{w}_{+1,r}^{(0)}, \mathbf{v}\rangle| \leq |\langle \mathbf{w}_{+1,r}^{(0)}, \mathbf{v}\rangle| + \eta\|\boldsymbol{\mu}\|_2^2/(2m)$, assume that $|\langle \mathbf{w}_{+1,r}^{(t)}, \mathbf{v}\rangle| \leq |\langle \mathbf{w}_{+1,r}^{(0)}, \mathbf{v}\rangle| + \eta\|\boldsymbol{\mu}\|_2^2/(2m)$, we then prove $|\langle \mathbf{w}_{+1,r}^{(t+1)}, \mathbf{v}\rangle| \leq |\langle \mathbf{w}_{+1,r}^{(0)}, \mathbf{v}\rangle| + \eta\|\boldsymbol{\mu}\|_2^2/(2m)$. Recall the update rule, we have

$$
\begin{aligned}
\langle \mathbf{w}_{+1,r}^{(t+1)}, \mathbf{v}\rangle = {} & \langle \mathbf{w}_{+1,r}^{(t)}, \mathbf{v}\rangle - \frac{\eta}{nm} \sum_{i \in S_{+\mathbf{u},+1}\cup S_{-\mathbf{u},-1}} \ell_i'^{(t)} \cdot \mathbf{1}\{\langle \mathbf{w}_{+1,r}^{(t)}, \boldsymbol{\mu}_i\rangle > 0\}\|\boldsymbol{\mu}\|_2^2 \cos\theta \\
& + \frac{\eta}{nm} \sum_{i \in S_{-\mathbf{u},+1}\cup S_{+\mathbf{u},-1}} \ell_i'^{(t)} \cdot \mathbf{1}\{\langle \mathbf{w}_{+1,r}^{(t)}, \boldsymbol{\mu}_i\rangle > 0\}\|\boldsymbol{\mu}\|_2^2 \cos\theta \\
& + \frac{\eta}{nm} \sum_{i \in S_{+\mathbf{v},-1}\cup S_{-\mathbf{v},+1}} \ell_i'^{(t)} \cdot \mathbf{1}\{\langle \mathbf{w}_{+1,r}^{(t)}, \boldsymbol{\mu}_i\rangle > 0\}\|\boldsymbol{\mu}\|_2^2 \\
& - \frac{\eta}{nm} \sum_{i \in S_{-\mathbf{v},-1}\cup S_{+\mathbf{v},+1}} \ell_i'^{(t)} \cdot \mathbf{1}\{\langle \mathbf{w}_{+1,r}^{(t)}, \boldsymbol{\mu}_i\rangle > 0\}\|\boldsymbol{\mu}\|_2^2
\end{aligned}
$$

We prove the case for $\langle \mathbf{w}_{+1,r}^{(t)}, \mathbf{v}\rangle > 0$. The opposite case for $\langle \mathbf{w}_{+1,r}^{(t)}, \mathbf{v}\rangle < 0$ is similar and we thus omit it. If $\langle \mathbf{w}_{+1,r}^{(t)}, \mathbf{v}\rangle > 0$, we immediately get that

$$
\begin{aligned}
\langle \mathbf{w}_{+1,r}^{(t+1)}, \mathbf{v}\rangle \geq {} & \frac{\eta}{nm} \sum_{i \in S_{-\mathbf{u},+1}\cup S_{+\mathbf{u},-1}} \ell_i'^{(t)} \cdot \mathbf{1}\{\langle \mathbf{w}_{+1,r}^{(t)}, \boldsymbol{\mu}_i\rangle > 0\}\|\boldsymbol{\mu}\|_2^2 \cos\theta \\
& + \frac{\eta}{nm} \sum_{i \in S_{+\mathbf{v},-1}\cup S_{-\mathbf{v},+1}} \ell_i'^{(t)} \cdot \mathbf{1}\{\langle \mathbf{w}_{+1,r}^{(t)}, \boldsymbol{\mu}_i\rangle > 0\}\|\boldsymbol{\mu}\|_2^2 \\
\geq {} & -\eta\|\boldsymbol{\mu}\|_2^2/m,
\end{aligned}
$$

where the second inequality is by $-\ell_i'^{(t)} \leq 1$ and Lemma C.2. As for the upper bound, if $\langle \mathbf{w}_{j,r}^{(t)}, \mathbf{u}\rangle < 0$, we have

$$
\begin{aligned}
\langle \mathbf{w}_{+1,r}^{(t+1)}, \mathbf{v}\rangle = {} & \langle \mathbf{w}_{+1,r}^{(t)}, \mathbf{v}\rangle - \frac{\eta\|\boldsymbol{\mu}\|_2^2}{nm} \left( \sum_{i \in S_{-\mathbf{u},-1}} \ell_i'^{(t)} \cos\theta - \sum_{i \in S_{-\mathbf{u},+1}} \ell_i'^{(t)} \cos\theta \right) \\
& - \frac{\eta\|\boldsymbol{\mu}\|_2^2}{nm} \left( - \sum_{i \in S_{+\mathbf{v},-1}} \ell_i'^{(t)} + \sum_{i \in S_{+\mathbf{v},+1}} \ell_i'^{(t)} \right) \\
\leq {} & \langle \mathbf{w}_{+1,r}^{(t)}, \mathbf{v}\rangle,
\end{aligned}
$$

where the inequality is by

$$
\sum_{i \in S_{-\mathbf{u},-1}} \ell_i'^{(t)} \cos\theta - \sum_{i \in S_{-\mathbf{u},+1}} \ell_i'^{(t)} \cos\theta > 0,
$$

$$
- \sum_{i \in S_{+\mathbf{v},-1}} \ell_i'^{(t)} + \sum_{i \in S_{+\mathbf{v},+1}} \ell_i'^{(t)} > 0
$$

from Lemma C.2 and G.16. If $\langle \mathbf{w}_{j,r}^{(t)}, \mathbf{u}\rangle > 0$, the update rule can be written as

$$
\begin{aligned}
\langle \mathbf{w}_{+1,r}^{(t+1)}, \mathbf{v}\rangle = {} & \langle \mathbf{w}_{+1,r}^{(t)}, \mathbf{v}\rangle - \frac{\eta\|\boldsymbol{\mu}\|_2^2}{nm} \left( \sum_{i \in S_{+\mathbf{u},+1}} \ell_i'^{(t)} \cos\theta - \sum_{i \in S_{+\mathbf{u},-1}} \ell_i'^{(t)} \cos\theta \right) \\
& - \frac{\eta\|\boldsymbol{\mu}\|_2^2}{nm} \left( - \sum_{i \in S_{+\mathbf{v},-1}} \ell_i'^{(t)} + \sum_{i \in S_{+\mathbf{v},+1}} \ell_i'^{(t)} \right) \\
= {} & \langle \mathbf{w}_{+1,r}^{(t)}, \mathbf{v}\rangle - \frac{\eta\|\boldsymbol{\mu}\|_2^2 \sum_{i \in S_{+\mathbf{v},-1}} \ell_i'^{(t)}}{nm} \left( \frac{\sum_{i \in S_{+\mathbf{u},+1}} \ell_i'^{(t)}}{\sum_{i \in S_{+\mathbf{v},-1}} \ell_i'^{(t)}} \cdot \cos\theta - \frac{\sum_{i \in S_{+\mathbf{u},-1}} \ell_i'^{(t)}}{\sum_{i \in S_{+\mathbf{v},-1}} \ell_i'^{(t)}} \cdot \cos\theta \right) \\
& - \frac{\eta\|\boldsymbol{\mu}\|_2^2 \sum_{i \in S_{+\mathbf{v},-1}} \ell_i'^{(t)}}{nm} \left( -1 + \frac{\sum_{i \in S_{+\mathbf{v},+1}} \ell_i'^{(t)}}{\sum_{i \in S_{+\mathbf{v},-1}} \ell_i'^{(t)}} \right)
\end{aligned}
$$

$$\leq \langle \mathbf{w}_{+1,r}^{(t)}, \mathbf{v} \rangle - \frac{\eta \|\boldsymbol{\mu}\|_2^2}{nm} \cdot \sum_{i \in S_{+\mathbf{v},-1}} \ell_i'^{(t)} \cdot \left( \frac{1-2p}{1-p} \cdot (\cos\theta - 1) + \frac{C(2\cos\theta + 1)}{8} \sqrt{\frac{\log(8n/\delta)}{n}} \right)$$

$$\leq \langle \mathbf{w}_{+1,r}^{(t)}, \mathbf{v} \rangle.$$

Here, the first inequality is by Lemma G.16, the second inequality is by the condition of $\theta$ in Condition 3.3. We conclude that when $\langle \mathbf{w}_{+1,r}^{(t)}, \mathbf{v} \rangle > 0$, $\left| \langle \mathbf{w}_{+1,r}^{(t+1)}, \mathbf{v} \rangle \right| \leq \left| \langle \mathbf{w}_{+1,r}^{(0)}, \mathbf{v} \rangle \right| + \eta \|\boldsymbol{\mu}\|_2^2/m$. The proof for $\langle \mathbf{w}_{+1,r}^{(t)}, \mathbf{v} \rangle < 0$ is quite similar and we thus omit it. Similarly, we can also prove that $\left| \langle \mathbf{w}_{-1,r}^{(t)}, \mathbf{u} \rangle \right| \leq \left| \langle \mathbf{w}_{-1,r}^{(0)}, \mathbf{u} \rangle \right| + \eta \|\boldsymbol{\mu}\|_2^2/m$. As for the precise conclusions for upper bound, by the update rule, we can easily conclude from $1 + \cos\theta \leq 2$ that

$$\langle \mathbf{w}_{j,r}^{(t+1)}, \mathbf{u} \rangle \leq \langle \mathbf{w}_{j,r}^{(t)}, \mathbf{u} \rangle - \frac{\eta j}{nm} \sum_{i \in S_{+\mathbf{u},+1} \cup S_{-\mathbf{u},-1}} \ell_i'^{(t)} \cdot \mathbf{1}\{\langle \mathbf{w}_{j,r}^{(t)}, \boldsymbol{\mu}_i \rangle > 0\} \|\boldsymbol{\mu}\|_2^2$$

$$- \frac{\eta j}{nm} \sum_{i \in S_{-\mathbf{v},-1} \cup S_{+\mathbf{v},+1}} \ell_i'^{(t)} \cdot \mathbf{1}\{\langle \mathbf{w}_{j,r}^{(t)}, \boldsymbol{\mu}_i \rangle > 0\} \|\boldsymbol{\mu}\|_2^2 \cos\theta$$

$$\leq \langle \mathbf{w}_{+1,r}^{(t)}, \mathbf{u} \rangle - \frac{2\eta \|\boldsymbol{\mu}\|_2^2}{nm} \sum_{i \in [n]} \ell_i'^{(t)}.$$

Similarly, we can get the other conclusions. This completes the proof of Lemma G.17. ∎

The next proposition gives us a precise characterization for $\mathbf{w}_{j,r}^{(t)}$ and $\mathbf{u}, \mathbf{v}$.

**Proposition G.18 (Conclusion 2 in Proposition A.1)** *For* $\langle \mathbf{w}_{+1,r}^{(0)}, \mathbf{u} \rangle, \langle \mathbf{w}_{-1,r}^{(0)}, \mathbf{v} \rangle > 0$ *it holds that*

$$\langle \mathbf{w}_{+1,r}^{(t)}, \mathbf{u} \rangle, \langle \mathbf{w}_{-1,r}^{(t)}, \mathbf{v} \rangle$$

$$\geq -2\sqrt{\log(12m/\delta)} \cdot \sigma_0 \|\boldsymbol{\mu}\|_2 + \frac{n\|\boldsymbol{\mu}\|_2^2(1-\cos\theta)}{20\sigma_p^2 d/(1-2p)} \cdot \log\left( \frac{\eta\sigma_p^2 d}{4nm}(t-1) + \frac{2}{3} \right) - \eta \cdot \|\boldsymbol{\mu}\|_2^2/m.$$

*For* $\langle \mathbf{w}_{+1,r}^{(0)}, \mathbf{u} \rangle, \langle \mathbf{w}_{-1,r}^{(0)}, \mathbf{v} \rangle < 0$, *it holds that*

$$\langle \mathbf{w}_{+1,r}^{(t)}, \mathbf{u} \rangle, \langle \mathbf{w}_{-1,r}^{(t)}, \mathbf{v} \rangle$$

$$\leq 2\sqrt{\log(12m/\delta)} \cdot \sigma_0 \|\boldsymbol{\mu}\|_2 - \frac{n\|\boldsymbol{\mu}\|_2^2(1-\cos\theta)}{20\sigma_p^2 d/(1-2p)} \cdot \log\left( \frac{\eta\sigma_p^2 d}{4nm}(t-1) + \frac{2}{3} \right) + \eta \cdot \|\boldsymbol{\mu}\|_2^2/m.$$

**Proof** [Proof of Proposition G.18] We prove the first case, the other cases are all similar and we thus omit it. By Lemma E.1, when $\langle \mathbf{w}_{+1,r}^{(0)}, \mathbf{u} \rangle > 0$,

$$\langle \mathbf{w}_{+1,r}^{(t+1)}, \mathbf{u} \rangle \geq \langle \mathbf{w}_{+1,r}^{(t)}, \mathbf{u} \rangle - \frac{\eta\|\boldsymbol{\mu}\|_2^2}{2nm} \cdot \sum_{i \in S_{+\mathbf{u},+1}} \ell_i'^{(t)} \cdot (1-\cos\theta) \cdot \frac{1-2p}{1-p}$$

$$\geq \langle \mathbf{w}_{+1,r}^{(t)}, \mathbf{u} \rangle + \frac{\eta\|\boldsymbol{\mu}\|_2^2}{2nm} \cdot \sum_{i \in S_{+\mathbf{u},+1}} \frac{1-\cos\theta}{1 + e^{\frac{1}{m}\sum_{r=1}^m \overline{\rho}_{y_i,r,i}^{(t)} + \kappa/2}} \cdot \frac{1-2p}{1-p}$$

$$\geq \langle \mathbf{w}_{+1,r}^{(t)}, \mathbf{u} \rangle + \frac{\eta\|\boldsymbol{\mu}\|_2^2}{9m} \frac{1-2p}{1 + \overline{b}e^{\overline{x}_t}} \cdot (1-\cos\theta)$$

$$\geq \langle \mathbf{w}_{+1,r}^{(0)}, \mathbf{u} \rangle + \frac{\eta\|\boldsymbol{\mu}\|_2^2}{9m} \cdot \sum_{\tau=0}^t \frac{1-2p}{1 + \overline{b}e^{\overline{x}_\tau}} \cdot (1-\cos\theta)$$

$$\geq \langle \mathbf{w}_{+1,r}^{(0)}, \mathbf{u} \rangle + \frac{\eta\|\boldsymbol{\mu}\|_2^2(1-\cos\theta)}{9m} \int_1^{t-1} \frac{1-2p}{1 + \overline{b}e^{\overline{x}_\tau}} d\tau$$

$$\geq \langle \mathbf{w}_{+1,r}^{(0)}, \mathbf{u} \rangle + \frac{\eta\|\boldsymbol{\mu}\|_2^2(1-\cos\theta)}{9m} \int_1^{t-1} \frac{1-2p}{\overline{c}} d\overline{x}_\tau$$

$$\geq \langle \mathbf{w}_{+1,r}^{(0)}, \mathbf{u} \rangle + \frac{n\|\boldsymbol{\mu}\|_2^2(1-\cos\theta)(1-2p)}{20\sigma_p^2 d} \overline{x}_{t-1} - \frac{\overline{c}n\|\boldsymbol{\mu}\|_2^2(1-\cos\theta)(1-2p)}{10\sigma_p^2 d}$$

$$\geq -\sqrt{2\log(12m/\delta)} \cdot \sigma_0 \|\mathbf{u}\|_2 + \frac{n\|\boldsymbol{\mu}\|_2^2(1-\cos\theta)(1-2p)}{20\sigma_p^2 d} \cdot \overline{x}_{t-1} - \eta \cdot \|\boldsymbol{\mu}\|_2^2/m.$$

Here, the second inequality is by Lemma G.4, the third inequality is by $p < 1/2$ for in Condition 3.3, $|S_{+\mathbf{u},+1}| \geq (1-p)n/4.5$ in Lemma C.2 and Proposition G.6 which gives the bound for the summation over $r$ of $\overline{\rho}_{y_i,r,i}$, the fifth inequality is by the definition of $\overline{x}_t$, the sixth inequality is by $\overline{x}_1 \leq 2\overline{c}$ and the last inequality is by the definition $\overline{c}$ in Proposition G.6 and Lemma C.4. By the definition of $\overline{x}_t$ in Proposition G.6 and results in Lemma G.7, we can easily see that

$$\langle \mathbf{w}_{+1,r}^{(t+1)}, \mathbf{u} \rangle \geq -\sqrt{2\log(12m/\delta)} \cdot \sigma_0 \|\mathbf{u}\|_2 + \frac{n\|\boldsymbol{\mu}\|_2^2(1-\cos\theta)}{20\sigma_p^2 d/(1-2p)} \cdot \log\left(\frac{\eta\sigma_p^2 d}{4nm}(t-1) + \frac{2}{3}\right) - \eta \cdot \|\boldsymbol{\mu}\|_2^2/m$$

The proof of Proposition G.18 completes. ∎

We give the upper bound of the inner product of the filter and signal component.

**Proposition G.19** *For $\langle \mathbf{w}_{+1,r}^{(0)}, \mathbf{u} \rangle, \langle \mathbf{w}_{-1,r}^{(0)}, \mathbf{v} \rangle > 0$ it holds that*

$$\langle \mathbf{w}_{+1,r}^{(t)}, \mathbf{u} \rangle, \langle \mathbf{w}_{-1,r}^{(t)}, \mathbf{v} \rangle \leq 2\sqrt{\log(12m/\delta)} \cdot \sigma_0 \|\boldsymbol{\mu}\|_2 + \frac{5n\|\boldsymbol{\mu}\|_2^2}{\sigma_p^2 d} \cdot \log\left(\frac{\eta\sigma_p^2 d}{2nm}t + 1\right).$$

*For $\langle \mathbf{w}_{+1,r}^{(0)}, \mathbf{u} \rangle, \langle \mathbf{w}_{-1,r}^{(0)}, \mathbf{v} \rangle < 0$ it holds that*

$$\langle \mathbf{w}_{+1,r}^{(t)}, \mathbf{u} \rangle, \langle \mathbf{w}_{-1,r}^{(t)}, \mathbf{v} \rangle \geq -2\sqrt{\log(12m/\delta)} \cdot \sigma_0 \|\boldsymbol{\mu}\|_2 - \frac{5n\|\boldsymbol{\mu}\|_2^2}{\sigma_p^2 d} \cdot \log\left(\frac{\eta\sigma_p^2 d}{2nm}t + 1\right).$$

**Proof** [Proof of Proposition G.19] We prove the case when $\langle \mathbf{w}_{+1,r}^{(0)}, \mathbf{u} \rangle > 0$, the other cases are all similar and we thus omit it. By Lemma G.17, when $\langle \mathbf{w}_{+1,r}^{(0)}, \mathbf{u} \rangle > 0$,

$$\begin{aligned}
\langle \mathbf{w}_{+1,r}^{(t+1)}, \mathbf{u} \rangle &\leq \langle \mathbf{w}_{+1,r}^{(t)}, \mathbf{u} \rangle - \frac{2\eta\|\boldsymbol{\mu}\|_2^2}{nm} \cdot \sum_{i \in [n]} \ell_i'^{(t)} \\
&\leq \langle \mathbf{w}_{+1,r}^{(t)}, \mathbf{u} \rangle + \frac{2\eta\|\boldsymbol{\mu}\|_2^2}{nm} \cdot \sum_{i \in [n]} \frac{1}{1 + e^{\frac{1}{m}\sum_{r=1}^m \overline{\rho}_{y_i,r,i} - \kappa/2}} \\
&\leq \langle \mathbf{w}_{+1,r}^{(0)}, \mathbf{u} \rangle + \frac{2\eta\|\boldsymbol{\mu}\|_2^2}{m} \cdot \sum_{\tau=0}^{t} \frac{1}{1 + \underline{b}e^{\underline{x}_\tau}} \\
&\leq \langle \mathbf{w}_{+1,r}^{(0)}, \mathbf{u} \rangle + \frac{2\eta\|\boldsymbol{\mu}\|_2^2}{m} \int_0^t \frac{1}{1 + \underline{b}e^{\underline{x}_\tau}} d\tau \\
&\leq \langle \mathbf{w}_{+1,r}^{(0)}, \mathbf{u} \rangle + \frac{2\eta\|\boldsymbol{\mu}\|_2^2}{m} \int_0^t \frac{1}{\underline{c}} d\underline{x}_\tau \\
&\leq \sqrt{2\log(12m/\delta)} \cdot \sigma_0 \|\boldsymbol{\mu}\|_2 + \frac{5n\|\boldsymbol{\mu}\|_2^2}{\sigma_p^2 d} \cdot \underline{x}_t.
\end{aligned}$$

Here, the second inequality is by the output bound in Lemma G.4, the third inequality is by Proposition G.6, the fifth inequality is by the definition of $\underline{x}_t$, and the last inequality is by the definition of $\underline{c}$ in Proposition G.6 and Lemma C.4. By the results in Lemma G.7, we have

$$\underline{x}_t \leq \log\left(\frac{\eta\sigma_p^2 d}{2nm}t + 1\right).$$

The proof of Proposition G.19 completes. ∎

For the noise memorization, we give the following lemmas and propositions.

**Proposition G.20** *Let $\overline{c}$ and $\overline{x}_t$ be defined in Proposition G.6, then it holds that*

$$3n\underline{x}_t \geq \sum_{i=1}^{n} \overline{\rho}_{j,r,i}^{(t)} \geq \frac{n}{5} \cdot (\overline{x}_{t-1} - \overline{x}_1)$$

*for all $t \in [T^*]$ and $r \in [m]$.*

**Proof** [Proof of Proposition G.20] Recall the update rule for $\overline{\rho}_{j,r,i}^{(t+1)}$, that we have

$$\overline{\rho}_{j,r,i}^{(t+1)} = \overline{\rho}_{j,r,i}^{(t)} - \frac{\eta}{nm} \cdot \ell_i'^{(t)} \cdot \mathbf{1}\big(\langle \mathbf{w}_{j,r}^{(t)}, \boldsymbol{\xi}_i \rangle \geq 0\big) \cdot \mathbf{1}\left(y_i = j\right) \|\boldsymbol{\xi}_i\|_2^2.$$

Hence

$$
\begin{aligned}
\sum_i \overline{\rho}_{y_i,r,i}^{(t)} &\geq \sum_i \overline{\rho}_{y_i,r,i}^{(t-1)} + \frac{\eta}{nm} \cdot \frac{1}{1 + \overline{b}\overline{x}_{t-1}} \cdot |S_{j,r}^{(0)}| \cdot \|\boldsymbol{\xi}_i\|_2^2 \\
&= \sum_{\tau=1}^{t-1} \frac{\eta}{nm} \cdot \frac{1}{1 + \overline{b}\overline{x}_{\tau-1}} \cdot |S_{j,r}^{(0)}| \cdot \|\boldsymbol{\xi}_i\|_2^2 \\
&\geq \frac{\eta|S_{j,r}^{(0)}| \cdot \|\boldsymbol{\xi}_i\|_2^2}{nm} \int_2^t \frac{1}{1 + \overline{b}\overline{x}_{\tau-1}} d\tau = \frac{\eta|S_{j,r}^{(0)}| \cdot \|\boldsymbol{\xi}_i\|_2^2}{\overline{c}nm} \cdot (\overline{x}_{t-1} - \overline{x}_1) \\
&\geq \frac{n}{5} \cdot (\overline{x}_{t-1} - \overline{x}_1).
\end{aligned}
$$

Here, the first equality is by $|S_{j,r}^{(t)}| = |S_{j,r}^{(0)}|$, the first inequality is by Proposition G.6 and the last inequality is by the definition of $\overline{c}$. Similarly, we have

$$
\begin{aligned}
\sum_i \overline{\rho}_{y_i,r,i}^{(t)} &\leq \sum_i \overline{\rho}_{y_i,r,i}^{(t-1)} + \frac{\eta}{nm} \cdot \frac{1}{1 + \underline{b} \cdot \underline{x}_{t-1}} \cdot |S_{j,r}^{(0)}| \cdot \|\boldsymbol{\xi}_i\|_2^2 \\
&\leq \frac{\eta|S_{j,r}^{(0)}| \cdot \|\boldsymbol{\xi}_i\|_2^2}{nm} \int_2^t \frac{1}{1 + \underline{b} \cdot \underline{x}_{\tau-1}} d\tau \\
&\leq 3n\underline{x}_t.
\end{aligned}
$$

This completes the proof of Proposition G.20. ∎

The next proposition gives the $L_2$ norm of $\mathbf{w}_{j,r}^{(t)}$.

**Proposition G.21** *Under Condition 3.3, for $t \in [T^*]$, it holds that*

$$\big\|\mathbf{w}_{j,r}^{(t)}\big\|_2 = \Theta(\sigma_0\sqrt{d}).$$

**Proof** [Proof of Proposition G.21] We first handle the noise memorization part, and get that

$$
\begin{aligned}
&\left\|\sum_{i=1}^{n} \rho_{j,r,i}^{(t)} \cdot \|\boldsymbol{\xi}_i\|_2^{-2} \cdot \boldsymbol{\xi}_i\right\|_2^2 \\
&= \sum_{i=1}^{n} \rho_{j,r,i}^{(t)} \cdot \|\boldsymbol{\xi}_i\|_2^{-2} + 2 \sum_{1 \leq i_1 < i_2 \leq n} \rho_{j,r,i_1}^{(t)} \rho_{j,r,i_2}^{(t)} \cdot \|\boldsymbol{\xi}_{i_1}\|_2^{-2} \cdot \|\boldsymbol{\xi}_{i_2}\|_2^{-2} \cdot \langle \boldsymbol{\xi}_{i_1}, \boldsymbol{\xi}_{i_2} \rangle \\
&= 4\Theta\left(\sigma_p^{-2} d^{-1} \sum_{i=1}^{n} \rho_{j,r,i}^{(t)}{}^2 + 2 \sum_{1 \leq i_1 < i_2 \leq n} \rho_{j,r,i_1}^{(t)} \rho_{j,r,i_2}^{(t)} \cdot \left(16\sigma_p^{-4} d^{-2}\right) \cdot \left(2\sigma_p^2 \sqrt{d \log\left(6n^2/\delta\right)}\right)\right) \\
&= \Theta\left(\sigma_p^{-2} d^{-1}\right) \sum_{i=1}^{n} \rho_{j,r,i}^{(t)} + \widetilde{\Theta}\left(\sigma_p^{-2} d^{-3/2}\right) \left(\sum_{i=1}^{n} \rho_{j,r,i}^{(t)}\right)^2 \\
&= \Theta\left(\sigma_p^{-2} d^{-1} n^{-1}\right) \left(\sum_{i=1}^{n} \overline{\rho}_{j,r,i}^{(t)}\right)^2 = o(\sigma_0^2 d),
\end{aligned}
$$

where the first quality is by Lemma C.3; for the second to second last equation we plugged in coefficient orders. The last equality is by the definition of $\sigma_0$ and Condition 3.3. Moreover, we have

$$\frac{\left|\langle \mathbf{w}_{j,r}^{(t)}, \mathbf{u}\rangle\right| \cdot \|\boldsymbol{\mu}\|_2^{-1}}{\Theta\left(\sigma_0\sqrt{d}\right)} \leq \widetilde{\Theta}\left(\frac{n\|\boldsymbol{\mu}\|_2}{\sigma_p^2 d(\sigma_0\sqrt{d})}\right) \leq \Theta\left(\frac{\sqrt{n}}{\sigma_p\sqrt{d}\cdot\sigma_0\sqrt{d}}\right) = o(1),$$

where the first inequality is by Lemma G.19 and $\sigma_0\|\boldsymbol{\mu}\|_2 \ll n\|\boldsymbol{\mu}\|_2^2/(\sigma_p^2 d)$. The second inequality utilizes $\frac{n\|\boldsymbol{\mu}\|_2}{\sigma_p^2 d} \ll \frac{\sqrt{n}}{\sigma_p\sqrt{d}}$, the last equality is due to $\sigma_0 = \frac{nm}{\sigma_p d\delta}\cdot \text{polylog}(d)$. Similarly we have $\left|\langle \mathbf{w}_{j,r}^{(t)}, \mathbf{u}\rangle\right| \cdot \|\boldsymbol{\mu}\|_2^{-1} = o(\sigma_0\sqrt{d})$. Moreover, we have that $\left|\langle \mathbf{w}_{j,r}^{(0)}, \mathbf{u}\rangle\right| \cdot \|\boldsymbol{\mu}\|_2^{-1} \leq \widetilde{\Theta}(\sigma_0) = o(\sigma_0\sqrt{d})$. We can thus bound the norm of $\mathbf{w}_{j,r}^{(t)}$ as:

$$\left\|\mathbf{w}_{j,r}^{(t)}\right\|_2 = \|\mathbf{w}_{j,r}^{(0)}\|_2 \pm o(\sigma_0\sqrt{d}) = \Theta(\sigma_0\sqrt{d}). \tag{G.12}$$

The second equality is by Lemma C.4. This completes the proof of Proposition G.21. ∎

## H    PROOF OF THEOREM 3.4

We prove Theorem 3.4 in this section. Here, we let $\delta = 1/\text{polylog}(d)$. First of all, we prove the convergence of training. For the training convergence, Lemma G.4, Proposition G.6 and Lemma G.7 show that

$$\begin{aligned} y_i f(\mathbf{W}^{(t)}, \mathbf{x}_i) &\geq -\frac{\kappa}{2} + \frac{1}{m}\sum_{r=1}^{m} \overline{\rho}_{y_i,r,i}^{(t)} \\ &\geq -\frac{\kappa}{2} + \underline{x}_t \\ &\geq -\frac{\kappa}{2} + \log\left(\frac{\eta\sigma_p^2 d}{nm}t + \frac{2}{3}\right) \\ &\geq -\kappa + \log\left(\frac{\eta\sigma_p^2 d}{4nm}t + \frac{2}{3}\right). \end{aligned}$$

$\kappa$ is defined in equation G.3. Here, the first inequality is by the conclusion in Lemma G.4 and the second and third inequalities are by $\underline{x}_t \geq \log(\underline{c}t/\underline{b} + 1) \geq \log\left(\frac{\eta\sigma_p^2 d}{4nm}t + \frac{2}{3}\right)$ in Proposition G.6, the last inequality is by the definition of $\kappa$ in equation G.3. Therefore we have

$$L(\mathbf{W}^{(t)}) \leq \log\left(1 + \exp\{\kappa\}/\left(\frac{\eta\sigma_p^2 d}{4nm}t + \frac{2}{3}\right)\right) \leq \frac{e^\kappa}{\frac{\eta\sigma_p^2 d}{4nm}t + \frac{2}{3}}.$$

The last inequality is by $\log(1 + x) \leq x$. When $t \geq \frac{2nm}{\eta\sigma_p^2 d\varepsilon}$, we conclude that

$$L(\mathbf{W}^{(t)}) \leq \frac{e^\kappa}{\frac{\eta\sigma_p^2 d}{4nm}t + \frac{2}{3}} \leq \frac{e^\kappa}{2/\varepsilon + \frac{2}{3}} \leq \varepsilon.$$

Here, the last inequality is by $e^\kappa \leq 1.5$. This completes the proof for the convergence of training loss.

As for the second conclusion, it is easy to see that

$$P(yf(\mathbf{W}^{(t)}, \mathbf{x}) > 0) = \sum_{\boldsymbol{\mu}\in\{\pm\mathbf{u},\pm\mathbf{v}\}} P(yf(\mathbf{W}^{(t)}, \mathbf{x}) > 0|\mathbf{x}_{\text{signal part}} = \boldsymbol{\mu}) \cdot \frac{1}{4}, \tag{H.1}$$

without loss of generality we can assume that the test data $\mathbf{x} = (\mathbf{u}^\top, \boldsymbol{\xi}^\top)^\top$, for $\mathbf{x} = (-\mathbf{u}^\top, \boldsymbol{\xi}^\top)^\top$, $\mathbf{x} = (\mathbf{v}^\top, \boldsymbol{\xi}^\top)^\top$ and $\mathbf{x} = (-\mathbf{v}^\top, \boldsymbol{\xi}^\top)^\top$ the proof is all similar and we omit it. We investigate

$$P(yf(\mathbf{W}^{(t)}, \mathbf{x}) > 0|\mathbf{x}_{\text{signal part}} = \mathbf{u}).$$

When $\mathbf{x} = (\mathbf{u}^\top, \boldsymbol{\xi}^\top)^\top$, the true label $\overline{y} = +1$. We remind that the true label for $\mathbf{x}$ is $\overline{y}$, and the observed label is $y$. Therefore we have

$$
\begin{aligned}
P(yf(\mathbf{W}^{(t)}, \mathbf{x}) < 0 | \mathbf{x}_{\text{signal part}} = \mathbf{u}) &= P(\overline{y}f(\mathbf{W}^{(t)}, \mathbf{x}) < 0, \overline{y} = y | \mathbf{x}_{\text{signal part}} = \mathbf{u}) \\
&\quad + P(\overline{y}f(\mathbf{W}^{(t)}, \mathbf{x}) > 0, \overline{y} \neq y | \mathbf{x}_{\text{signal part}} = \mathbf{u}) \\
&\leq p + P(\overline{y}f(\mathbf{W}^{(t)}, \mathbf{x}) < 0 | \mathbf{x}_{\text{signal part}} = \mathbf{u}).
\end{aligned}
$$

It therefore suffices to provide an upper bound for $P(\overline{y}f(\mathbf{W}^{(t)}, \mathbf{x}) < 0 | \mathbf{x}_{\text{signal part}} = \mathbf{u})$. Note that for any test data with the conditioned event $\mathbf{x}_{\text{signal part}} = \mathbf{u}$, it holds that

$$
\overline{y}f(\mathbf{W}^{(t)}, \mathbf{x}) = \frac{1}{m}\sum_{r=1}^{m} F_{+1,r}(\mathbf{W}^{(t)}, \mathbf{u}) + F_{+1,r}(\mathbf{W}^{(t)}, \boldsymbol{\xi}) - \frac{1}{m}\sum_{r=1}^{m}\left(F_{-1,r}(\mathbf{W}^{(t)}, \mathbf{u}) + F_{-1,r}(\mathbf{W}^{(t)}, \boldsymbol{\xi})\right).
$$

We have for $t \geq \Omega(nm/(\eta\sigma_p^2 d))$, $\overline{x}_t \geq C > 0$, and

$$
\begin{aligned}
\overline{y}f(\mathbf{W}^{(t)}, \mathbf{x}) &\geq \frac{1}{m}\sum_{r=1}^{m}\texttt{ReLU}(\langle \mathbf{w}_{+1,r}^{(t)}, \mathbf{u}\rangle) - \frac{1}{m}\sum_{r=1}^{m}\texttt{ReLU}(\langle \mathbf{w}_{-1,r}^{(t)}, \boldsymbol{\xi}\rangle) - \frac{1}{m}\sum_{r=1}^{m}\texttt{ReLU}(\langle \mathbf{w}_{-1,r}^{(t)}, \mathbf{u}\rangle) \\
&\geq -2\sqrt{\log(12m/\delta)}\cdot\sigma_0\|\boldsymbol{\mu}\|_2 + \frac{n\|\boldsymbol{\mu}\|_2^2(1-\cos\theta)(1-2p)}{60\sigma_p^2 d}\cdot\overline{x}_{t-1} - \frac{\eta\|\boldsymbol{\mu}\|_2^2}{m} \\
&\quad - \frac{1}{m}\sum_{r=1}^{m}\texttt{ReLU}(\langle \mathbf{w}_{-1,r}^{(t)}, \boldsymbol{\xi}\rangle) - \frac{1}{m}\sum_{r=1}^{m}\texttt{ReLU}(\langle \mathbf{w}_{-1,r}^{(t)}, \mathbf{u}\rangle),
\end{aligned}
$$

where the first inequality is by $F_{y,r}(\mathbf{W}^{(t)}, \boldsymbol{\xi}) \geq 0$, and the second inequality is by Proposition G.18 and $|\{r \in [m], \langle \mathbf{w}_{+1,r}^{(0)}, \mathbf{u}\rangle > 0\}|/m \geq 1/3$. Then for $t \geq T = \Omega(nm/(\eta\sigma_p^2 d))$, $\overline{x}_t \geq \underline{x}_t \geq C > 0$, it holds that

$$
\begin{aligned}
\overline{y}f(\mathbf{W}^{(t)}, \mathbf{x}) &\geq \frac{n\|\boldsymbol{\mu}\|_2^2(1-\cos\theta)(1-2p)}{60\sigma_p^2 d}\cdot\overline{x}_{t-1} - \frac{1}{m}\sum_{r=1}^{m}\texttt{ReLU}(\langle \mathbf{w}_{-1,r}^{(t)}, \boldsymbol{\xi}\rangle) - \frac{1}{m}\sum_{r=1}^{m}\texttt{ReLU}(\langle \mathbf{w}_{-1,r}^{(t)}, \mathbf{u}\rangle) \\
&\quad - 2\sqrt{\log(12mn/\delta)}\cdot\sigma_0\|\boldsymbol{\mu}\|_2 \\
&\geq \frac{n\|\boldsymbol{\mu}\|_2^2(1-\cos\theta)(1-2p)}{60\sigma_p^2 d}\cdot\overline{x}_{t-1} - \frac{1}{m}\sum_{r=1}^{m}\texttt{ReLU}(\langle \mathbf{w}_{-1,r}^{(t)}, \boldsymbol{\xi}\rangle) \\
&\quad - 4\sqrt{\log(12mn/\delta)}\cdot\sigma_0\|\boldsymbol{\mu}\|_2 - 2\eta\|\boldsymbol{\mu}\|_2^2/m \\
&\geq \frac{n\|\boldsymbol{\mu}\|_2^2(1-\cos\theta)(1-2p)}{100\sigma_p^2 d}\cdot\overline{x}_{t-1} - \frac{1}{m}\sum_{r=1}^{m}\texttt{ReLU}(\langle \mathbf{w}_{-1,r}^{(t)}, \boldsymbol{\xi}\rangle). \quad\quad (\text{H.2})
\end{aligned}
$$

Here, the first inequality is by the condition of $\sigma_0$, $\eta$ in Condition 3.3, and $\overline{x}_{t-1} \geq C$, the second inequality is by the third conclusion in Lemma G.17 and the third inequality is still by the condition of $\sigma_0$, $\eta$ in Condition 3.3. We have $\sigma_0\|\boldsymbol{\mu}\|_2 \leq \widetilde{O}(n\|\boldsymbol{\mu}\|_2^2/(\sigma_p^2 d))$. We denote by $h(\boldsymbol{\xi}) = \frac{1}{m}\sum_{r=1}^{m}\texttt{ReLU}(\langle \mathbf{w}_{-1,r}^{(t)}, \boldsymbol{\xi}\rangle)$. By Theorem 5.2.2 in Vershynin (2018), we have

$$
P(h(\boldsymbol{\xi}) - \mathbb{E}h(\boldsymbol{\xi}) \geq x) \leq \exp\left(-\frac{cx^2}{\sigma_p^2\|h\|_{\text{Lip}}^2}\right). \quad\quad (\text{H.3})
$$

By $\|\boldsymbol{\mu}\|_2^4(1-\cos\theta)^2 \geq \widetilde{\Omega}(m^2\sigma_p^4 d)$ and Proposition G.21, we directly have

$$
\frac{n\|\boldsymbol{\mu}\|_2^2(1-\cos\theta)(1-2p)}{100\sigma_p^2 d}\cdot\overline{x}_{t-1} \geq \mathbb{E}h(\boldsymbol{\xi}) = \frac{\sigma_p}{\sqrt{2\pi}m}\sum_{r=1}^{m}\|\mathbf{w}_{-\widehat{y},r}^{(t)}\|_2 = \Theta(\sigma_p\sigma_0\sqrt{d}).
$$

Here, the inequality is by the following equation under the condition $\|\boldsymbol{\mu}\|_2^4(1-\cos\theta)^2 \geq \widetilde{\Omega}(m^2\sigma_p^4 d)$:

$$
\frac{n\|\boldsymbol{\mu}\|_2^2(1-\cos\theta)}{\sigma_p^3 d^{3/2}\sigma_0} \geq \widetilde{\Omega}\left(\frac{\|\boldsymbol{\mu}\|_2^2(1-\cos\theta)}{\sigma_p^2\sqrt{d}m}\right) \gg 1.
$$

Now using methods in equation H.3 we get that

$$P(\overline{y}f(\mathbf{W}^{(t)}, \mathbf{x}) < 0 | \mathbf{x}_{\text{signal part}} = \mathbf{u}) \leq P\left( h(\boldsymbol{\xi}) - \mathbb{E}h(\boldsymbol{\xi}) \geq \sum_r \sigma(\langle \mathbf{w}_{+1,r}^{(t)}, \mathbf{u} \rangle)/m - \frac{\sigma_p}{\sqrt{2\pi}m} \sum_{r=1}^m \left\| \mathbf{w}_{-1,r}^{(t)} \right\|_2 \right)$$

$$\leq \exp\left[ -\frac{c\left( \sum_r \sigma(\langle \mathbf{w}_{+1,r}^{(t)}, \mathbf{u} \rangle)/m - (\sigma_p/(m\sqrt{2\pi})) \sum_{r=1}^m \left\| \mathbf{w}_{-1,r}^{(t)} \right\|_2 \right)^2}{\sigma_p^2 \left( \sum_{r=1}^m \left\| \mathbf{w}_{-1,r}^{(t)} \right\|_2/m \right)^2} \right]$$

$$\leq \exp\{ -Cn^2 \|\boldsymbol{\mu}\|_2^4 (1 - \cos\theta)^2/(\sigma_p^4 d^2 \cdot \sigma_p^2 \sigma_0^2 d) \}$$

$$\leq \exp\{ -C\|\boldsymbol{\mu}\|_2^4 (1 - \cos\theta)^2/(m^2 \sigma_p^4 d \cdot \text{polylog}(d)) \} = o(1).$$

Here, $C = O(1)$ is some constant. The first inequality is directly by equation H.2, the second inequality is by equation H.3 and the third inequality is by Proposition G.18 which directly gives the lower bound of signal learning, and Proposition G.21 which directly gives the scale of $\left\| \mathbf{w}_{-1,r}^{(t)} \right\|_2$. The last inequality is by $\|\boldsymbol{\mu}\|_2^4 (1 - \cos\theta)^2 \geq \widetilde{\Omega}(m^2 \sigma_p^4 d)$. We can similarly get the inequality on the condition $\mathbf{x}_{\text{signal part}} = -\mathbf{u}$ and $\mathbf{x}_{\text{signal part}} = \pm\mathbf{v}$. Combined the results with equation H.1, we have

$$P(yf(\mathbf{W}^{(t)}, \mathbf{x}) < 0) \leq p + o(1).$$

for all $t \geq \Omega(nm/(\eta\sigma_p^2 d))$. Here, constant $C$ in different inequalities is different.

For the proof of the third conclusion in Theorem 3.4, we have

$$P(yf(\mathbf{W}^{(t)}, \mathbf{x}) < 0) = P(\overline{y}f(\mathbf{W}^{(t)}, \mathbf{x}) < 0, \overline{y} = y) + P(\overline{y}f(\mathbf{W}^{(t)}, \mathbf{x}) > 0, \overline{y} \neq y)$$

$$= p + (1 - 2p)P(\overline{y}f(\mathbf{W}^{(t)}, \mathbf{x}) \leq 0). \tag{H.4}$$

We investigate the probability $P(\overline{y}f(\mathbf{W}^{(t)}, \mathbf{x}) \leq 0)$, and have

$$P(\overline{y}f(\mathbf{W}^{(t)}, \mathbf{x}) \leq 0)$$

$$= P\left( \sum_r \sigma(\langle \mathbf{w}_{-\overline{y},r}^{(t)}, \boldsymbol{\xi} \rangle) - \sum_r \sigma(\langle \mathbf{w}_{\overline{y},r}^{(t)}, \boldsymbol{\xi} \rangle) \geq \sum_r \sigma(\langle \mathbf{w}_{\overline{y},r}^{(t)}, \boldsymbol{\mu} \rangle) - \sum_r \sigma(\langle \mathbf{w}_{-\overline{y},r}^{(t)}, \boldsymbol{\mu} \rangle) \right).$$

Here, $\boldsymbol{\mu}$ is the signal part of the test data. Define $g(\boldsymbol{\xi}) = \left( \frac{1}{m} \sum_{r=1}^m \text{ReLU}(\langle \mathbf{w}_{+1,r}^{(t)}, \boldsymbol{\xi} \rangle) - \frac{1}{m} \sum_{r=1}^m \text{ReLU}(\langle \mathbf{w}_{-1,r}^{(t)}, \boldsymbol{\xi} \rangle) \right)$, it is easy to see that

$$P(\overline{y}f(\mathbf{W}^{(t)}, \mathbf{x}) \leq 0) \geq 0.5P\left( |g(\boldsymbol{\xi})| \geq \max\left\{ \sum_r \sigma(\langle \mathbf{w}_{\overline{y},r}^{(t)}, \boldsymbol{\mu} \rangle)/m, \sum_r \sigma(\langle \mathbf{w}_{-\overline{y},r}^{(t)}, \boldsymbol{\mu} \rangle)/m \right\} \right), \tag{H.5}$$

since if $|g(\boldsymbol{\xi})|$ is large, we can always select $\overline{y}$ given $\boldsymbol{\xi}$ to make a wrong prediction. Define the set

$$\boldsymbol{\Omega} = \left\{ \boldsymbol{\xi} : |g(\boldsymbol{\xi})| \geq \max\left\{ \sum_r \sigma(\langle \mathbf{w}_{+1,r}^{(t)}, \boldsymbol{\mu} \rangle)/m, \sum_r \sigma(\langle \mathbf{w}_{-1,r}^{(t)}, \boldsymbol{\mu} \rangle)/m \right\} \right\},$$

it remains for us to proceed $P(\boldsymbol{\Omega})$. To further proceed, we prove that there exists a fixed vector $\boldsymbol{\zeta}$ with $\|\boldsymbol{\zeta}\|_2 \leq 0.02\sigma_p$, such that

$$\sum_{j' \in \{\pm 1\}} \left[ g(j'\boldsymbol{\xi} + \boldsymbol{\zeta}) - g(j'\boldsymbol{\xi}) \right] \geq 4 \max\left\{ \sum_r \sigma(\langle \mathbf{w}_{+1,r}^{(t)}, \boldsymbol{\mu} \rangle)/m, \sum_r \sigma(\langle \mathbf{w}_{-1,r}^{(t)}, \boldsymbol{\mu} \rangle)/m \right\}.$$

Here, $\boldsymbol{\mu} \in \{\pm\mathbf{u}, \pm\mathbf{v}\}$ is the signal part of test data. If so, we can see that there must exist at least one of $\boldsymbol{\xi}, \boldsymbol{\xi} + \boldsymbol{\zeta}, -\boldsymbol{\xi} + \boldsymbol{\zeta}$ and $-\boldsymbol{\xi}$ that belongs to $\boldsymbol{\Omega}$. We immediately get that

$$P\big( \boldsymbol{\Omega} \cup (-\boldsymbol{\Omega}) \cup (\boldsymbol{\Omega} - \boldsymbol{\zeta}) \cup (-\boldsymbol{\Omega} - \boldsymbol{\zeta}) \big) = 1.$$

Then we can see that there exists at least one of $\boldsymbol{\Omega}, -\boldsymbol{\Omega}, \boldsymbol{\Omega} - \boldsymbol{\zeta}, -\boldsymbol{\Omega} - \boldsymbol{\zeta}$ such that the probability is larger than 0.25. Also note that $\|\boldsymbol{\zeta}\|_2 \leq 0.02\sigma_p$, we have

$$|P(\boldsymbol{\Omega}) - P(\boldsymbol{\Omega} - \boldsymbol{\zeta})| = \left| P_{\boldsymbol{\xi} \sim \mathcal{N}(0, \sigma_p^2 \mathbf{I}_d)}(\boldsymbol{\xi} \in \boldsymbol{\Omega}) - P_{\boldsymbol{\xi} \sim \mathcal{N}(\boldsymbol{\zeta}, \sigma_p^2 \mathbf{I}_d)}(\boldsymbol{\xi} \in \boldsymbol{\Omega}) \right|$$

$$\leq \frac{\|\boldsymbol{\zeta}\|_2}{2\sigma_p} \leq 0.01.$$

Here, the first inequality is by Proposition 2.1 in Devroye et al. (2018) and the second inequality is by the condition $\|\boldsymbol{\zeta}\|_2 \leq 0.02\sigma_p$. Combined with $P(\boldsymbol{\Omega}) = P(-\boldsymbol{\Omega})$, we conclude that $P(\boldsymbol{\Omega}) \geq 0.24$. We can obtain from equation H.4 that

$$P(yf(\mathbf{W}^{(t)}, \mathbf{x}) < 0) = p + (1 - 2p)P(\overline{y}f(\mathbf{W}^{(t)}, \mathbf{x}) \leq 0)$$
$$\geq p + (1 - 2p) * 0.12 \geq 0.76p + 0.12 \geq p + C_4$$

if we find the existence of $\boldsymbol{\zeta}$. Here, the last inequality is by $p < 1/2$ is a fixed value.

The only thing remained for us is to prove the existence of $\boldsymbol{\zeta}$ with $\|\boldsymbol{\zeta}\|_2 \leq 0.02\sigma_p$ such that

$$\sum_{j' \in \{\pm 1\}} \left[ g(j'\boldsymbol{\xi} + \boldsymbol{\zeta}) - g(j'\boldsymbol{\xi}) \right] \geq 4 \max \left\{ \sum_r \sigma(\langle \mathbf{w}_{+1,r}^{(t)}, \boldsymbol{\mu} \rangle)/m, \sum_r \sigma(\langle \mathbf{w}_{-1,r}^{(t)}, \boldsymbol{\mu} \rangle)/m \right\}.$$

The existence of $\boldsymbol{\zeta}$ is proved by the following construction:

$$\boldsymbol{\zeta} = \lambda \cdot \sum_r \mathbf{w}_{+1,r}^{(0)},$$

where $\lambda = c\sigma_p/(\sqrt{\sigma_0^2 md})$ for some small constant $c$. It is worth noting that

$$\|\boldsymbol{\zeta}\|_2 = \frac{c\sigma_p}{\sqrt{\sigma_0^2 md}} \cdot \sqrt{m \cdot \sigma_0^2 d} \cdot (1 \pm o(1)) \leq 0.02\sigma_p,$$

where the first equality is by the concentration. By the construction of $\boldsymbol{\zeta}$, we have almost surely that

$$\sigma(\langle \mathbf{w}_{+1,r}^{(t)}, \boldsymbol{\xi} + \boldsymbol{\zeta} \rangle) - \sigma(\langle \mathbf{w}_{+1,r}^{(t)}, \boldsymbol{\xi} \rangle) + \sigma(\langle \mathbf{w}_{+1,r}^{(t)}, -\boldsymbol{\xi} + \boldsymbol{\zeta} \rangle) - \sigma(\langle \mathbf{w}_{+1,r}^{(t)}, -\boldsymbol{\xi} \rangle)$$
$$\geq \sigma'(\langle \mathbf{w}_{+1,r}^{(t)}, \boldsymbol{\xi} \rangle) \langle \mathbf{w}_{+1,r}^{(t)}, \boldsymbol{\zeta} \rangle + \sigma'(\langle \mathbf{w}_{+1,r}^{(t)}, -\boldsymbol{\xi} \rangle) \langle \mathbf{w}_{+1,r}^{(t)}, \boldsymbol{\zeta} \rangle$$
$$\geq \langle \mathbf{w}_{+1,r}^{(t)}, \boldsymbol{\zeta} \rangle$$
$$\geq \lambda \left[ \sigma_0^2 d - 2m\sqrt{\log(12mn/\delta)} \cdot \sigma_0^2 \sqrt{d} - \widetilde{\Omega}(nm\sigma_0/(\sigma_p\sqrt{d})) \right]$$
$$\geq \lambda \sigma_0^2 d \cdot (1 - o(1)), \tag{H.6}$$

where the first inequality is by the convexity of ReLU, the second last inequality is by Lemma C.4 and G.2, and the last inequality is by the definition of $\sigma_0$. For the inequality with $j = -1$, we have

$$\sigma(\langle \mathbf{w}_{-1,r}^{(t)}, \boldsymbol{\xi} + \boldsymbol{\zeta} \rangle) - \sigma(\langle \mathbf{w}_{-1,r}^{(t)}, \boldsymbol{\xi} \rangle) + \sigma(\langle \mathbf{w}_{-1,r}^{(t)}, -\boldsymbol{\xi} + \boldsymbol{\zeta} \rangle) - \sigma(\langle \mathbf{w}_{-1,r}^{(t)}, -\boldsymbol{\xi} \rangle)$$
$$\leq 2 |\langle \mathbf{w}_{-1,r}^{(t)}, \boldsymbol{\zeta} \rangle|$$
$$\leq 2\lambda \cdot o(\sigma_0^2 d), \tag{H.7}$$

where the first inequality is by the Lipschitz continuous of ReLU, and the second inequality is by Lemma C.4 and G.2. Combing equation H.6 with equation H.7, we have

$$g(\boldsymbol{\xi} + \boldsymbol{\zeta}) - g(\boldsymbol{\xi}) + g(-\boldsymbol{\xi} + \boldsymbol{\zeta}) - g(-\boldsymbol{\xi})$$
$$\geq (\lambda/2) \cdot \sigma_0^2 d/2$$
$$= \frac{c\sigma_p\sigma_0\sqrt{d}}{4\sqrt{m}} = \frac{cn\sqrt{m}}{4\sqrt{d}}\text{polylog}(d)$$
$$\geq \widetilde{\Omega}\left( \frac{n\|\boldsymbol{\mu}\|_2^2}{\sigma_p^2 d} \right) \geq 4 \sum_r \sigma(\langle \mathbf{w}_{+1,r}^{(t)}, \boldsymbol{\mu} \rangle)/m.$$

Here, the second inequality is by the definition of $\sigma_0$ and the condition $\|\boldsymbol{\mu}\|_2^4 \leq \widetilde{O}(m\sigma_p^4 d)$ and $\delta = 1/\text{polylog}(d)$, the last inequality is by Proposition G.19 and $\sigma_0\|\boldsymbol{\mu}\|_2 \ll n\|\boldsymbol{\mu}\|_2^2/(\sigma_p^2 d)$ from the condition $\|\boldsymbol{\mu}\|_2 = \widetilde{\Omega}(m\sigma_p/\delta)$. Wrapping all together, the proof for the existence of $\boldsymbol{\zeta}$ completes. This completes the proof of Theorem 3.4.

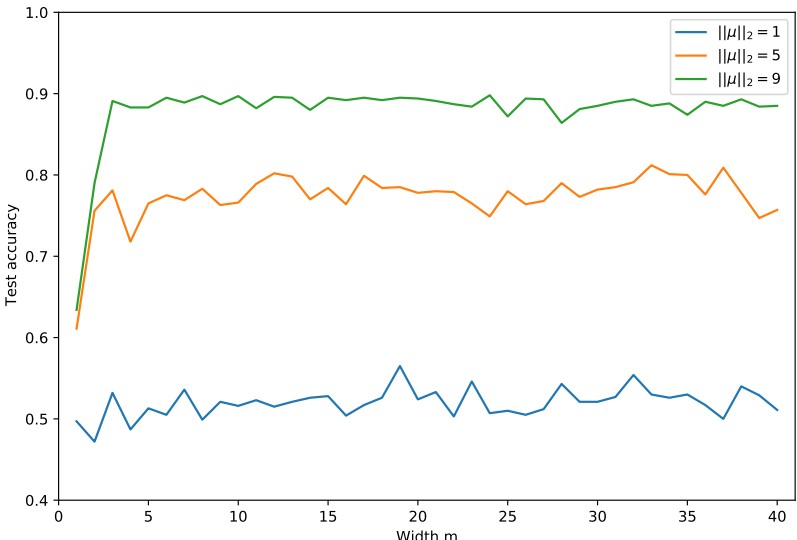

Figure 3: Test accuracy of neural networks with different widths $m$ under different signal strengths. The $x$-axis represents the value of width $m$ ranging from 1 to 40, whereas the $y$-axis is the test accuracy. The different lines on the plot represent the test accuracy achieved under different norms of $\|\boldsymbol{\mu}\|_2$.

# I  ADDITIONAL EXPERIMENTS

In this section, we conducted additional experiments to investigate the impact of the width of the neural networks on the test accuracy. We generate XOR-type data following the distribution in Definition 2.2. We fix $d = 200$ and $n = 80$, and report the test accuracy on different choices of the width $m$ based on 1000 i.i.d test data.

We give the hyper-parameters in the data distribution below. The setting is similar to Section 5. The vectors $\mathbf{a} + \mathbf{b}$ and $\mathbf{a} - \mathbf{b}$ has an angle $\theta$ with $\cos\theta = 0.8$. The directions of vectors $\mathbf{a}$ and $\mathbf{b}$ are uniformly generated as two orthogonal vectors, and their norms are determined by solving

$$\|\mathbf{a}\|_2^2 + \|\mathbf{b}\|_2^2 = \|\boldsymbol{\mu}\|_2^2, \quad \|\mathbf{a}\|_2^2 - \|\mathbf{b}\|_2^2 = \|\boldsymbol{\mu}\|_2^2 \cdot \cos\theta.$$

The signal patch $\boldsymbol{\mu}$ and the clean label $\overline{y}$ are jointly generated from $\mathcal{D}_{\text{XOR}}(\mathbf{a}, \mathbf{b})$ in Definition 2.1. The noise patch $\boldsymbol{\xi}$ is generated following the noise distribution defined in Definition 2.2 with $\sigma_p = 1$, and the observed label $y$ is given by flipping $\overline{y}$ with probability $p = 0.1$.

We consider training CNN models with different widths $m$ under the cases $\|\boldsymbol{\mu}\|_2 = 1$, $\|\boldsymbol{\mu}\|_2 = 5$ and $\|\boldsymbol{\mu}\|_2 = 9$. We initialize the model parameters as entry-wisely independent Gaussian random variables $N(0, \sigma_0^2)$ and set $\sigma_0 = 0.01$. We set the learning rate as $\eta = 10^{-3}$, and run full-batch gradient descent for $T = 200$ training epochs. The final test accuracy of neural networks with different widths are reported in Figure 3.

As shown in Figure 3, the case when $\|\boldsymbol{\mu}\|_2 = 1$ corresponds to harmful overfitting and the test accuracy is always around $0.5$ regardless of the width of the network. When $\|\boldsymbol{\mu}\|_2 = 5$ or $\|\boldsymbol{\mu}\|_2 = 9$, as the width $m$ increases, there is a significant growth of test accuracy when $m \leq 3$, and the test accuracy does not significantly change with $m$ after $m > 3$. This demonstrates that as long as $m$ is large enough, the test accuracy is not sensitive to the neural network width $m$, which aligns with our theoretical analysis. It is also worth noting that the conditions of benign and harmful overfitting in Theorem 3.4 explicitly depend on $m$. This is because our proof technique for the "asymptotically challenging XOR regime" can only cover a specific initialization scale $\sigma_0$ which depends on $m$. It remains open questions for the general initialization scales that do not depend on

$m$ in the "aysmptotically challenging XOR regime", and we believe that the test error should still not be sensitive to $m$.

