# OpenReview forum: "Benign Overfitting in Two-Layer ReLU Convolutional Neural Networks for XOR Data"
_ICLR.cc/2024/Conference — Submitted to ICLR 2024_

### Official Review · Reviewer_VjnG · 2023-10-18

**Soundness:** 3 good
**Presentation:** 3 good
**Contribution:** 3 good
**Rating:** 6
**Confidence:** 3

**Summary:**

The authors study the learning of a 2-layer CNN on a XOR-type dataset, and evidence under certain conditions the onset of benign overfitting when the number of samples is large enough. Complementarily, they provide a related lower bound establishing that below this order of magnitude for the number of samples, the test error is at least a constant away from the Bayes-optimal test error. The main novelty of the work is its data distribution, for which the Bayes-optimal classifier is not given by a linear method.

**Strengths:**

This works extends previous studies of benign overfitting to a data distribution which is not amenable to being learnt by a linear classifier, and thus better justifies the use of neural network learners. The paper is clear and well organised, and sufficient discussion of related works is provided. The experiment section at the end is a good illustration of the theory. I have not read the proof, and cannot judge of its technical novelty. Overall, I feel that this is a solid paper, which makes an interesting addition to the settings where benign overfitting occurs, and am in overall favour of its acceptance.

**Weaknesses:**

Minor:
- It would be good to cite Hu \& Lu, 2022 and Xiao et al., 2022 alongside Misiakiewicz (2022); Xiao & Pennington (2022) for completeness when discussing multiple descents in kernel ridge regression in the related works.
- More discussions would be helpful regarding the linearity of the Bayes-optimal classifier in previous studies. Was this assumption in some way instrumental in the derivation of these results? Is the phenomenology different in the present work, or is the point mainly to consider a more complex data distribution, and exhibit another setting where benign overfitting occurs? Currently, I feel this point is insufficiently motivated.

**Questions:**

- In condition 3.1, is a max or min missing in the $\Omega$ notation?

---

> ### Author Response · Authors · 2023-11-17
> **Response to Reviewer VjnG**
>
> Thank you for your supportive review. Please find our responses to your comments as follows.
>
> >**Q1**: It would be good to cite Hu \& Lu, 2022 and Xiao et al., 2022 alongside Misiakiewicz (2022); Xiao \& Pennington (2022) when discussing multiple descents in kernel ridge regression.
>
> **A1**: Thank you for pointing out the missing references. We have cited them in the revision.
>
> >**Q2**: More discussions on the linearity of the Bayes-optimal classifier. The motivation of XOR.
>
> **A2**: Previous studies investigating the benign overfitting phenomenon in learning neural networks may not fully demonstrate the power of neural networks compared to linear models, as the data used in those studies can also be well classified by linear models. In order to showcase the power of ReLU neural networks, we need to carefully select a dataset that cannot be effectively classified by linear models. The XOR data is a natural choice for such a dataset, as it can not be classified by any linear model, but can be classified by a two-layer ReLU CNN.
>
> For learning the XOR-type data, our analysis of benign overfitting demonstrates that different neurons in the network can learn different signal directions. This is in contrast to previous works that rely on learning a single signal direction (in which case all neurons with the same second layer sign may be almost the same after training). Therefore, the XOR setting better highlights the strength of the non-linear structure of neural networks. We have added this discussion in the contribution part on page 2 to emphasize the significance of our approach in demonstrating the power of ReLU neural networks.
>
> >**Q3**: In condition 3.1, is a max or min missing in the $\Omega$ notation?
>
> **A3**: Thanks for pointing it out.  We have corrected the typo and the correct version should be $d= \widetilde{\Omega}(\mathrm{max}\\{n^2,n||\boldsymbol{\mu}||_2^2\sigma_p^{-2}\\})\cdot \mathrm{polylog}(1/\varepsilon)\cdot \mathrm{polylog}(1/\delta)$.

---

> ### Comment · Reviewer_VjnG · 2023-11-20
> **Acknowledgement of rebuttal**
>
> I thank the authors for the clarifications. I remain in favour of acceptance and maintain my score.

---

### Official Review · Reviewer_vd3N · 2023-10-26

**Soundness:** 3 good
**Presentation:** 3 good
**Contribution:** 3 good
**Rating:** 6
**Confidence:** 4

**Summary:**

The work studies benign overfitting of Convolutional neural networks trained on non-linear data. The analysis focuses on a generalization of the XOR problem, where the data is centered around 4 possible vectors, giving the same label to vectors in opposite direction. This problem is non-linear in the sense that no linear classifier can approximate the target. In this setting, the authors analyze the conditions under which benign overfitting occurs. They show that benign overfitting depends on the noise level and the angle between the vectors, analogously to similar results on benign overfitting on linear data.

**Strengths:**

The study of benign overfitting in the context of non-linear data is important, and the paper seems to make novel contributions to this field of study. Most works studying benign overfitting so far focused on linear data, which arguably doesn't capture the full complexity of neural networks trained on real data.

**Weaknesses:**

1. The paper studies two-layer CNNs, which is a valid and interesting model to study. However, to my understanding the analysis is for networks convolving over only 2 input patches, which seems very restrictive. In this case, it is not clear what is the purpose of studying CNNs instead of MLPs. The authors should explain why they made this choice of architecture? Do similar results hold for standard MLPs, and if so, why do they choose to focus on 2-patch CNNs? Can the results be extended to CNNs with larger number of patches?

2. Some details about the choice of input distribution are not clear. Specifically, in the choice the randomly generated patch, why is the Gaussian covariance matrix chosen to be dependent on the "true" XOR vectors? Can you justify why such choice of random vectors is natural? Would a similar result hold for standard Normal distribution (or other more natural distributions)?

3. In stating the conditions for the theoretical analysis (Condition 3.1), it is better to separate conditions on the learning problem (which are essentially assumptions about the "world"), from choices of learning parameters (learning rate, initialization) and sample complexity. The way the conditions are stated, it seems that e.g. the dimension should grow with the sample complexity. It makes more sense to first assume a problem is given with some fixed dimension, and then adapt the sample size to the problem (in which case, does this mean that the sample size needs to be upper bounded?).

4. The authors state the necessary and sufficient conditions for benign overfitting to be $n ||\mu||_2^4=\Omega(\sigma_p^4d)$. Can you interpret this results? In what regimes of parameters does this hold? Is this always compatible with Condition 3.1?

Minor:
- First sentence in the introduction: highly parameterized => over-parameterized
- "To solve the chaos caused by the high correlation in the training process" - not sure "chaos" is the right term..
- Page 5: "These results also ensure that" - I think you mean conditions

**Questions:**

See above

---

> ### Author Response · Authors · 2023-11-17
> **Response to Reviewer vd3N**
>
> Thank you for your supportive comments and suggestions. We address your comments as follows.
>
> >**Q1**: 2 input patches seem very restrictive. Can the results be extended to CNNs with a larger number of patches? Why study CNNs instead of MLPs? Do similar results hold for standard MLPs?
>
> **A1**: The two-layer CNN model considered in this paper follows the same definition as in [1,2,3]. Considering a case with two patches is just for the simplicity of the presentation, and our analysis can be easily extended to the case where the data inputs consist of multiple patches. The proof will be almost identical to the current proof given in the paper.
>
> The primary goal of this paper is to explore the capabilities of non-linear neural networks in learning complex and "non-linear" data. We believe our analysis can be extended to MLPs with some necessary modifications, and the condition for benign overfitting can still be derived as a function of signal strength, noise level, sample size and data dimension, similar to what we have presented in Theorem 3.2.
>
> >**Q2**: Why is the Gaussian covariance matrix chosen to be dependent on the "true" XOR vectors? What about standard Normal distribution?
>
> **A2**: We define the covariance matrix of the Gaussian vector in this way so that the noise vectors are orthogonal to the signal vectors. This is to simplify the setting and give a clearer and more refined analysis. Similar definitions have been studied in [1,2,3]. If the noise is given by standard Gaussian, the general proof idea does not change but we will need to rely on the fact that in the high dimensional setting (when $d$ is sufficiently large), the noise and signal vectors will be almost orthogonal to each other (with high probability). Therefore, the proof details will be more complicated and stronger assumptions on $d$ may be required.
>
> >**Q3**: In Condition 3.1, it seems that the dimension should grow with the sample complexity. It is better to separate conditions on the learning problem from choices of hyper-parameters and sample complexity. It makes more sense to assume a problem given with some fixed dimension, and then adapt the sample size. Does it mean that the sample size needs to be upper bounded?
>
> **A3**: We clarify that this paper indeed studies the case where the dimension $d$ grows with the sample size $n$. This is because the goal of this paper is to study the phenomena of benign and harmful overfitting. Achieving overfitting necessitates the growth of dimension $d$ alongside sample size $n$. And we indeed need to assume that the sample size $n$ is upper bounded by a quantity related to $d$ so that the problem setting allows overfitting. It is noteworthy that the setting where $d$ increases with $n$ is also a common scenario studied in learning theory (e.g., [4,5,6]).
>
> >**Q4**: Condition for benign overfitting is $n||\boldsymbol{\mu}||_2^4=\Omega(\sigma_p^4d)$. Can you interpret this result? In what regimes of parameters does this hold? Is this always compatible with Condition 3.1?
>
> **A4**: To interpret the results, we can see that the condition in Theorem 3.2 indicates that large $||\boldsymbol{\mu}||_2$ and sample size $n$ can help achieve benign overfitting, while large noise level $\sigma_p$ and data dimension $d$ can lead to harmful overfitting. The regime for Theorem 3.2 to hold is exactly Condition 3.1. Regarding whether Condition 3.1 is compatible with the benign overfitting condition $n|||\boldsymbol{\mu}||_2^4=\Omega(\sigma_p^4d)$, we can see that Condition 3.1 imposes a condition on the signal strength $||\boldsymbol{\mu}||_2$ requiring $||\boldsymbol{\mu}||_2\leq \sigma_p\sqrt{d/n}$, and the condition for benign overfitting in Theorem 3.2 is $||\boldsymbol{\mu}||_2\geq \sigma_p (d/n)^{1/4}$. Clearly, these two conditions do not conflict with each other as long as $d>n$.
>
> >**Q5**: Typos.
>
> **A5**: Thank you for pointing out the typos. We have revised the paper and fixed the typos.
>
> References:
>
> [1] Yuan Cao, Zixiang Chen, Mikhail Belkin and Quanquan Gu. Benign overfitting in two-layer convolutional neural networks. NeurIPS, 2022.
>
> [2] Xuran Meng, Yuan Cao and Difan Zou. Per-example gradient regularization improves learning signals from noisy data.  arXiv:2303.17940, 2023.
>
> [3] Samy Jelassi and Yuanzhi Li. Towards understanding how momentum improves generalization in deep learning. ICML, 2022.
>
> [4] Spencer Frei, Niladri S Chatterji and Peter Bartlett. Benign overfitting without linearity: Neural network classifiers trained by gradient descent for noisy linear data. COLT, 2022.
>
> [5] Song Mei and Andrea Montanari. The generalization error of random features regression: Precise asymptotics and the double descent curve. CPAM, 2022.
>
> [6] Zeyuan Allen-Zhu and Yuanzhi Li. Feature purification: How adversarial training performs robust deep learning. FOCS, 2022.

---

### Official Review · Reviewer_X4FC · 2023-10-31

**Soundness:** 3 good
**Presentation:** 3 good
**Contribution:** 3 good
**Rating:** 5
**Confidence:** 3

**Summary:**

This paper studies benign overfitting on neural networks, using gradient descent to train XOR-type data on a two-layer ReLU CNN. The authors establish the conditions for when benign overfitting and harmful overfitting occurs in this setting.

**Strengths:**

Compared to previous work, this paper goes beyond the scope of simple learning problems where the Bayes-optimal classifier is linear.

**Weaknesses:**

1: Compared to the results in the Classic XOR regime (Section 3.1), the results in Section 3.2 are of greater interest. However, this part of the results relies on strong assumptions. Firstly, the first point of Condition 3.3 is a stronger high-dimensional assumption compared with other benign overfitting papers such as [1]. Then, considering Condition 3.3, we can derive the conditions in the second point of Theorem 3.4 as $\|\| \mu \|\|_2^2 \geq \frac{m^5 n^2 \sigma_p^2}{(1-\cos \theta)^2}$ which is much stronger than [1].

2: I do not agree with the author's claim that they have discovered a sharp phase transition between benign overfitting and harmful overfitting, as in the case of Theorem 3.2, the transition between the second and third points may not necessarily be a phase transition but rather a continuous change.

**Questions:**

Could the author discuss the relationship and differences between this work and [2]? And is there any connection between this work and the Grokking?

[1] Spencer Frei, Niladri S Chatterji, and Peter Bartlett. Benign overfitting without linearity: Neural network classifiers trained by gradient descent for noisy linear data. In Conference on Learning Theory, pages 2668–2703. PMLR, 2022.

[2] Zhiwei Xu, Yutong Wang, Spencer Frei, Gal Vardi, and Wei Hu. Benign Overfitting and Grokking in ReLU Networks for XOR Cluster Data.  arxiv 2310.02541.

---

> ### Author Response · Authors · 2023-11-17
> **Response to Reviewer X4FC**
>
> Thank you for your review and feedback. We address your questions and concerns as follows.
>
> >**Q1**: Compared to Section 3.1, the results in Section 3.2 are of greater interest.
>
> **A1**: We believe that the classic XOR regime (Section 3.1) and the asymptotically challenging XOR regime (Section 3.2) are equally important. As commented in our paper, Section 3.1 studies the regime that covers the standard XOR problem, and therefore it is more appropriate to compare the results in Section 3.1 with previous works. In comparison, Section 3.2 discusses a less standard, but more challenging setting, which requires different theoretical analyses. Therefore, we believe that the two regimes are both important.
>
> >**Q2**: Section 3.2 relies on strong assumptions. Condition 3.3 and the condition for benign overfitting in Theorem 3.4 are stronger than [1].
>
> **A2**: We respectfully disagree with your comment. We strongly believe this comparison is unfair, and the weakness pointed out here is unreasonable. Our work investigates the XOR-type data, where the Bayes optimal classifier is nonlinear. For this XOR-type learning problem, the difficulty of learning (or how the noiseless samples are separated) depends on both the norms of the signal vectors and the angle between them. And the setting studied in Section 3.2 is a particularly challenging setting where the angle can be asymptotically small. In comparison, [1] only studied a much simpler learning problem where the Bayes-optimal classifier is linear, and in the setting of [1], there is no such concept of the angle between signals. Therefore, the settings in [1] and our work are not comparable, and the comparison between the rates in [1] and in Section 3.2 is unfair.
>
> >**Q3**: The transition between benign and harmful overfitting is not necessarily a phase transition but rather a continuous change.
>
> **A3**:  We would like to explain that the terminology "sharp phase transition" has been used in existing works [3,4] to describe the same type of transition between benign and harmful overfitting. Therefore, "sharp phase transition" is not invented by this paper, and we use this terminology mainly to highlight that the same type of transition between benign and harmful overfitting also happens in learning the XOR-type data.
>
> We have also clarified in the revision (on Page 5) that by  "sharp phase transition", we mean that in "boundary cases", an increase of the sample size $n$ or the signal strength $|| \boldsymbol{\mu} ||\_2$ by a logarithmic factor is sufficient to change the test error from a constant level to the optimal rate $p+o(1)$.
>
> >**Q4**: The relationship and differences between this work and [2].
>
> **A4**: We would like to first point out that our paper was submitted to ICLR before the appearance of [2] on arXiv, and it is generally unreasonable to make a comparison between our paper and [2]. To answer your questions, we list several key differences between our work and [2] as follows:
>
> Differences in problem settings:
>
> - Our data model covers varying angles between the signal vectors, while [2] assumes that the signal vectors in the XOR data are orthogonal (therefore, [2] may be more comparable with our results in Section 3.1). Moreover, we consider the setting where the signals and noises are on different patches, while [2] considers the setting where the signals and noises are added together.
>
> - We analyze CNN models, while [2] studies fully connected neural networks.
>
> Differences in results:
>
> - [2] requires $||\boldsymbol{\mu} ||_2^2 = \Omega( n^{0.51} \sqrt{d} )$, and the signal strength needs to increase as the sample size increases. In comparison, our results indicate that if the sample size $n$ increases, then a smaller signal strength $|| \boldsymbol{\mu} ||_2$ may be sufficient to achieve benign overfitting.
>
> - [2] does not give any direct guarantees of training loss minimization, while our results show that gradient descent can achieve an arbitrarily small training loss.
>
> - The results in [2] are for neural networks trained up to $\sqrt{n}$ iterations. In comparison, our results are for CNNs trained until convergence.
>
>
> - [2] establishes an upper bound of the test error of the trained network, while we establish upper and lower bounds of the test error under complimentary conditions, demonstrating that these upper and lower bounds are both tight.
>
> References:
>
> [1] Spencer Frei, Niladri S Chatterji and Peter Bartlett. Benign overfitting without linearity: Neural network classifiers trained by gradient descent for noisy linear data. COLT, 2022.
>
> [2] Zhiwei Xu, Yutong Wang, Spencer Frei, Gal Vardi and Wei Hu. Benign Overfitting and Grokking in ReLU Networks for XOR Cluster Data. arxiv 2310.02541.
>
> [3] Yiwen Kou, Zixiang Chen, Yuanzhou Chen and Quanquan Gu. Benign overfitting for two-layer relu networks. COLT, 2023.
>
> [4] Yuan Cao, Zixiang Chen, Mikhail Belkin and Quanquan Gu. Benign overfitting in two-layer convolutional neural networks. NeurIPS, 2022.

---

> > ### Comment · Reviewer_X4FC · 2023-11-20
> >
> > Dear Authors,
> >
> > Thank you for your response.
> >
> > The authors clarified that the "sharp phase transition" originated from previous work and elucidated the intended meaning here. However, I still have concerns about the presentation. I think such a statement can be used after explaining that there is a constant between $C_3$ and $C_1$ to satisfy the upper bound of the second part in Theorem 3.2 is smaller than the lower bound of the third part. I hope the authors can provide further explanation or make modifications or additions to this part.
> >
> > Thank you,
> >
> > Reviewer X4FC

---

> > > ### Author Response · Authors · 2023-11-21
> > > **Thanks for your further response**
> > >
> > > Dear reviewer:
> > >
> > > Thanks for your further response.  We guess that you have concerns on the gap between $C_1$ and $C_3$. In fact, Theorem 3.2 can be modified to the version where the conditions for the second and third part are given using a same absolute constant $C_1$. This modification is simply done by adjusting the constants. We have revised the statement of Theorem 3.2 and the proof (Appendix F) accordingly in the latest version.
> > >
> > > Hope this can address your concerns. We are happy to answer any further questions.
> > >
> > > Authors

---

> > > > ### Comment · Reviewer_X4FC · 2023-11-21
> > > >
> > > > Dear Authors,
> > > >
> > > > Thanks for the reply. I will increase the score to 5 since I still have doubts about the relatively harsh high-dimensional conditions of this paper.
> > > >
> > > > Thank you,
> > > >
> > > > Reviewer X4FC

---

> > > > > ### Author Response · Authors · 2023-11-23
> > > > >
> > > > > Dear Reviewer,
> > > > >
> > > > > Thank you for actively discussing with us and we are glad to see that you have raised your score.
> > > > >
> > > > > Regarding the high-dimensional conditions, we would like to point out that the condition on the dimension in Section 3.1 is among the mildest in related works on benign overfitting in classification [1,2,3,4]. In fact, even for the study of linear logistic regression, the same assumption on the dimension has been made [1]. Therefore, we believe our high-dimensional conditions are reasonable.
> > > > >
> > > > > The high-dimensional condition in Section 3.2 is more strict than Section 3.1. But as we have explained in our previous response, this is only because Section 3.2 considers a particularly challenging setting that has not been covered by previous works. In fact, the conditions on the dimension in Section 3.2 is still milder than the condition in [3].
> > > > >
> > > > > We hope that our explanation above can address your concerns on the high-dimensional conditions.
> > > > >
> > > > > Thank you!
> > > > >
> > > > > Authors
> > > > >
> > > > >
> > > > > [1] Niladri S. Chatterji and Philip M. Long. Finite-sample analysis of interpolating linear classifiers in the overparameterized regime. JMLR, 2021.
> > > > >
> > > > > [2] Spencer Frei, Niladri S. Chatterji and Peter Bartlett. Benign overfitting without linearity: Neural network classifiers trained by gradient descent for noisy linear data. COLT, 2022.
> > > > >
> > > > > [3] Yuan Cao, Zixiang Chen, Mikhail Belkin and Quanquan Gu. Benign overfitting in two-layer convolutional neural networks. NeurIPS, 2022.
> > > > >
> > > > > [4] Yiwen Kou, Zixiang Chen, Yuanzhou Chen and Quanquan Gu. Benign overfitting for two-layer relu networks. COLT, 2023.

---

### Official Review · Reviewer_jBsa · 2023-10-31

**Soundness:** 3 good
**Presentation:** 3 good
**Contribution:** 3 good
**Rating:** 6
**Confidence:** 4

**Summary:**

This paper considers the benign overfitting phenomenon of two-layer ReLU CNNs in an XOR problem and provides the generalization (test) error of the learned CNNs via gradient descent.

**Strengths:**

This paper demonstrates that CNNs can efficiently learn XOR-type data. Most importantly, this paper provides the test error bound for the learned model in the regime where the features in XOR-type data are highly correlated.

**Weaknesses:**

1. It is unclear how the test error changes as the number of samples, and CNN width change.

2. As the paper studies the benign overfitting of neural networks, a figure of test accuracy versus the neural network width $m$ should be included.

**Questions:**

Do we have any requirements on the number difference between two different data points? I am not clear about the high-level understanding of Lemma 4.1. If the data points are highly unbalanced, how do you obtain a result like Lemma 4.1?

---

> ### Author Response · Authors · 2023-11-17
> **Response to Reviewer jBsa**
>
> Thank you for your thoughtful review and support. We address your comments as follows.
>
> >**Q1**: It is unclear how the test error changes as the number of samples and CNN width change.  A figure of test accuracy versus the neural network width $m$ should be included.
>
> **A1**: Our results in Theorem 3.2 suggest that as the sample size increases, the test error should decrease. Regarding the CNN width $m$, Theorem 3.2 indicates that the test error is not sensitive to $m$ as long as $m$ is large enough ($m = \Omega(\log(n/\delta))$ as is given in Condition 3.1). This is because in learning XOR problems,  the test error is mainly affected by how much "signal" (the $\boldsymbol{\mu}$ vectors in Definitions 2.1 and 2.2) the CNN filters can learn from the training data. As long as $m$ is large enough to ensure a diverse enough random initialization of the CNN filters, further increasing $m$ does not significantly change the training dynamics of the CNN model. We also added a new set of experiments in Appendix I to demonstrate the relation between $m$ and the test error. As shown in Figure 3, for $m \geq 3$, the test error does not significantly change with $m$. This backs up our claim that as long as $m$ is large enough, the test error is not sensitive to $m$.
>
> It is worth noting that the conditions of benign and harmful overfitting in Theorem 3.4 explicitly depend on $m$. This is because our proof technique for the asymptotically challenging XOR regime can only cover a specific initialization scale $\sigma_0$ that depends on $m$. For general initialization scales that do not depend on $m$, we believe that the test error should not be sensitive to $m$.
>
> >**Q2**: A result like Lemma 4.1 in a highly unbalanced case? Any requirements on the number difference between different data points?
>
> **A2**: The balance among different types of data is implicitly given in Definitions 2.1 and 2.2. Following the definition of $\mathcal{D}\_{\mathrm{XOR}}(\mathbf{a},\mathbf{b})$ in Definition 2.1, it is easy to see that the different types of data are generated equally likely.
>
> Our results in both the "classic XOR regime" and the "asymptotically challenging XOR regime" rely on the balance of the data. In the "classic XOR regime", we believe that Lemma 4.1 will still hold with unbalanced data. However, the subsequent analysis on the training dynamics will be significantly affected by unbalanced data. For instance, Lemma E.1 in Appendix E, which is a key lemma to show how much "signals" (the $\boldsymbol{\mu}$ vectors in Definitions 2.1 and 2.2) the CNN filters can learn from the training data, will no longer be valid. Similarly, in the "asymptotically challenging XOR regime", when the data are highly unbalanced, the results in Lemma G.17 will no longer hold.
>
> >**Q3**: I am not clear about the high-level understanding of Lemma 4.1.
>
> **A3**: Lemma 4.1 shows that during training, the loss derivatives at each data point (i.e.,  $\ell’^{(t)}_i$, $i\in [n]$) maintain a constant level ratio over each other. This implies that each training data point is learned by the CNN model at a similar speed. The intuition that this result can hold is that in the high dimensional setting (where $d$ is sufficiently large), with high probability, the training data points should contain noise patches ($ \boldsymbol{\xi}_i$) that are almost orthogonal to each other, and are almost of the same length. This maintains a level of “symmetry” in the training data, which enables us to prove Lemma 4.1.

---

### Official Review · Reviewer_EAhF · 2023-11-03

**Soundness:** 4 excellent
**Presentation:** 3 good
**Contribution:** 3 good
**Rating:** 5
**Confidence:** 4

**Summary:**

This paper considers learning an XOR-type problem with an overparameterized two-layer neural convolutional ReLU network using the batch loss, where the input label can flip with a constant probability. This paper proves the convergence w.r.t. the empirical loss, and shows that there exists a certain condition on the sample complexity and signal-to-noise ratio which distinguishes whether near Bayes-optimal test error is achieved or the prediction accuracy is a constant away from the Bayes-optimal rate.

**Strengths:**

### The problem is well-motivated from both benign overfitting and feature learning literature.

I understand that proving benign overfitting for neural networks involves several difficulties due to nonlinearity of activation functions. On the other hand although recent works have shown the possibility of learning nonlinear features especially XOR data via gradient-based training, I am not aware of a result that the network can memorize label flipping noise of the training data as well. Therefore, this paper is a naturally motivated by both lines of researches.

### Solid technical contribution under reasonable assumptions and worth sharing to the theory community.

I need to note that the data distribution is not a standard XOR setting considered in pervious feature learning analyses ([Ji & Telgarsky (2020)](https://arxiv.org/abs/1909.12292); [Barak et al. (2022)](https://arxiv.org/abs/2207.08799); [Telgarsky (2023)](https://arxiv.org/pdf/2208.02789.pdf)), and requires larger signal-to-noise ratio (signal $\|\mu\|$ or noise $\sigma_p^{-1}$ should be large.). Except for that points, several technical assumptions are rather acceptable and the main result still has a solid contributions to the theory community as the first nonlinear Bayes optimal classifier.

### A sharp transition from sub-optimal classifier to the near Bayes-optimal classifier is interesting.

This paper derives the threshold where benign overfitting happens, analogous to the previous linear classifiers.

**Weaknesses:**

### Large signal-to-noise ratio

In p.5, the authors claim that
> However, we note that our data model with $\|\mu\|_2 = \sigma_p = \Theta(1)$ would represent the same signal strength and noise level as in the 2-sparse parity problem, and in this case, our result implies that two-layer ReLU CNNs can achieve near Bayes-optimal test error with $n=\omega(d)$, which is better than the necessary condition $n=\omega(d^2)$ for kernel methods.

However, as far as I understand, this explanation is wrong; by letting $\|\mu\|_2 = \sigma_p = \Theta(1)$, Condition 3.1 implies $d=\Omega(n^2)$ is required, while benign overfitting (in Theorem 3.2) requires $n =\Omega(d)$. To solve this issue, either $\|\mu\|_2$ or $\sigma_p$ should be larger than $O(1)$, in the asymptotic limit when $d$ and $n$ diverge. In other words, the signal-to-noise ratio should be larger than the standard analysis on the XOR data ([Ji & Telgarsky (2020)](https://arxiv.org/abs/1909.12292); [Barak et al. (2022)](https://arxiv.org/abs/2207.08799); [Telgarsky (2023)](https://arxiv.org/pdf/2208.02789.pdf)), although [Ba et al. (2023)](https://openreview.net/forum?id=HlIAoCHDWW) considers the anisotropic input data in the regression problem.

### Dependency on the initialization scale.

Suppose that we take $d=\Theta(n^2)$ and $\sigma_p=\Theta(1)$ in Condition 3.1. Then, the initialization scale is $d^{-0.75}$, which is relatively small. This means that the output scale of the network at initialization is $o(1)$.

### Motivation for considering the small angle case

I am basically satisfied with the first result (Theorem 3.2), but I do not see necessity to extend the problem into more general case, where two axes of the input XOR distribution is much closer.

**Questions:**

- I want to know how the small initialization (as I pointed out) is required in the analysis.

- In the proof sketch, the authors introduced "loss derivative comparison" technique. If the authors could provide more explanations about the motivation of such technique, I would appreciate it.

---

> ### Author Response · Authors · 2023-11-17
> **Response to Reviewer EAhF**
>
> Thank you for your detailed review and constructive comments. We address your questions as follows.
>
> >**Q1**: Comparison with the learning of 2-sparse parity is not rigorous: $n = \Omega(d)$ is not covered in Condition 3.1. The SNR in this paper needs to be larger than (Ji \& Telgarsky (2020); Barak et al. (2022); Telgarsky (2023)), Ba et al. (2023).
>
> **A1**: Thank you for pointing this out. We have revised the manuscript to emphasize that the comparison is not rigorous because Condition 3.1 requires $d = \tilde\Omega(n^2)$, which excludes the case $n=\omega(d)$. We have also discussed that the condition $d = \tilde\Omega(n^2)$ in Condition 3.1 is only a technical requirement to enable our current theoretical analysis (e.g. the proof of Lemma D.3). Therefore, Condition 3.1 can potentially be relaxed to cover the case $n=\omega(d)$ with different analysis tools, which can be an interesting future work direction. We have also cited the missing references in our discussion.
>
> >**Q2**: The initialization scale is small. How is the small initialization required in the analysis??
>
> **A2**: We assume a relatively small initialization to ensure sufficient "feature learning":
> If the initialization is sufficiently small, the training updates (which consist of features from training data) of neural network parameters can quickly dominate their random initialization, thus during  training, the neural network can achieve sufficient feature learning. On the other hand, if the initialization is large, then it may dominate the training updates, which can lead to the "Neural Tangent Kernel (NTK) regime" where the model essentially uses kernels or random features [1] to fit the training data. Our analysis in this paper does not fall in the NTK regime, and we study the case where training updates dominate the random initialization. This is why we assume that the initialization is sufficiently small. Similar conditions of small initialization have been widely assumed such as [2,3,4,5].
>
> >**Q3**:  I do not see the necessity to extend the problem into the small angle case, where two axes of the input XOR distribution are much closer.
>
> **A3**: Theorem 3.4 aims to study a more challenging case where the data are less separated. For simplicity, here we can consider a case without label flipping. In this case, the angle $\theta$ can be understood as the metric of the margin of the data, and a smaller angle means a smaller margin. For this more challenging case, our proof technique in Theorem 3.2 can no longer be applied, and therefore Theorem 3.4 and its proof may also have their unique technical value. This is why we also cover this setting in our paper.
>
> >**Q4**:  More explanations about the motivation of the "loss derivative comparison" technique.
>
> **A4**: The loss derivative comparison results serve as the key to analyze the training of the CNN with equation 4.1. Specifically, the result of loss derivative comparison in the "classic XOR regime" is given in Lemma 4.1 and the result in the "asymptotically challenging XOR regime" is given by Lemma 4.4.
>
> The motivation of these results is that in equation 4.1, we hope to show that the terms without the $\cos\theta$ factor can dominate the terms with the $\cos\theta$ factor, which can significantly simplify our analysis. For example, with Lemma 4.1, we can see that in the "classic XOR regime" where $\cos\theta < 1/2$, for any $i,i'\in[n]$, a non-zero $\ell'^{(t)}\_i\cdot\boldsymbol{1}\\{\langle\mathbf{w}^{(t)}\_{+1,r},\boldsymbol{\mu}\_i\rangle>0\\}||\boldsymbol{\mu}||\_2^2$ will always dominate $\ell'^{(t)}\_{i'}\cdot \boldsymbol{1}\\{\langle\mathbf{w}^{(t)}\_{+1,r},\boldsymbol{\mu}\_{i'}\rangle>0\\}||\boldsymbol{\mu}||\_2^2\cos\theta$. For the "asymptotically challenging XOR regime", since a larger $\cos\theta$ that is close to $1$ is considered, we need the tighter bound on the ratios between loss derivatives given in Lemma 4.4.
>
> We have added more explanations of the motivation of "loss derivative comparison" in the revision.
>
> References:
>
> [1] Arthur Jacot, Franck Gabriel and Clement Hongler. Neural tangent kernel: Convergence and generalization in neural networks. NeurIPS, 2018.
>
> [2] Yiwen Kou, Zixiang Chen, Yuanzhou Chen and Quanquan Gu. Benign overfitting for two-layer relu networks. COLT, 2023.
>
> [3] Yuan Cao, Zixiang Chen, Mikhail Belkin and Quanquan Gu. Benign overfitting in two-layer convolutional neural networks. NeurIPS, 2022.
>
> [4] Zeyuan Allen-Zhu and Yuanzhi Li. Feature purification: How adversarial training performs robust deep learning. FOCS, 2022.
>
> [5] Spencer Frei, Niladri S Chatterji and Peter Bartlett. Benign overfitting without linearity: Neural network classifiers trained by gradient descent for noisy linear data. COLT, 2022.

---

### Author Response · Authors · 2023-11-23
**Message to All Reviewers**

Dear Reviewers,

We sincerely appreciate the valuable feedback you have provided. We believe that all of your concerns have been well addressed in our response, and the paper has been further improved in the revision thanks to your constructive comments. Since the author-reviewer discussion period is ending, we genuinely hope that you could check whether you are satisfied with our response and revision.

Thank you!

Best regards,

Authors of Submission7181

---

### Meta-Review · Area_Chair_9ZMs · 2023-12-06

**Metareview:**

This work provides bounds on benign overfitting in a model with non-linearly separable data learned by a convolutional architecture. While I want to reassure you that I did not take into account in my evaluation of novelty works that appeared after the ICLR deadline or close to it, the paper was considered of borderline interest as the architecture studied considers only 2 patches, and while a generalization was claimed easily in the rebuttal, it was not presented in the paper or promised to be added. The paper was also unclear about existing works on benign overfitting that consider non-linear data. Such works are discussed, both in the paper and the rebuttal, but even after this in the discussion with referees, the significance of the differences was not clear. None of the referees was really enthusiastic about the results and defended acceptance. The decision is hence towards rejection with an encouragement to clarify the raised issues and resubmit the work.

**Justification For Why Not Higher Score:**

This paper could be accepted. All reviewers agreed it had limited novelty in terms of the considered model, and the considered architecture has limited interest, but at the same time, it provides a contribution to a well-studied topic that seems solid and correct.

**Justification For Why Not Lower Score:**

N/A

---

### Decision · Program_Chairs · 2024-01-16

Reject